# REFINING DUAL SPECTRAL SPARSITY IN TRANSFORMED TENSOR SINGULAR VALUES

## ABSTRACT

The Tensor Nuclear Norm (TNN), derived from the tensor Singular Value Decomposition, is a central low-rank modeling tool that enforces *element-wise sparsity* on frequency-domain singular values and has been widely used in multi-way data recovery for machine learning. However, as a direct extension of the matrix nuclear norm, it inherits the assumption of *single-level spectral sparsity*, which strictly limits its ability to capture the *multi-level spectral structures* inherent in real-world data, particularly the coexistence of low-rankness within and sparsity across frequency components. To address this, we propose the tensor $\ell_p$-Schatten-$q$ quasi-norm ($p, q \in (0, 1]$), a new metric that enables *dual spectral sparsity control* by jointly regularizing both types of structure. While this formulation generalizes TNN and unifies existing methods such as the tensor Schatten-$p$ norm and tensor average rank, it differs fundamentally in modeling principle by coupling global frequency sparsity with local spectral low-rankness. This coupling introduces significant theoretical and algorithmic challenges. To tackle these challenges, we provide a theoretical characterization by establishing the first minimax error bounds under dual spectral sparsity, and an algorithmic solution by designing an efficient reweighted optimization scheme tailored to the resulting nonconvex structure. Numerical experiments demonstrate the effectiveness of our method in modeling complex multi-way data.

## 1 INTRODUCTION

Modeling latent structural patterns in high-dimensional signals is a fundamental challenge across domains such as machine learning and signal processing (Liu et al., 2020; Zhang et al., 2014; Lu et al., 2019a). Real-world datasets are often inherently multi-modal and high-dimensional (tensor-form), containing intricate dependencies that cannot be adequately captured by naïve modeling or vector/matrix-based representations (Cichocki et al., 2016). A common strategy to uncover these relationships is to impose a *low-rank* prior, which isolates essential information and reduces the degrees of freedom, focusing on the principal components of the signal (Martin et al., 2013; Bergqvist & Larsson, 2010). Traditional tensor decomposition methods, such as CANDECOMP/PARAFAC (CP) (Carroll & Chang, 1970), Tucker (Tucker, 1966), and Tensor Train (Oseledets, 2011), have been widely used to model tensor signals (Cichocki et al., 2016; Liu et al., 2013; Imaizumi et al., 2017; Yuan & Zhang, 2016). While effective in certain scenarios, these methods rely on the assumption of intrinsic low-rankness in the *original domain*, which may fail to hold in complex, real-world applications. This limitation has led to the development of *transformed-domain* low-rankness, where linear transformations like the Discrete Fourier Transform (DFT) are applied to reveal more pronounced low-rank patterns. Within this paradigm, the tensor Singular Value Decomposition (t-SVD) has emerged as a powerful framework with notable success in applications such as image and video analysis (Liu et al., 2020; Zhang et al., 2014; Xie et al., 2018; Wang et al., 2025b).

Building on the t-SVD framework, the Tensor Nuclear Norm (TNN) has become an extensively adopted regularizer for low-rank tensor modeling (Lu et al., 2019b; Zhang et al., 2014; Song et al., 2020; Hou et al., 2021; Zhang & Ng, 2021; Lu, 2021; Zhou & Feng, 2017). By extending the matrix nuclear norm to the tensor setting, TNN promotes low-rankness by enforcing *element-wise sparsity* on singular values in the transformed domain (Li et al., 2019; Zhang et al., 2014). This formulation effectively captures low-rank dependencies within individual frequency components.

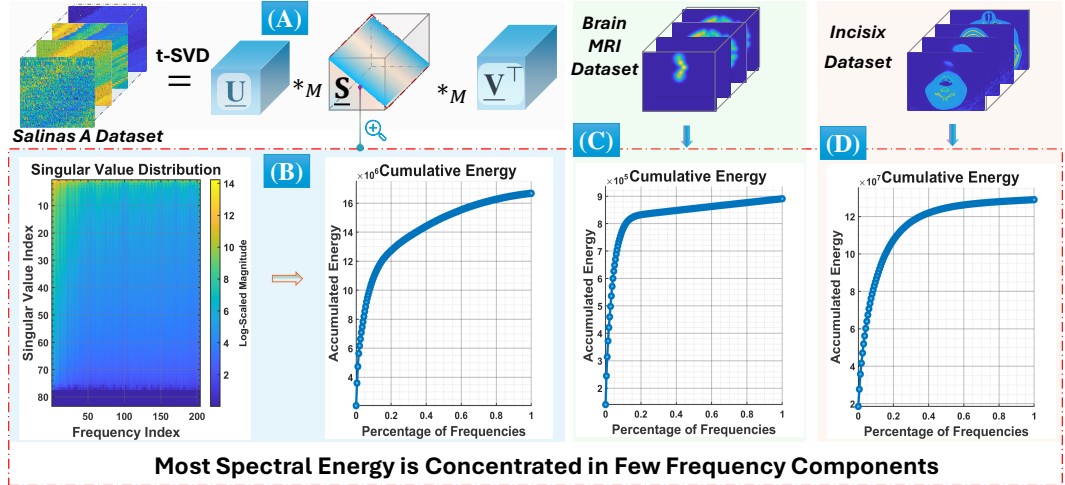

Figure 1: Empirical illustration of dual spectral sparsity in the transformed domain (DCT) via t-SVD. **(A)** The t-SVD decomposes a tensor into transformed-domain singular structures. **(B)**-Left: Singular value heatmap of the *Salinas A* dataset under t-SVD, with each column represents one frequency slice. Vertical decay reveals intra-frequency low-rankness, while horizontal variation indicates sparsity across frequencies. **(B)**-Right, **(C)**, **(D)**: Cumulative energy curves for *Salinas A*, *Brain MRI*, and *Incisix* datasets show that over 80% of total spectral energy is concentrated in the top 15%–30% frequency components, confirming frequency-wise sparsity.

However, a long-overlooked limitation of TNN lies in its assumption of *uniform spectral regularization*, which treats all frequency components equally regardless of their relative importance. From a signal processing perspective, this *single-level sparsity* design fails to account for the *two-level structure* prevalent in transformed tensor data. Real-world tensors often exhibit low-rank behavior within each frequency component together with sparsity across the frequency domain. As illustrated in Fig. 1 and confirmed quantitatively in Appendix D.10, analyses of hyperspectral images, multispectral images, medical imaging volumes and videos consistently show that a small subset of frequency slices concentrates most of the spectral energy, and that these dominant slices exhibit pronounced low-rank structure within each frequency component. This *dual spectral sparsity* pattern in real tensor data reveals that sparse frequency activation and low-rank organization within active slices emerge as stable and complementary characteristics.

**Main Research Questions.** The dual spectral sparsity necessitates a more flexible model capable of characterizing inter-frequency sparsity and intra-frequency low-rankness as two distinct structural components, moving beyond the limitations of uniform schemes like TNN. This motivation leads to three central research questions:

**RQ1 (Modeling):** *How can a tensor model explicitly represent sparsity across frequency slices and low-rankness within active slices?*

**RQ2 (Theory):** *What statistical guarantees can be established for such a two-level spectral structure?*

**RQ3 (Algorithm):** *Can we design an efficient solver for the nonconvex, coupled spectral structure, so that the benefits of modeling this two-level sparsity can be realized in tensor learning tasks?*

**Contributions.** To address these questions, we propose the *tensor $\ell_p$-Schatten-q quasi-norm*, a novel framework introducing *dual spectral sparsity control* to simultaneously model both within-frequency and across-frequency dependencies. While our framework offers promising modeling capabilities, the *coupled nature of this dual spectral sparsity* introduces significant theoretical and computational challenges. Our main contributions in developing and validating this framework are as follows:

- **Structural Modeling (RQ1):** To the best of our knowledge, this work is the first to formalize the prevalent dual spectral sparsity within the t-SVD framework, where inter-frequency sparsity coexists with intra-frequency low-rankness (Section 3). The proposed $\ell_p$-Schatten-q quasi-norm jointly models both inter-frequency sparsity and intra-frequency low-rankness, while allowing

separate control over each via parameters $p$ and $q$. This framework generalizes TNN, unifying existing methods such as the tensor Schatten-$p$ quasi-norm (Kong et al., 2018) and tensor average rank (Wang et al., 2021b) into a single, versatile framework.

- **Theoretical Guarantees (RQ2):** We establish matched minimax lower and upper bounds for tensor estimation under dual spectral sparsity, covering both hard and soft regimes (Section 4). The analysis introduces new techniques to characterize the complexity of coupled parameter spaces, extending classical tools such as covering numbers and metric entropy to the tensor spectral setting.

- **Optimization and Empirical Validation (RQ3):** We develop a scalable proximal algorithm tailored to the proposed quasi-norm (Section 5). It employs a reweighted approximation and frequency-wise singular value updates in the transform domain, effectively handling the nonconvexity and structural coupling induced by dual spectral sparsity. Experiments on real-world tensor recovery and clustering tasks demonstrate the potential applicability of our method (Section 6).

The remainder of the paper is organized as follows. Section 2 reviews basic preliminaries. Section 3 introduces the proposed quasi-norm. Sections 4 and 5 present the theoretical analysis and optimization algorithm, respectively. Experimental results are reported in Section 6, followed by the conclusion in Section 7. Details on related work, proofs, algorithms, and experiments are provided in the appendix.

## 2 NOTATIONS AND PRELIMINARIES

**Notations.** For any positive integer $d$, let $[d] = \{1, \ldots, d\}$. Vectors are denoted by lowercase bold letters (e.g., $\mathbf{a}$), matrices by uppercase bold letters (e.g., $\mathbf{A}$), and third-order tensors by underlined uppercase letters (e.g., $\underline{\mathbf{A}}$). Constants such as $c$, $c_1$, and $C$ may change from line to line. For a tensor of size $d_1 \times d_2 \times m$, we assume $d_1 \geq d_2$ for convenience.

For a matrix $\mathbf{A} \in \mathbb{R}^{d_1 \times d_2}$, let $\boldsymbol{\sigma}(\mathbf{A})$ be its singular values in descending order. The spectral and nuclear norms are $\|\mathbf{A}\|_{\text{spec}}$ and $\|\mathbf{A}\|_*$, defined as the largest and the sum of these values. For any tensor $\underline{\mathbf{A}}$, we set $\|\underline{\mathbf{A}}\|_p := \|\operatorname{vec}(\underline{\mathbf{A}})\|_p$ and $\|\underline{\mathbf{A}}\|_F := \|\operatorname{vec}(\underline{\mathbf{A}})\|_2$, where $\operatorname{vec}(\cdot)$ denotes vectorization (Kolda & Bader, 2009). The inner product is $\langle \underline{\mathbf{A}}, \underline{\mathbf{B}} \rangle := \operatorname{vec}(\underline{\mathbf{A}})^\top \operatorname{vec}(\underline{\mathbf{B}})$. For $\underline{\mathbf{A}} \in \mathbb{R}^{d_1 \times d_2 \times m}$, its $i$-th frontal slice is $\underline{\mathbf{A}}_{:,:,i}$, abbreviated as $\underline{\mathbf{A}}_i$ when clear from context.

**The t-SVD Framework.** The t-SVD framework relies on the t-product, a tensor generalization of matrix multiplication that operates under an invertible linear transform $M$ (Kernfeld et al., 2015). Suitable transforms can strengthen low-rank structure and capture correlations across slices (Zhang & Ng, 2021; Wang et al., 2021a). Following common practice, we use an orthogonal matrix $\mathbf{M} \in \mathbb{R}^{m \times m}$ for stability and efficiency (Lu, 2021; Wang et al., 2023a). For a tensor $\underline{\mathbf{T}} \in \mathbb{R}^{d_1 \times d_2 \times m}$, the $M$-transform and its inverse are defined by

$$M(\underline{\mathbf{T}}) := \underline{\mathbf{T}} \times_3 \mathbf{M}, \qquad M^{-1}(\underline{\mathbf{T}}) := \underline{\mathbf{T}} \times_3 \mathbf{M}^{-1}, \tag{1}$$

where $\times_3$ is the mode-3 tensor–matrix product (Kernfeld et al., 2015). These operators permit the basic notions of the t-SVD framework.

**Definition 2.1** (t-product (Kernfeld et al., 2015)). The t-product of two tensors $\underline{\mathbf{A}} \in \mathbb{R}^{d_1 \times d_2 \times m}$ and $\underline{\mathbf{B}} \in \mathbb{R}^{d_2 \times d_3 \times m}$ under the transform $M$ in (1) is denoted by $\underline{\mathbf{A}} *_M \underline{\mathbf{B}} = \underline{\mathbf{C}} \in \mathbb{R}^{d_1 \times d_3 \times m}$, where $M(\underline{\mathbf{C}}) = M(\underline{\mathbf{A}}) \odot M(\underline{\mathbf{B}})$ in the transformed domain, and $\odot$ denotes the frontal-slice-wise product.

**Definition 2.2** ($M$-block-diagonal matrix (Wang et al., 2023a)). For a tensor $\underline{\mathbf{T}} \in \mathbb{R}^{d_1 \times d_2 \times m}$, its $M$-block-diagonal matrix $\bar{\mathbf{T}} \in \mathbb{R}^{d_1 m \times d_2 m}$ is $\bar{\mathbf{T}} := \operatorname{bdiag}(M(\underline{\mathbf{T}})) = \operatorname{diag}(M(\underline{\mathbf{T}})_{:,:,1}, \ldots, M(\underline{\mathbf{T}})_{:,:,m})$, where $M(\underline{\mathbf{T}})$ is the mode-3 transform of $\underline{\mathbf{T}}$ and $\operatorname{bdiag}(\cdot)$ places the frontal slices on the diagonal.

We now formally introduce the t-SVD, as illustrated in Fig. 1-(A).

**Definition 2.3** (t-SVD and tensor tubal rank (Kernfeld et al., 2015)). The tensor Singular Value Decomposition (t-SVD) of a tensor $\underline{\mathbf{T}} \in \mathbb{R}^{d_1 \times d_2 \times m}$ under the invertible linear transform $M$ in (1) is:

$$\underline{\mathbf{T}} = \underline{\mathbf{U}} *_M \underline{\mathbf{S}} *_M \underline{\mathbf{V}}^\top, \tag{2}$$

where $\underline{\mathbf{U}} \in \mathbb{R}^{d_1 \times d_1 \times m}$ and $\underline{\mathbf{V}} \in \mathbb{R}^{d_2 \times d_2 \times m}$ are t-orthogonal tensors, and $\underline{\mathbf{S}} \in \mathbb{R}^{d_1 \times d_2 \times m}$ is an f-diagonal tensor. The tubal rank of $\underline{\mathbf{T}}$ is defined as the number of non-zero tubes in $\underline{\mathbf{S}}$ in the t-SVD, i.e., $r_{\text{tb}}(\underline{\mathbf{T}}) := \#\{i \mid \underline{\mathbf{S}}_{i,i,:} \neq \mathbf{0}, i \leq \min\{d_1, d_2\}\}$.

To further model the low-rank structure of tensors in the transformed domain, the tensor nuclear norm is proposed as a key regularizer in low-rank tensor learning:

**Definition 2.4** (Tensor nuclear norm (Lu et al., 2019b)). The tensor nuclear norm (TNN) of a tensor $\underline{\mathbf{T}} \in \mathbb{R}^{d_1 \times d_2 \times m}$ under the transform $M$ are defined as $\|\underline{\mathbf{T}}\|_* := \|\bar{\mathbf{T}}\|_* = \|\boldsymbol{\sigma}(\bar{\mathbf{T}})\|_1$.

By definition, TNN measures *element-wise sparsity of the transformed spectrum* $\boldsymbol{\sigma}(\bar{\mathbf{T}}) \in \mathbb{R}^{m \cdot \min\{d_1, d_2\}}$, thereby promoting low-rank structure in the transformed domain. It becomes a key tool for low-rank tensor recovery such as image/video denoising and inpainting (Lu et al., 2019a).

## 3 MODELING DUAL SPECTRAL SPARSITY IN THE t-SVD FRAMEWORK

This subsection answers **RQ1** by proposing a modeling tool for the dual spectral sparsity commonly observed in real tensor data, which is insufficient for TNN to fully leverage.

**Prevalence of Dual Spectral Sparsity.** Real tensor data exhibit a characteristic two-level spectral structure that appears consistently across modalities. The visual evidence in Fig. 1 shows that only a small subset of frequency bands carries substantial spectral energy for representative datasets, including *Salinas A*[1] , *Brain MRI* (Xu et al., 2015), and *Incisix* (Gandy et al., 2011). In these examples, more than eighty percent of the total energy concentrates in roughly fifteen to thirty percent of the frequency bands. The singular-value heatmap in Fig. 1(B) further illustrates two distinct features. Many frequency slices contribute negligibly, and the dominant slices display rapid spectral decay, which reflects clear low-rank organization within each active slice.

To assess whether this behavior extends beyond the illustrated cases, we provide a statistical analysis in Appendix D.10. The hypothesis tests consistently distinguish real tensors from matched random baselines and confirm that both inter-frequency sparsity and within-band low-rankness occur systematically across multispectral imaging and video datasets. Together, these results demonstrate that dual spectral sparsity is a stable and pervasive property of real tensor data rather than an artifact of specific examples.

**Limitations of TNN from a Group Sparsity View.** Definition 2.4 shows that TNN promotes low-rankness by imposing element-wise sparsity on the transformed singular values $\boldsymbol{\sigma}(\bar{\mathbf{T}})$. The limitation lies in its *uniform treatment of all frequencies*: every slice is penalized equally, regardless of their spectral importance. From a group-sparsity view, the spectrum $\boldsymbol{\sigma}(\bar{\mathbf{T}})$ decomposes into groups $\{\boldsymbol{\sigma}(M(\underline{\mathbf{T}})_{:,:,i})\}$, yet TNN penalizes all groups uniformly, ignoring their unequal contributions. This uniformity limits its capability to model tensors with dual spectral sparisity, motivating a framework that can distinguish between frequency selection and within-frequency decay.

**Hard Dual Spectral Sparsity.** We begin with a hard dual spectral sparsity model as follows.

**Definition 3.1** (Hard Dual Spectral Sparsity). A tensor $\underline{\mathbf{T}} \in \mathbb{R}^{d_1 \times d_2 \times m}$ is said to exhibit $(s, r)$-dual sparsity under a linear transform $M$ if it satisfies two constraints:

**I.** *Inter-frequency sparsity:* The number of active frequency components is limited to at most $s$. Specifically, only $s$ out of the $m$ frequency components can have non-zero singular value vectors: $\sum_{i=1}^{m} \mathbb{I}(\boldsymbol{\sigma}(M(\underline{\mathbf{T}})_{:,:,i}) \neq \mathbf{0}) \leq s$, where $\boldsymbol{\sigma}(M(\underline{\mathbf{T}})_{:,:,i})$ denotes the singular value vector of the $i$-th frontal slice in the transformed domain.

**II.** *Intra-frequency low-rankness:* Within each active frequency component, the number of non-zero singular values is constrained to at most $r$. This condition ensures a low-rank structure for each frequency slice ($\forall i \in [m]$): $\sum_{j=1}^{\min\{d_1, d_2\}} \mathbb{I}(\sigma_j(M(\underline{\mathbf{T}})_{:,:,i}) \neq 0) \leq r$, where $\sigma_j(M(\underline{\mathbf{T}})_{:,:,i})$ denotes the $j$-th singular value of the $i$-th frontal slice of $M(\underline{\mathbf{T}})$.

This definition imposes a strict dual-level structure by selecting a small frequency set and enforcing low-rankness within each selection. Although it may be overly rigid when spectral energy decays gradually, it offers a clean baseline that helps motivate and analyze more flexible soft formulations.

**Soft Dual Spectral Sparsity.** In practice, spectral energy typically decays smoothly, making the boundary between active and inactive frequencies both gradual and noise–dependent. To model this

---

[1]https://www.ehu.eus/ccwintco/index.php?title=Hyperspectral_Remote_Sensing_Scenes

softer structure, we introduce the $\ell_p$–Schatten-$q$ quasi-norm, which replaces hard selection and fixed ranks with approximate frequency sparsity and smooth within-slice spectral decay.

**Definition 3.2** (Tensor $\ell_p$–Schatten-$q$ quasi-norm). For exponents $(p,q) \in (0,1]^2$ and a tensor $\underline{\mathbf{T}} \in \mathbb{R}^{d_1 \times d_2 \times m}$, we define the tensor $\ell_p$-Schatten-$q$ quasi-norm (abbreviated as $\ell_p(S_q)$-norm) as:

$$\|\underline{\mathbf{T}}\|_{\ell_p(S_q)} := \left( \sum_{i=1}^{m} \left( \sum_{j=1}^{d_1 \wedge d_2} \sigma_j \big( M(\underline{\mathbf{T}})_{:,:,i} \big)^q \right)^{p/q} \right)^{1/p}. \tag{3}$$

In this quasi-norm, $p$ governs the inter-frequency sparsity by promoting a group-wise regularization across frequency components, effectively highlighting significant groups while suppressing others. Simultaneously, $q$ controls the intra-frequency low-rankness by encouraging sparsity in the singular values within each frequency slice, thereby modeling the intrinsic low-rank structure of the data. This soft dual spectral sparsity framework provides a unified yet versatile approach to address the hierarchical complexity of tensor data.

The $\ell_p$-Schatten-$q$ quasi-norm encompasses several existing regularization methods: it recovers TNN when $(p,q) = (1,1)$(Lu et al., 2019b), approximates the average rank as $(p,q) \to (1,0)$(Wang et al., 2021b), and reduces to the tensor Schatten-$q$ norm when $p = q$ (Kong et al., 2018), thereby offering greater modeling flexibility. *Despite generalizing these regularizers, it fundamentally differs by jointly enforcing global frequency sparsity and local spectral low-rankness.*

# 4 STATISTICAL ESTIMATION UNDER DUAL SPECTRAL SPARSITY

This section develops the theory of tensor estimation with dual spectral sparsity (**RQ2**).

**Challenges.** The $\ell_p$-Schatten-$q$ quasi-norm, combining inter-frequency sparsity with intra-frequency low-rankness, leads to *a globally coupled structure* that fundamentally differs from classical decoupled models like TNN. It results in a highly non-convex parameter space with nested sparsity patterns. Characterization of the estimation complexity demands novel extensions of covering numbers and metric entropy that jointly capture inter-frequency sparsity and intra-frequency low-rankness under both discrete and continuous constraints.

To uncover the statistical limits of learning under dual spectral sparsity, we analyze a simplified but representative model: the *Gaussian location model*, where the observed tensor is corrupted by Gaussian noise. This setting preserves the core structural properties, that is, inter-frequency sparsity and intra-frequency low-rankness, while avoiding complications unrelated to sparsity itself. We define structured parameter spaces that capture hard and soft variants of dual spectral sparsity, and establish sharp minimax lower and upper bounds under each. These results reveal how the joint effects of frequency selection and within-slice spectral decay determine the estimation limits.

## 4.1 GAUSSIAN LOCATION MODEL

Consider the Gaussian location model (GLM) (Li et al., 2024), where $n$ independent noisy realizations of the target tensor $\underline{\mathbf{L}}^* \in \mathbb{R}^{d_1 \times d_2 \times m}$ are observed as:

$$\underline{\mathbf{Y}}_i = \underline{\mathbf{L}}^* + \underline{\mathbf{E}}_i, \quad i \in [n], \tag{4}$$

where $\underline{\mathbf{Y}}_i$ is the observed tensor, $\underline{\mathbf{L}}^*$ represents the ground truth tensor of interest, and $\underline{\mathbf{E}}_i$ denotes the noise tensor with *i.i.d.* entries from $\mathcal{N}(0, \sigma^2)$. The parameter $\sigma$ characterizes the noise level. To simplify the analysis, we consider the sample mean $\underline{\bar{\mathbf{Y}}} = n^{-1} \sum_{i=1}^{n} \underline{\mathbf{Y}}_i = \underline{\mathbf{L}}^* + \underline{\bar{\mathbf{E}}}$, where $\underline{\bar{\mathbf{E}}} = n^{-1} \sum_{i=1}^{n} \underline{\mathbf{E}}_i$ is the aggregated noise tensor with entries independently distributed as $\mathcal{N}(0, \sigma^2/n)$. The goal is to estimate the ground truth $\underline{\mathbf{L}}^*$ based on the noisy observations $\{\underline{\mathbf{Y}}_i\}_{i=1}^{n}$.

**Remark 4.1.** We adopt GLM to isolate the core effects of dual spectral sparsity and the $\ell_p$-Schatten-$q$ regularization, avoiding additional complications from sampling operators in tensor regression (Zhang et al., 2020; Wang et al., 2021a; Qiu et al., 2024). This simplified setting enables cleaner analysis and yields insights that extend naturally to regression problems like tensor compressed sensing or tensor completion under standard conditions such as RIP (Zhang et al., 2020) or RSC (Wang et al., 2021a; Qiu et al., 2024; Negahban & Wainwright, 2012).

**Dual Spectral Sparsity Assumptions.** We consider three distinct sparsity models for $\underline{\mathbf{L}}^*$:

**A1.** *Hard dual spectral sparsity*: Let $\underline{\mathbf{L}}^*$ belong to the parameter space

$$\mathbf{T}_{0,0}(s,r) = \{\underline{\mathbf{L}} : \text{at most } s \text{ active frequency slices, each of rank at most } r\}. \tag{5}$$

This model enforces exact inter-frequency sparsity and intra-frequency low-rankness.

**A2.** *Hard frequency sparsity and soft rank constraint (hard–soft sparsity)*: Let $\underline{\mathbf{L}}^*$ lie in

$$\mathbf{T}_{0,q}(s,R) = \left\{\underline{\mathbf{L}} : |\{i : M(\underline{\mathbf{L}})_{:,:,i} \neq \mathbf{0}\}| \leq s, \; \|M(\underline{\mathbf{L}})_{:,:,i}\|_{S_q}^q \leq R, \; \forall i \in [m]\right\}. \tag{6}$$

It imposes hard inter-frequency sparsity and soft Schatten-$q$ constraints within each active slice.

**A3.** *Soft dual spectral sparsity*: Let $\underline{\mathbf{L}}^*$ belong to the parameter space

$$\mathbf{T}_{p,q}(R) = \left\{\underline{\mathbf{L}} : \|\underline{\mathbf{L}}\|_{\ell_p(S_q)}^p \leq R\right\}, \tag{7}$$

where the $\ell_p(S_q)$ quasi-norm provides a soft spectral structure combining frequency selection with within-slice spectral decay, and $R$ specifies the radius of the constraint set.

These parameter spaces represent increasing flexibility: the hard model imposes strict thresholds, the hard–soft model relaxes intra-slice structure, and the fully soft model allows gradual spectral decay. Our goal is to estimate $\underline{\mathbf{L}}^*$ and establish minimax bounds under each setting.

## 4.2 MINIMAX RISK OVER DUAL SPECTRAL SPARSITY

A central problem in tensor estimation is to understand the intrinsic difficulty of recovering a tensor that exhibits dual spectral sparsity from noisy observations. To quantify this difficulty, we derive matching minimax lower and upper bounds for the estimation risk

$$\mathfrak{M}(\mathbf{T}) = \inf_{\hat{\underline{\mathbf{L}}}} \sup_{\underline{\mathbf{L}}^* \in \mathbf{T}} \mathbb{E}\left[\|\hat{\underline{\mathbf{L}}} - \underline{\mathbf{L}}^*\|_{\mathrm{F}}^2\right], \tag{8}$$

where $\mathbf{T}$ denotes the parameter space. Following Lu (2021), we consider $d_1 = d_2 = d$ for simplicity.

**Theorem 4.2.** *The minimax risk can be bounded under certain conditions[2]:*

**I.** *Hard constraints on both frequency sparsity and per-slice low-rankness:*

$$\mathfrak{M}(\mathbf{T}_{0,0}(s,r)) \asymp \frac{\sigma^2}{n}\left(s \log \frac{em}{s} + srd\right).$$

**II.** *Hard frequency sparsity with soft intra-slice Schatten-q constraints:*

$$\mathfrak{M}(\mathbf{T}_{0,q}(s,R)) \asymp \frac{\sigma^2}{n} s \log \frac{em}{s} + sR\left(\frac{\sigma^2}{n}d\right)^{1-\frac{q}{2}}.$$

**III.** *Soft $\ell_p(S_q)$ constraints over both frequency sparsity and intra-slice low-rankness:*

$$\mathfrak{M}(\mathbf{T}_{p,q}(R)) \asymp \begin{cases} R\left(\frac{n}{d\sigma^2}\right)^{\frac{p-2}{2}} + R\left(\frac{n}{\sigma^2 \log m}\right)^{\frac{p-2}{2}}, & p > q, \\ R^{\frac{q}{p}}\left(\frac{n}{d\sigma^2}\right)^{\frac{q-2}{2}} + R\left(\frac{n}{\sigma^2 \log m}\right)^{\frac{p-2}{2}}, & p \leq q, \; m > d^2, \\ R^{\frac{q}{p}}\left(\frac{n}{d\sigma^2}\right)^{\frac{q-2}{2}}, & p \leq q, \; m \leq d^2. \end{cases}$$

Theorem 4.2 shows that estimation difficulty is governed jointly by sparsity across frequencies and low-rankness within each active frequency. In **I** (hard setting), selecting the $s$ informative frequencies costs $s \log(em/s)$ and estimating their rank–$r$ slices costs $srd$. In **II** (hard–soft setting), fixed rank is replaced by Schatten–$q$ decay, and the second term becomes $sR(n^{-1}d)^{1-q/2}$, reflecting smoother within-frequency spectra. In **III** (fully soft setting), both structures are relaxed and the rate is dictated by the balance between the $\ell_p$ and $S_q$ components: $\ell_p$ dominates when $p > q$, whereas for $p \leq q$ the $S_q$ term governs and becomes independent of $m$ when $m \leq d^2$. This reflects how frequency decay and within-frequency spectral decay co-determine the model complexity.

---

[2]The conditions in each setting are provided in the Appendix C.

# 5 OPTIMIZATION FOR DUAL SPECTRAL SPARSE TENSOR ESTIMATION

Efficiently solving tensor estimation problems with dual spectral sparsity (**RQ3**) is key to leveraging the proposed $\ell_p$-Schatten-$q$ quasi-norm in practice. However, this task presents substantial challenges due to the non-convexity and coupled structure of this regularization.

**Challenges.** Even in the vector setting, optimizing dual-level sparse structures is notoriously difficult due to the combination of *non-convexity* and *structural coupling* (Hu et al., 2017; Lin et al., 2024). In our tensor case, these challenges are further compounded by the need to simultaneously enforce inter-frequency sparsity and intra-frequency low-rankness. Most existing tensor optimization methods either treat frequency components independently or impose low-rank constraints without spectral sparsity considerations, making them ill-suited for the proposed dual-spectral regularization. The $\ell_p$-Schatten-$q$ quasi-norm is non-convex whenever $p, q \in (0, 1)$, ruling out standard convex optimization techniques and necessitating a structure-aware, non-convex optimization strategy.

To address these difficulties, our approach is naturally motivated by the structural properties of the problem. We adopt a *proximal update scheme* that takes advantage of the separability of the transform-domain representation $M(\underline{\mathbf{L}})$, allowing frequency-wise updates, along with an iterative reweighting strategy that facilitates optimization in the presence of non-convex regularization.

**Proximal Operator Formulation.** To handle the non-convex $\ell_p$-Schatten-$q$ regularization, we adopt a proximal update scheme that enforces dual spectral sparsity while remaining computationally efficient. Specifically, at iteration $t$, the update is given by solving:

$$\underline{\mathbf{L}}^{t+1} \in \arg\min_{\underline{\mathbf{L}}} \frac{1}{2}\|\underline{\mathbf{L}} - \underline{\mathbf{Z}}\|_{\mathrm{F}}^2 + \lambda \sum_{k=1}^{m} \|M(\underline{\mathbf{L}})_{:,:,k}\|_{S_q}^p, \tag{9}$$

where $\underline{\mathbf{Z}}$ denotes the intermediate variable aggregating previous updates and gradient information. Since the transform $M(\cdot)$ allows slice-wise decomposition (Kilmer et al., 2013), Problem (9) reduces to $m$ subproblems over frequency components $k \in [m]$:

$$\min_{\mathbf{A}_k} \frac{1}{2} \|\mathbf{A}_k - M(\underline{\mathbf{Z}})_{:,:,k}\|_{\mathrm{F}}^2 + \lambda \|\mathbf{A}_k\|_{S_q}^p, \tag{10}$$

where $\mathbf{A}_k := M(\underline{\mathbf{L}})_{:,:,k}$ denotes the $k$-th frontal slice of the transformed tensor $M(\underline{\mathbf{L}})$. Problem (10) is difficult due to the non-convexity and lack of smoothness of the Schatten-$q$ quasi-norm, which admits no closed-form or standard proximal solution in general.

To tackle Problem (10), we adopt a reweighted $\ell_{1/2}$-proxy for $\|\mathbf{A}_k\|_{S_q}^p$ based on singular values:

$$\sum_{i=1}^{d} w_{i,k} \cdot \sigma_i(\mathbf{A}_k)^{1/2}, \tag{11}$$

with weights defined as $w_{i,k} = \left( \sum_{j=1}^{d} \varsigma_{j,k}^q + \epsilon \right)^{p/q-1} \cdot \left( \varsigma_{i,k}^{1/2} + \epsilon \right)^{2q-1}$, where $\epsilon$ is a small regularization constant and $\varsigma_{j,k} := \sigma_j(M(\underline{\mathbf{L}}^t)_{:,:,k})$ are the singular values from the previous iterate. The update for each singular value then becomes a soft-thresholding step:

$$\sigma_i^{(t+1)}(M(\underline{\mathbf{L}})_{:,:,k}) = \mathcal{S}_{\lambda w_{i,k}}^{\ell_{1/2}} \left( \sigma_i(M(\underline{\mathbf{Z}})_{:,:,k}) \right), \tag{12}$$

where $\mathcal{S}^{\ell_{1/2}}$ is the proximal operator for the $\ell_{1/2}$-norm (see Eq. (45) in Appendix D.2).

After singular value shrinkage, we reconstruct each slice $M(\underline{\mathbf{L}}^{t+1})_{:,:,k} = \mathbf{U}_k \cdot \mathrm{diag}(\boldsymbol{\sigma}^{(t+1)}) \cdot \mathbf{V}_k^\top$, where $\mathbf{U}_k$ and $\mathbf{V}_k$ are from the SVD of $M(\underline{\mathbf{Z}})_{:,:,k}$. Finally, applying the inverse transform yields the updated tensor $\underline{\mathbf{L}}^{t+1}$ in the original domain.

# 6 EXPERIMENTS

The goal of our experiments is to evaluate the effectiveness and scope of the proposed $\ell_p$-Schatten-$q$ regularizer across diverse tensor learning tasks. We consider three settings: (1) **noisy tensor completion** with random missing entries and Gaussian noise; (2) **Poisson tensor completion** with signal-dependent photon noise and structured sampling; and (3) **image clustering** for low-rank representation learning. Further details and ablations are provided in Appendix D.

Table 1: Noisy tensor completion results on benchmark datasets. The proposed $\ell_p(S_q)$ is evaluated under four parameter settings of $(p, q)$, denoted as **S0–S3**: **S0** $(0.8961, 0.8966)$, **S1** $(0.60, 0.61)$, **S2** $(0.70, 0.71)$, and **S3** $(0.80, 0.81)$. TNN-DFT and TNN-DCT appear as TNN-F and TNN-C for compactness. (Best in **bold**; second-best underlined.)

| Dataset | SR | Metric | NN | SNN | TNN-F | TNN-C | k-Sup | $\ell_{1-2}$ | $S_{1/2}$ | S0 | S1 | S2 | S3 |
|---|---|---|---|---|---|---|---|---|---|---|---|---|---|
| SalinasA | 5% | PSNR | 15.21 | 20.79 | 22.55 | 26.52 | 22.58 | 22.21 | 22.45 | 28.43 | 28.50 | **28.72** | 28.55 |
| | | SSIM | 0.2594 | **0.7547** | 0.5667 | 0.7384 | 0.5689 | 0.5524 | 0.4474 | 0.7374 | 0.7399 | 0.7505 | 0.7401 |
| | 10% | PSNR | 20.62 | 25.56 | 25.72 | 29.61 | 25.89 | 26.14 | 25.86 | **31.81** | 31.58 | 31.57 | 31.62 |
| | | SSIM | 0.4775 | 0.8284 | 0.7027 | 0.8403 | 0.7231 | 0.7197 | 0.6058 | **0.8484** | 0.8423 | 0.8413 | 0.8428 |
| | 15% | PSNR | 23.09 | 27.99 | 28.06 | 31.32 | 28.09 | 28.13 | 26.98 | 33.23 | 33.16 | 33.17 | **33.28** |
| | | SSIM | 0.5643 | 0.8622 | 0.7804 | 0.8798 | 0.7810 | 0.7795 | 0.6505 | 0.8830 | 0.8832 | 0.8846 | **0.8856** |
| IndianPines | 5% | PSNR | 20.44 | 22.01 | 25.68 | 26.26 | 25.70 | 25.73 | 24.68 | **27.05** | 26.68 | 26.86 | 26.87 |
| | | SSIM | 0.3895 | 0.6359 | 0.6293 | 0.6727 | 0.6289 | 0.6316 | 0.5361 | **0.6740** | 0.6537 | 0.6658 | 0.6636 |
| | 10% | PSNR | 22.23 | 24.94 | 27.45 | 28.40 | 27.48 | 27.52 | 25.72 | 28.92 | 28.96 | **29.02** | 28.97 |
| | | SSIM | 0.4836 | 0.7171 | 0.7226 | **0.7744** | 0.7219 | 0.7249 | 0.5991 | 0.7617 | 0.7657 | 0.7684 | 0.7653 |
| | 15% | PSNR | 23.52 | 26.61 | 28.54 | 29.52 | 28.53 | 28.63 | 26.24 | 29.89 | 29.93 | **30.01** | 29.97 |
| | | SSIM | 0.5438 | 0.7668 | 0.7713 | **0.8177** | 0.7709 | 0.7741 | 0.6258 | 0.7997 | 0.8026 | 0.8090 | 0.8039 |
| Cloth | 5% | PSNR | 20.10 | 20.95 | 25.00 | 26.09 | 25.08 | 25.09 | 24.96 | 26.99 | 26.79 | 27.02 | **27.24** |
| | | SSIM | 0.3762 | 0.5096 | 0.6773 | 0.7283 | 0.6792 | 0.6793 | 0.6305 | 0.7422 | 0.7227 | 0.7342 | **0.7447** |
| | 10% | PSNR | 21.14 | 22.72 | 28.00 | 29.24 | 28.12 | 28.14 | 27.98 | 30.63 | 30.17 | 30.36 | 30.52 |
| | | SSIM | 0.4341 | 0.5983 | 0.8132 | 0.8540 | 0.8143 | 0.8163 | 0.7668 | 0.8658 | 0.8473 | 0.8546 | 0.8576 |
| | 15% | PSNR | 22.05 | 24.18 | 30.03 | 31.36 | 30.08 | 30.11 | 29.50 | 32.71 | 32.31 | 32.51 | 32.61 |
| | | SSIM | 0.4889 | 0.6783 | 0.8722 | 0.9054 | 0.8727 | 0.8733 | 0.8153 | **0.9090** | 0.8977 | 0.9025 | 0.9047 |
| Hair | 5% | PSNR | 25.33 | 30.09 | 33.16 | 35.31 | 33.19 | 33.27 | 33.43 | **36.95** | 36.47 | 36.81 | 36.84 |
| | | SSIM | 0.7147 | 0.8631 | 0.8917 | **0.9248** | 0.8921 | 0.8919 | 0.8240 | 0.9196 | 0.9151 | 0.9197 | 0.9182 |
| | 10% | PSNR | 29.52 | 33.35 | 36.22 | 38.18 | 36.17 | 36.30 | 35.69 | **39.91** | 39.70 | 39.86 | 39.83 |
| | | SSIM | 0.8008 | 0.9122 | 0.9292 | **0.9535** | 0.9286 | 0.9296 | 0.8640 | 0.9517 | 0.9510 | 0.9526 | 0.9515 |
| | 15% | PSNR | 31.12 | 35.24 | 38.00 | 39.88 | 37.91 | 38.07 | 36.46 | **41.52** | 41.31 | 41.47 | 41.51 |
| | | SSIM | 0.8364 | 0.9336 | 0.9449 | 0.9650 | 0.9442 | 0.9448 | 0.8735 | 0.9641 | 0.9640 | **0.9653** | 0.9645 |
| JellyBeans | 5% | PSNR | 16.33 | 18.21 | 25.43 | 26.47 | 25.38 | 25.62 | 25.39 | 27.91 | 27.91 | 28.08 | **28.16** |
| | | SSIM | 0.2397 | 0.4942 | 0.6726 | **0.7223** | 0.6714 | 0.6733 | 0.5504 | 0.7115 | 0.6916 | 0.7001 | 0.7083 |
| | 10% | PSNR | 18.12 | 22.11 | 28.50 | 30.14 | 28.47 | 28.67 | 28.41 | **31.95** | 31.52 | 31.78 | 31.90 |
| | | SSIM | 0.3169 | 0.6629 | 0.7900 | **0.8518** | 0.7902 | 0.7932 | 0.6905 | 0.8486 | 0.8321 | 0.8411 | 0.8461 |
| | 15% | PSNR | 19.92 | 24.67 | 30.51 | 32.33 | 30.52 | 30.61 | 29.50 | **33.97** | 33.39 | 33.81 | 33.89 |
| | | SSIM | 0.4053 | 0.7592 | 0.8489 | **0.9030** | 0.8504 | 0.8499 | 0.7516 | 0.8980 | 0.8818 | 0.8932 | 0.8952 |
| *Thermal* | 5% | PSNR | 13.19 | 15.83 | 28.06 | 27.99 | 28.01 | 28.19 | 28.11 | 30.06 | 30.21 | 30.09 | **30.49** |
| | | SSIM | 0.1848 | 0.4759 | 0.8584 | 0.8707 | 0.8579 | 0.8603 | 0.7928 | 0.8759 | 0.8701 | 0.8752 | **0.8763** |
| | 10% | PSNR | 14.67 | 19.75 | 31.30 | 31.62 | 31.28 | 31.60 | 30.51 | 33.67 | **33.96** | 33.76 | 33.85 |
| | | SSIM | 0.2509 | 0.6594 | 0.9151 | 0.9326 | 0.9147 | 0.9168 | 0.8358 | 0.9272 | 0.9323 | **0.9331** | 0.9291 |
| | 15% | PSNR | 16.27 | 22.52 | 33.02 | 33.51 | 33.05 | 33.11 | 30.99 | 35.09 | 35.42 | **35.50** | 35.36 |
| | | SSIM | 0.3273 | 0.7621 | 0.9315 | **0.9509** | 0.9321 | 0.9318 | 0.8373 | 0.9404 | 0.9473 | 0.9507 | 0.9456 |

**Noisy Tensor Completion.** We first consider the noisy tensor completion which involves reconstructing a tensor from noisy incomplete observations. Given a clean tensor $\underline{\mathbf{L}}$ of size $d_1 \times d_2 \times d_3$, we introduce *i.i.d.* Gaussian noise with standard deviation $\sigma = c\sigma_0$, where $c = 0.05$ and $\sigma_0 = \|\underline{\mathbf{L}}\|_F / \sqrt{d_1 d_2 d_3}$. A uniform sampling strategy is applied with sampling ratios (SR) $p \in \{5\%, 10\%, 15\%\}$. Each setting is tested over 10 trials, and the averaged PSNR (dB) and SSIM values are reported. To benchmark our method, we compare the proposed $\ell_p(S_q)$-quasi-norm against several low-rank regularizers, including matrix nuclear norm (NN) (Candès & Tao, 2010), Tucker-based tensor nuclear norm (SNN) (Liu et al., 2013), TNN-DFT (Zhang & Aeron, 2017), TNN-DCT (Lu et al., 2019b), tensor $k$-Support norm ($k$-Supp) ($k = 2$) (Wang et al., 2021a), tensor $\ell_{1-2}$-norm ($\ell_{1-2}$) (Tan et al., 2023), tensor Schatten-$p$-norm ($p = 1/2$) (Kong et al., 2018). For the proposed method, we adopt the DFT as the spectral operator $M(\cdot)$, and explore four configurations of the sparsity parameters $(p, q)$, denoted as: **S0**: $(0.8961, 0.8966)$, **S1**: $(0.60, 0.61)$, **S2**: $(0.70, 0.71)$, and **S3**: $(0.80, 0.81)$. The default setting **S0** was determined via a grid search on the *Indian Pines* dataset at 5% sampling rate. The remaining configurations serve to evaluate robustness under different regularization strengths, and no further tuning was performed on test datasets. A comprehensive sensitivity analysis of $(p, q)$ is provided in Appendix D.15.

We conduct experiments on a variety of remote sensing data: hyperspectral (*Indian Pines*, *Salinas A*), multispectral (Columbia MSI: *Cloth*, *Hair*, *Jelly Beans*), and thermal imagery (*OSU Thermal*). Table 1 reports the PSNR and SSIM values across methods and sampling rates. The proposed $\ell_p(S_q)$ regularizer consistently achieves the highest PSNRs, and ranks among the top in SSIM,

demonstrating strong noise robustness and fidelity in both spectral and spatial domains. Further experimental comparisons are provided in Appendix D.

**Poisson Tensor Completion.** To further assess the modeling capacity of the $\ell_p(S_q)$ , we extend it to the Bi-Module Tensor Regularization (BTR) framework (Wang et al., 2025a) and apply the resulting *Bi-$\ell_p(S_q)$ model* to the challenging task of Poisson tensor completion (Zhang & Ng, 2021). Following the observation model and benchmark protocol of Wang et al. (2025a), we apply non-uniform sampling masks that remove structured rows, columns, and tubes, and compare our Bi-$\ell_p(S_q)$ model against Poisson completion

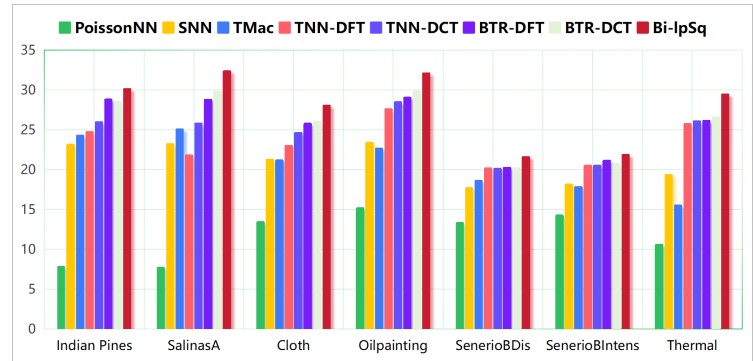

Figure 2: PSNR under 20% non-uniform sampling for Poisson tensor completion.

baselines including PoissonNN (Cao & Xie, 2015), SNN (Liu et al., 2013), TMac (Xu et al., 2015), TNN-DFT/DCT (Zhang & Ng, 2021), and BTR-DFT/DCT (Wang et al., 2025a). Figure 2 summarizes the PSNR values across seven datasets at a non-uniform sampling ratio of 20%. More results are shown in Table 19. Across the datasets, Bi-$\ell_p(S_q)$ achieves the highest reconstruction quality, demonstrating its robustness to signal-dependent noise and irregular sampling patterns.

**Image Clustering.** We further evaluate the Bi-$\ell_p(S_q)$ model on image clustering. As shown in Figure 3, the method consistently improves ACC, NMI, and PUR on four benchmarks (*FRDUE, FRDUE-100, PIE-10*, and *USPS1000*). These gains arise from the same dual spectral principle: the model emphasizes the most informative frequency components while enforcing low-rank structure within each slice, which leads to cleaner and more discriminative representations. The numerical results in Table 20 confirm that this spectral geometry stabilizes the extracted features and benefits clustering.

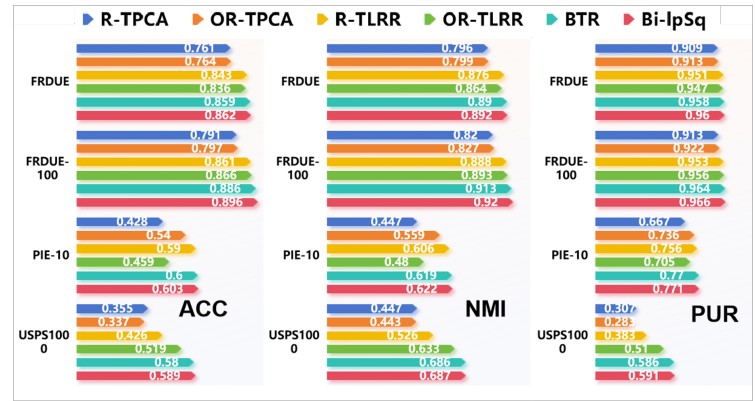

Figure 3: Clustering accuracy (ACC), normalized mutual information (NMI), and purity (PUR) on four benchmark datasets.

## 7 CONCLUSION

We identify a coupled spectral structure within the t-SVD framework, where inter-frequency sparsity and intra-frequency low-rankness appear simultaneously. The proposed $\ell_p$–Schatten-$q$ quasi-norm models these two forms of structure in a unified way and extends classical tensor regularizers such as TNN and Schatten-$q$ norms. We establish minimax guarantees that characterize the fundamental estimation limits under this dual spectral model and develop a proximal reweighted solver to handle the resulting nonconvex optimization. Empirically, the method achieves consistent gains across noisy tensor completion, Poisson tensor completion, and clustering, indicating that dual spectral regularization provides a useful inductive bias for structured tensor data.

## ETHICS STATEMENT

This work is primarily theoretical and relies solely on computational methods and publicly available datasets, involving no human subjects or private data. It adheres to the ICLR Code of Ethics, with no conflicts of interest. While potential dual-use concerns are acknowledged, we emphasize responsible use and compliance with research integrity.

## REPRODUCIBILITY STATEMENT

We provide code and illustrative data in the supplementary material to demonstrate the implementation details of our method and support reproduction of the main results.

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

# Appendix for
# *Refining Dual Spectral Sparsity in Transformed*
# *Tensor Singular Values*

This appendix provides a comprehensive supplement to the main paper, elaborating on theoretical foundations, algorithmic components, and additional empirical validations. The materials are organized to follow the logical development of our contributions and are structured as follows:

- **Comparison with Prior Work and Unique Technical Challenges (Appendix A).** We clarify the conceptual motivation behind our formulation and explain why the proposed framework is not a direct extension of either tensor nuclear norm (TNN) models or classical $\ell_u(\ell_q)$ norms. We also highlight the new technical challenges introduced by dual spectral coupling.

- **Additional Notations and Preliminaries (Appendix B).** This section formalizes key notations under the t-SVD framework and presents auxiliary lemmas that support our entropy calculations and complexity analysis.

- **Theoretical Results for Understanding Dual-Level Sparsity (Appendix C).** We provide detailed proofs of the main theorems concerning minimax lower and upper bounds, along with the structural properties of the $\ell_p(S_q)$ space. We also include a numerical illustration of Theorem 4.2 to empirically validate key trends.

- **Experimental Details and Additive Results (Appendix D).** This section describes the experimental setup, the ADMM-based solver, and the parameter selection strategy. It also includes ablations on surrogate choices, the role of dual spectral sparsity, different transforms (fixed and adaptive), robustness to sampling ratios and noise models, and additional noiseless completion benchmarks. In addition, we provide a systematic statistical validation of the dual spectral structure (Appendix D.10), extended experiments on diverse 3D and 4D tensors including YUV videos and seismic volumes (Appendix D.11-D.12), a Poisson tensor completion benchmark (Appendix D.13), clustering experiments (Appendix D.14), and a comprehensive sensitivity analysis of $(p, q)$ (Appendix D.15).

- **Discussion of Limitations and Future Directions (Appendix E).** We reflect on the scope of this work and articulate several potential extensions, including theoretical challenges and promising algorithmic refinements.

## USE OF LARGE LANGUAGE MODELS (LLMS)

We used LLMs solely as an assistive tool for language refinement and code polishing. Specifically, LLMs were employed to improve the clarity and fluency of the writing, as well as to standardize the presentation of pseudocode and LaTeX formatting. LLMs did not contribute to research ideation, methodological design, theoretical analysis, or experimental results.

APPENDIX CONTENTS

## A  COMPARISON WITH PRIOR WORK AND UNIQUE TECHNICAL CHALLENGES

### A.1  GENERAL CONTEXT AND RELATED WORK

Our work intersects with two major research areas: double sparse structures and tensor recovery methods. We discuss each in turn and highlight how our work advances these fields.

**Double Sparse Structures.**  Research on double sparse structures has demonstrated their effectiveness in capturing hierarchical sparsity patterns across various domains. In genomics, these structures model pathway-level and SNP-level sparsity in genome-wide association studies (Silver et al., 2013), while similar hierarchical patterns appear in classification tasks (Rao et al., 2015; Huo et al., 2020) and network analysis (Tugnait, 2021). Methodologically, bi-level selection approaches (Breheny & Huang, 2009) have dominated this field, evolving from the fundamental group bridge method (Huang et al., 2009) to more sophisticated techniques like sparse group lasso (Simon et al., 2013), which unified individual (Tibshirani, 1996) and group-level (Yuan & Lin, 2006) sparsity. Recent theoretical work by (Tony Cai et al., 2022) has established minimax bounds for double sparse regression, building on earlier investigations of general sparsity structures (Raskutti et al., 2011). Li et al. (2024) developed fundamental theoretical bounds for high-dimensional double sparse structures by establishing novel metric entropy bounds over $\ell_u(\ell_q)$-balls using Gilbert-Varshamov techniques, providing insights into the simultaneous estimation of group-wise and element-wise sparsity. However, these approaches, while successful for vector and matrix data, cannot address the fundamental challenge in transformed-domain tensor analysis: how to simultaneously model sparsity across different frequency components and low-rank structures within each frequency component—a gap our work bridges through the $\ell_p$-Schatten-$q$ framework.

**Tensor Recovery Methods.**  Tensor recovery research has evolved along multiple methodological paths, each addressing different aspects of multi-dimensional data analysis. Traditional approaches utilize various decomposition frameworks: CP decomposition with techniques ranging from sum-of-squares (Barak & Moitra, 2016) to gradient descent (Cai et al., 2019), Tucker decomposition employing nuclear norm minimization (Gandy et al., 2011) and manifold optimization (Xia & Yuan, 2019), and tensor train/ring decompositions using Riemannian methods (Cai et al., 2022). Recently, low-tubal-rank recovery has gained attention, with methods spanning both convex approaches like tensor nuclear norm minimization (Lu et al., 2019a) and non-convex alternatives such as Schatten-$p$ norm regularization (Kong et al., 2018). However, existing tensor methods, particularly those based on tensor nuclear norm (TNN), apply uniform regularization across all frequencies in the transformed domain. While effective, this single-level sparsity treatment can be further enhanced to fully capture the natural hierarchical patterns in real-world tensor data - for example, in hyperspectral images where some frequency bands carry more information than others while each band itself exhibits low-rank structure. Our $\ell_p$-Schatten-$q$ framework extends TNN's uniform regularization by introducing separate parameters to control sparsity across frequencies ($p$) and low-rank structure within each frequency ($q$).

### A.2  WHY OUR FRAMEWORK IS NOT A DIRECT EXTENSION OF TNN OR $\ell_u(\ell_q)$ NORMS

While our formulation includes the Tensor Nuclear Norm (TNN) (Lu et al., 2019b; Zhang et al., 2014) as a special case when $(p, q) = (1, 1)$ and shares some similarities with $\ell_u(\ell_q)$ norms (Li et al., 2024) used in vector models, it is ***structurally different*** from both. In particular, our framework captures a new interaction: sparsity across frequency slices and low-rankness within each slice are jointly modeled and cannot be separated. This type of coupling does not appear in previous models and leads to new technical challenges in both theory and algorithm design. We explain the key differences below.

1. **Comparison with TNN.** The TNN penalizes the sum of all singular values across transformed slices uniformly, assuming that each frontal slice contributes equally and independently. This leads to a separable regularization structure, where each slice is processed in isolation.

   In contrast, our proposed $\ell_p$–Schatten-$q$ quasi-norm:

   - imposes sparsity across frequency slices via the outer $\ell_p$ term;
   - induces per-slice low-rankness through the inner Schatten-$q$ quasi-norm;

- couples these two effects non-separably, such that frequency selection and rank penalization jointly influence the model.

This interdependence alters both the structure of the parameter space and the behavior of the regularizer, breaking the assumptions underpinning existing theoretical and algorithmic analyses of TNN.

2. **Comparison with $\ell_u(\ell_q)$ norms.** The $\ell_u(\ell_q)$ class has been studied extensively for vector-valued and matrix-valued double sparse models, where group structures are known and fixed (Li et al., 2024). Our setting differs fundamentally in both domain and geometry:

   - The group structure (i.e., frequency slices) is not fixed a priori, but emerges from a linear transform applied to the third mode;
   - Each group is a matrix with structured spectral decay, rather than a vector block;
   - The regularization interacts with both the spectral geometry and the transform, which invalidates direct use of existing entropy results.

   These differences imply that established results for $\ell_u(\ell_q)$-norm regularization do not extend to our setting, and necessitate new techniques tailored to transform-domain tensor analysis.

This distinction in modeling leads to substantial new technical challenges, which we outline next.

## A.3    NEW TECHNICAL CHALLENGES

The coupled spectral structure in our framework introduces analytical and algorithmic challenges that go beyond prior work. These arise across three key aspects:

1. **Coupled Parameter Geometry and Entropy Analysis.** The induced constraint set is no longer a product of independent low-dimensional balls, but a *hierarchically coupled* space shaped by both inter-group and intra-group constraints. To analyze its minimax complexity:

   - We construct nonstandard packing sets that jointly encode group selection and low-rank structure;
   - We extend Gilbert–Varshamov-based arguments to spectral-domain tensor geometry (see Lemmas C.9–C.15);
   - Our bounds reveal multiple scaling regimes depending on $(p, q)$ and tensor size, which are not captured by standard sparsity or low-rank models.

2. **Coupled Structure Complicates Optimization Design.** Unlike models such as TNN, where the regularizer decouples across frequency slices and admits closed-form proximal mappings, our $\ell_p(S_q)$ quasi-norm introduces two intertwined sources of difficulty:

   - First, the nesting of $\ell_p$ and Schatten-$q$ induces nonconvexity and nonseparability, for which no closed-form proximal operator is available—even when one of the parameters is fixed;
   - Second, the global $\ell_p$ term couples the optimization across slices, meaning that the regularization strength on one slice implicitly depends on the spectrum of others, breaking slice-wise independence.

To address these challenges, we introduce a reweighted surrogate based on the $\ell_{1/2}$-norm, enabling efficient frequency-wise updates that approximate the original regularization while preserving its structural intent. These updates are embedded into an ADMM framework that maintains computational tractability and spectral coherence across slices.

These challenges preclude the direct reuse of existing matrix or tensor tools, and motivate the new statistical and optimization techniques developed in this work.

# B  ADDITIONAL NOTATIONS, PRELIMINARIES AND LEMMAS

**Additional Notations.** We use several asymptotic notations to describe the relationships between functions. For the sake of clarity, we provide their definitions here:

- The notation $f(n) \lesssim g(n)$ means that there exists a positive constant $c$ and a positive integer $n_0$ such that for all $n \geq n_0$, we have $f(n) \leq c \cdot g(n)$. This is equivalent to saying $f(n) = O(g(n))$.
- Similarly, $f(n) \gtrsim g(n)$ means that there exists a positive constant $c$ and a positive integer $n_0$ such that for all $n \geq n_0$, we have $f(n) \geq c \cdot g(n)$. This is equivalent to saying $g(n) = O(f(n))$.
- We write $f(n) \asymp g(n)$ if both $f(n) \lesssim g(n)$ and $f(n) \gtrsim g(n)$ hold. This means that $f(n)$ and $g(n)$ are of the same order.
- The notation $f(n) = o(g(n))$ means that for every positive constant $\epsilon$, there exists a positive integer $n_0$ such that for all $n \geq n_0$, we have $|f(n)| \leq \epsilon \cdot |g(n)|$.

These notations allow us to express the asymptotic behavior of functions concisely, which is particularly useful in our analysis of algorithmic complexity and error bounds.

## B.1  PRELIMINARIES OF t-SINGULAR VALUE DECOMPOSITION

Due to space limitations, some concepts related to t-SVD were omitted in the main text. We provide additional notions here.

**Definition B.1** (Frontal-slice-wise product (Lu et al., 2019a))**.** The frontal-slice-wise product of any two tensors $\underline{\mathbf{A}} \in \mathbb{R}^{d_1 \times d_2 \times m}$ and $\underline{\mathbf{B}} \in \mathbb{R}^{d_1 \times d_2 \times m}$, denoted by $\underline{\mathbf{A}} \odot \underline{\mathbf{B}}$, is defined as a tensor $\underline{\mathbf{T}}$ such that

$$\underline{\mathbf{T}}_{:,:,i} = \underline{\mathbf{A}}_{:,:,i} \cdot \underline{\mathbf{B}}_{:,:,i}, \ i \in [K]$$

where $\cdot$ denotes the standard matrix multiplication. The frontal-slice-wise product performs matrix multiplication on each frontal slice of the tensors, resulting in a new tensor.

**Definition B.2** ($M$-block-diagonal matrix)**.** The $M$-block-diagonal matrix of any tensor $\underline{\mathbf{T}} \in \mathbb{R}^{d_1 \times d_2 \times m}$, denoted by $\bar{\mathbf{T}}$, is the block diagonal matrix whose diagonal blocks are the frontal slices of $M(\underline{\mathbf{T}})$:

$$\bar{\mathbf{T}} := \texttt{bdiag}(M(\underline{\mathbf{T}})) := \begin{bmatrix} M(\underline{\mathbf{T}})_{:,:,1} & & & \\ & M(\underline{\mathbf{T}})_{:,:,2} & & \\ & & \ddots & \\ & & & M(\underline{\mathbf{T}})^{(K)}) \end{bmatrix} \in \mathbb{R}^{d_1 m \times d_2 m}.$$

This concept arranges the slices of a tensor in the frequency domain into a block diagonal matrix, facilitating the theoretical analysis of t-SVD.

We further provide some definitions and properties related to t-SVD:

**Definition B.3.** (Kernfeld et al., 2015) The t-transpose of a tensor $\underline{\mathbf{T}} \in \mathbb{R}^{d_1 \times d_2 \times m}$ under the $M$ transform (as shown in (1)), denoted by $\underline{\mathbf{T}}^\top$, satisfies

$$M(\underline{\mathbf{T}}^\top)_{:,:,i} = (M(\underline{\mathbf{T}})_{:,:,i})^\top, \ i \in [K].$$

In other words, the t-transpose performs a transpose on each slice in the frequency domain and then transforms back to the time domain. This operation is one of the foundations of t-SVD theory.

**Definition B.4.** (Kernfeld et al., 2015) The t-identity tensor $\underline{\mathbf{I}} \in \mathbb{R}^{d \times d \times m}$ under the $M$ transform satisfies that each frontal slice of $M(\underline{\mathbf{I}})$ is an $m \times m$ identity matrix, i.e.,

$$M(\underline{\mathbf{I}})_{:,:,i} = \mathbf{I}, \ i \in [K].$$

It is easy to verify that $\underline{\mathbf{T}} *_M \underline{\mathbf{I}} = \underline{\mathbf{T}}$ and $\underline{\mathbf{I}} *_M \underline{\mathbf{T}} = \underline{\mathbf{T}}$ hold for appropriate dimensions. The t-identity tensor plays a role similar to the identity matrix in t-SVD.

**Definition B.5.** (Kernfeld et al., 2015) A tensor $\underline{\mathbf{Q}} \in \mathbb{R}^{d \times d \times m}$ is called t-orthogonal under the $M$ transform if it satisfies

$$\underline{\mathbf{Q}}^\top *_M \underline{\mathbf{Q}} = \underline{\mathbf{Q}} *_M \underline{\mathbf{Q}}^\top = \underline{\mathbf{I}}.$$

T-orthogonality is an important property of tensor transformations, ensuring that the inner product and norm of tensors remain invariant before and after the transformation.

**Decomposability of Tensor Nuclear Norm.** Consider the reduced t-SVD of $\underline{\mathbf{L}}^\star$ given by

$$\underline{\mathbf{L}}^\star = \underline{\mathbf{U}} *_M \underline{\mathbf{S}} *_M \underline{\mathbf{V}}^\top$$

where $\underline{\mathbf{U}} \in \mathbb{R}^{d_1 \times r \times m}$ and $\underline{\mathbf{V}} \in \mathbb{R}^{d_2 \times r \times m}$ are orthogonal tensors, and $\underline{\mathbf{S}} \in \mathbb{R}^{r \times r \times m}$ is an f-diagonal tensor. We define the projection operators $\mathcal{P}_\star(\cdot)$ and $\mathcal{P}_{\star^\perp}(\cdot)$ as follows:

$$\mathcal{P}_\star(\underline{\mathbf{T}}) = \underline{\mathbf{U}} *_M \underline{\mathbf{U}}^\top *_M \underline{\mathbf{T}} + \underline{\mathbf{T}} *_M \underline{\mathbf{V}} *_M \underline{\mathbf{V}}^\top - \underline{\mathbf{U}} *_M \underline{\mathbf{U}}^\top *_M \underline{\mathbf{T}} *_M \underline{\mathbf{V}} *_M \underline{\mathbf{V}}^\top \quad (13)$$

$$\mathcal{P}_{\star^\perp}(\underline{\mathbf{T}}) = (\underline{\mathbf{I}} - \underline{\mathbf{U}} *_M \underline{\mathbf{U}}^\top) *_M \underline{\mathbf{T}} *_M (\underline{\mathbf{I}} - \underline{\mathbf{V}} *_M \underline{\mathbf{V}}^\top). \quad (14)$$

These operators decompose the tensor $\underline{\mathbf{T}}$ into components aligned with the sub-modules t-spanned by $\underline{\mathbf{U}}$ and $\underline{\mathbf{V}}$, and their orthogonal complements, respectively.

As shown in the appendix of Wang et al. (2020), the following properties hold:

a). Any tensor $\underline{\mathbf{T}} \in \mathbb{R}^{d_1 \times d_2 \times m}$ can be uniquely decomposed as $\underline{\mathbf{T}} = \mathcal{P}_\star(\underline{\mathbf{T}}) + \mathcal{P}_{\star^\perp}(\underline{\mathbf{T}})$.

b). The inner product between the projections $\mathcal{P}_\star(\underline{\mathbf{X}})$ and $\mathcal{P}_{\star^\perp}(\underline{\mathbf{Y}})$ is zero, i.e., $\langle \mathcal{P}_\star(\underline{\mathbf{X}}), \mathcal{P}_{\star^\perp}(\underline{\mathbf{Y}}) \rangle = \mathbf{0}$, for all tensors $\underline{\mathbf{X}}, \underline{\mathbf{Y}} \in \mathbb{R}^{d_1 \times d_2 \times m}$.

c). The tubal rank of the projected tensor $\mathcal{P}_\star(\underline{\mathbf{T}})$ is at most twice the rank of $\underline{\mathbf{L}}^\star$, i.e., $r_{\mathsf{tb}}(\mathcal{P}_\star(\underline{\mathbf{T}})) \leq 2 \cdot r_{\mathsf{tb}}(\underline{\mathbf{L}}^\star)$, for all $\underline{\mathbf{T}} \in \mathbb{R}^{d_1 \times d_2 \times m}$.

Additionally, the following properties related to the tensor nuclear norm (TNN) can be established:

a). **(Decomposability of TNN)** For any tensors $\underline{\mathbf{X}}, \underline{\mathbf{Y}} \in \mathbb{R}^{d_1 \times d_2 \times m}$ satisfying $\underline{\mathbf{X}} *_M \underline{\mathbf{Y}}^\top = \mathbf{0}$ and $\underline{\mathbf{X}}^\top *_M \underline{\mathbf{Y}} = \mathbf{0}$, the tensor nuclear norm decomposes additively:

$$\|\underline{\mathbf{X}} + \underline{\mathbf{Y}}\|_\star = \|\mathcal{P}_\star(\underline{\mathbf{X}})\|_\star + \|\mathcal{P}_{\star^\perp}(\underline{\mathbf{Y}})\|_\star.$$

b). **(Norm compatibility inequality)** For any tensor $\underline{\mathbf{T}} \in \mathbb{R}^{d_1 \times d_2 \times m}$, the tensor nuclear norm can be related to the tensor Frobenius norm and the tensor rank as follows:

$$\|\underline{\mathbf{T}}\|_\star \leq \sqrt{r_{\mathsf{tb}}(\underline{\mathbf{T}}) \cdot m} \cdot \|\underline{\mathbf{T}}\|_{\mathrm{F}}.$$

### B.2 ADDITIONAL LEMMAS

The concept of covering and packing numbers play an important role in our remaining analysis.

**Definition B.6** (Covering and Packing Numbers, (Raskutti et al., 2011)). Consider a compact metric space consisting of a set $\mathcal{S}$ and a metric $\varrho : \mathcal{S} \times \mathcal{S} \to \mathbb{R}^+$

- An $\epsilon$-covering of $\mathcal{S}$ with respect to the metric $\varrho$ is a collection $\left\{\underline{\mathbf{L}}^1, \ldots, \underline{\mathbf{L}}^N\right\} \subset \mathcal{S}$ such that for all $\underline{\mathbf{L}} \in \mathcal{S}$, there exists some $\underline{\mathbf{L}}^i, i \in \{1, \ldots, N\}$ with $\varrho\left(\underline{\mathbf{L}}, \underline{\mathbf{L}}^i\right) \leq \epsilon$. The $\epsilon$-covering number $N(\epsilon; \mathcal{S}, \varrho)$ is the cardinality of the smallest $\epsilon$-covering.

- A $\delta$-packing of $\mathcal{S}$ with repsect to the metric $\varrho$ is a collection $\left\{\underline{\mathbf{L}}^1, \ldots, \underline{\mathbf{L}}^M\right\} \subset \mathcal{S}$ such that $\varrho\left(\underline{\mathbf{L}}^i, \underline{\mathbf{L}}^j\right) > \delta$ for all distinct $i, j$. The $\delta$-packing number $M(\delta; \mathcal{S}, \varrho)$ is the cardinality of the largest $\delta$-packing.

Covering and packing numbers provide essentially the same measure of the massiveness of a set. In particular, the relation between covering number and packing number is described as $M(2\epsilon; \mathcal{S}, \varrho) \leq N(\epsilon; \mathcal{S}, \varrho) \leq M(\epsilon; \mathcal{S}, \varrho)$. These two quantities exhibit the same scaling behavior as $\epsilon \to 0$. Additionally, the logarithm of the covering number $\log N(\epsilon; \mathcal{S}, \varrho)$ is known as the metric entropy of $\mathcal{S}$ with respect to $\varrho$.

**Definition B.7** (entropy number). Consider a quasi-Banach space consisting a compact set $\mathcal{S}$ and a quasi-metric $\varrho$. $N(\epsilon; \mathcal{S}, \varrho)$ denotes the covering number with radius $\epsilon$. For $k = 1, 2, \ldots$ the dyadic entropy number is defined as

$$\epsilon_k(\mathcal{S}, \varrho) := \inf\{\epsilon > 0 : N(\epsilon; \mathcal{S}, \varrho) \leq 2^{k-1}\}.$$

**Lemma B.8** (Entropy number of Schatten-$q$ ball ([Hinrichs et al., 2017](#))). *Consider a $d \times d$-dimensional vector space. Suppose $\mathcal{S}$ is a $S_q$ unit-ball and $\varrho$ is the metric induced by F-norm. Then, we have the following theorem for $q \leq 2$:*

$$\epsilon_k(\mathbb{B}^d_{S_q}(1), \|\cdot\|_F) \asymp_q \begin{cases} 1 & \text{for } 1 \leq k \leq d & \text{(15a)} \\ \left(\dfrac{d}{k}\right)^{\frac{1}{q}-\frac{1}{2}} & \text{for } d \leq k \leq d^2 & \text{(15b)} \\ 2^{-\frac{k}{d^2}} \cdot d^{\frac{1}{2}-\frac{1}{q}} & \text{for } k \geq d^2. & \text{(15c)} \end{cases}$$

## C  THEORETICAL RESULTS FOR UNDERSTANDING OF DUAL-LEVEL SPARSITY FOR GLM

We now present a unified minimax analysis of our dual-level sparse framework. We begin by establishing lower bounds on the Frobenius-norm loss for estimators operating over dual-level sparse parameter spaces, showing the fundamental difficulty imposed by having to both *identify nonzero frequency components* and *estimate each low-rank slice*. Then, we turn to upper bounds by analyzing constrained least squares estimators that match these lower limits. Finally, we extend our discussion to more general $\ell_p(S_q)$-type parameter spaces, deriving analogous minimax rates and discussing how the interplay between $\ell_p$-sparsity and Schatten-$q$ low-rankness affects the overall error.

**(I) Minimax Lower Bounds Over $\ell_0$-$S_0$ or $\ell_0$-$S_q$-Balls.**  We first characterizes the *worst-case* error any estimator must incur under two types of dual-level sparse constraints:

(a) $\ell_0$-$S_0$ case ($q = 0$), meaning we have at most $s$ nonzero frequency components and each slice is rank at most $r$.

(b) $\ell_0$-$S_q$ case with $q \in (0, 1]$, meaning we again allow at most $s$ active frequencies, but each slice's rank structure is relaxed into a Schatten-$q$-ball of radius $R$.

**Theorem C.1.** *Consider the linear observation model $\underline{\mathbf{Y}}_i = \underline{\mathbf{L}}^* + \underline{\mathbf{E}}_i$ under dual-level sparse spectrum, with $n$ i.i.d. observations and noise variance $\sigma^2$. Then the following bounds hold:*

*(a) If $q = 0$ and $\underline{\mathbf{L}}^* \in \mathbf{T}_{0,0}(s, r)$, any measurable estimator $\hat{\underline{\mathbf{L}}}$ satisfies*

$$\inf_{\hat{\underline{\mathbf{L}}}} \sup_{\underline{\mathbf{L}} \in \mathbf{T}_{0,0}(s,r)} \mathbb{P}\Big( \|\hat{\underline{\mathbf{L}}} - \underline{\mathbf{L}}\|_{\mathrm{F}}^2 \geq C_\ell \, \frac{\sigma^2}{n} \Big[ s \, \log\big(\tfrac{em}{s}\big) + s \, r \, d \Big] \Big) \geq \tfrac{1}{2},$$

*implying*

$$\mathfrak{M}(\mathbf{T}_{0,0}(s, r)) \geq \tfrac{1}{2} \, C_\ell \, \frac{\sigma^2}{n} \Big[ s \, \log\big(\tfrac{em}{s}\big) + s \, r \, d \Big].$$

*(b) If $q \in (0, 1]$ and $\underline{\mathbf{L}}^* \in \mathbf{T}_{0,q}(s, R)$, for any estimator $\hat{\underline{\mathbf{L}}}$,*

$$\inf_{\hat{\underline{\mathbf{L}}}} \sup_{\underline{\mathbf{L}} \in \mathbf{T}_{0,q}(s,R)} \mathbb{P}\Big( \|\hat{\underline{\mathbf{L}}} - \underline{\mathbf{L}}\|_{\mathrm{F}}^2 \geq C_\ell \Big\{ \tfrac{\sigma^2}{n} s \, \log\big(\tfrac{em}{s}\big) + s \, R \, \big(\tfrac{\sigma^2}{n} d\big)^{1 - \frac{q}{2}} \Big\} \Big) \geq \tfrac{3}{8},$$

*which likewise implies*

$$\mathfrak{M}(\mathbf{T}_{0,q}(s, R)) \geq \tfrac{3}{8} \, C_\ell \Big\{ \tfrac{\sigma^2}{n} s \log\big(\tfrac{em}{s}\big) + s \, R \, \big(\tfrac{\sigma^2}{n} d\big)^{1 - \frac{q}{2}} \Big\}.$$

From the expressions in Theorem C.1, we see that the *estimation complexity* has two components:

- (a) A term of order $s \log(\tfrac{em}{s})$ captures the combinatorial cost of identifying which $s$ out of $m$ frequency slices are nonzero.
- (b) A second term quantifies the difficulty of *estimating each slice*, either in the rank-$r$ case ($s \, r \, d$) or in the Schatten-$q$ case $\big[ s \, R \, (\tfrac{\sigma^2}{n} d)^{1 - \frac{q}{2}} \big]$.

Hence, Theorem C.1 highlights a fundamental trade-off in learning dual-level sparse structures from noisy observations.

**(II) Minimax Upper Bounds Over $\ell_0$-$S_0$ or $\ell_0$-$S_q$-Balls.**  We next confirm that these lower bounds are sharp by analyzing a *constrained least-squares* (CLS) estimator, defined for $q \in [0, 1]$ via

$$\hat{\underline{\mathbf{L}}}_q \in \argmin_{\underline{\mathbf{L}} \in \mathbf{T}_{0,q}(s,R)} \|\underline{\mathbf{L}} - \bar{\underline{\mathbf{Y}}}\|_F^2, \tag{16}$$

where $\bar{\underline{\mathbf{Y}}} = \frac{1}{n} \sum_{i=1}^n \underline{\mathbf{Y}}_i$. One shows that $\hat{\underline{\mathbf{L}}}_q$ attains an $\varepsilon$-accurate solution with high probability as soon as $\varepsilon^2$ is on the same order as the lower-bound terms in Theorem C.1.

**Theorem C.2.** *Under the same dual-level sparse setups and i.i.d. noise model, the following hold:*

(a) *If $q = 0$ and we form the estimator $\hat{\underline{\mathbf{L}}}_0$ by minimizing $\|\underline{\mathbf{L}} - \bar{\underline{\mathbf{Y}}}\|_{\mathrm{F}}^2$ over $\mathbf{T}_{0,0}(s,r)$, then for any* $\epsilon^2 \geq C_u \frac{\sigma^2}{n} \left[ s \log(\frac{em}{s}) + s\,r\,d \right]$,

$$\sup_{\underline{\mathbf{L}} \in \mathbf{T}_{0,0}(s,r)} \|\hat{\underline{\mathbf{L}}}_0 - \underline{\mathbf{L}}\|_{\mathrm{F}}^2 \leq \epsilon^2, \tag{17}$$

*with probability at least $1 - C_1 \exp(-C_2\,n\,\epsilon^2)$.*

(b) *If $q \in (0,1]$ and we form $\hat{\underline{\mathbf{L}}}_q$ similarly over $\mathbf{T}_{0,q}(s,R)$, then for any $\epsilon^2 \geq C_u \left\{ \frac{\sigma^2}{n} s \log(\frac{em}{s}) + s\,R\left(\frac{\sigma^2}{n}d\right)^{1-\frac{q}{2}} \right\}$,*

$$\sup_{\underline{\mathbf{L}} \in \mathbf{T}_{0,q}(s,R)} \|\hat{\underline{\mathbf{L}}}_q - \underline{\mathbf{L}}\|_{\mathrm{F}}^2 \leq \epsilon^2, \tag{18}$$

*holds with probability at least $1 - C_1 \exp(-C_2\,n\,\epsilon^2)$.*

These upper bounds match the lower bounds (Theorem C.1) up to constant factors, establishing the $\ell_0$-$S_0$ or $\ell_0$-$S_q$ rates as *optimal*. Consequently, we arrive at the minimax rates:

$$\mathfrak{M}(\mathbf{T}_{0,0}(s,r)) \asymp \frac{\sigma^2}{n} \left[ s \log(\tfrac{em}{s}) + s\,r\,d \right] \quad \text{and} \quad \mathfrak{M}(\mathbf{T}_{0,q}(s,R)) \asymp \frac{\sigma^2}{n} \left[ s \log(\tfrac{em}{s}) + s\,R\left(\tfrac{\sigma^2}{n}d\right)^{1-\frac{q}{2}} \right].$$

Thus, Theorem C.2 shows that the above CLS estimators are *rate-optimal*.

**Remark C.3.** As a corollary, the joint results of Theorems C.1 and C.2 show that the minimax rates match up to constant factors in both the $\ell_0$-$S_0$ and $\ell_0$-$S_q$ scenarios. Specifically, for $q = 0$:

$$\mathfrak{M}(\mathbf{T}_{0,0}(s,r)) \asymp \frac{\sigma^2}{n} \left[ s \log(\tfrac{em}{s}) + s\,r\,d \right],$$

and for $q \in (0,1]$:

$$\mathfrak{M}(\mathbf{T}_{0,q}(s,R)) \asymp \frac{\sigma^2}{n} \left[ s \log(\tfrac{em}{s}) + s\,R\left(\tfrac{\sigma^2}{n}d\right)^{1-\frac{q}{2}} \right].$$

**(III) Minimax Rates Over General $\ell_p(S_q)$-Balls.** Finally, we move beyond the hard-sparsity constraints to the more flexible $\ell_p(S_q)$ spaces:

$$\mathbf{T}_{p,q}(R) = \left\{ \underline{\mathbf{L}} : \ \left\| \underline{\mathbf{L}} \right\|_{\ell_p(S_q)}^p \leq R \right\}.$$

By combining similar lower/upper bound arguments (now requiring more subtle entropy and covering results) we arrive at Theorem C.4: To avoid over-complicated scenarios, we assume

$$\begin{cases} \log m < R\left(\frac{n}{\sigma^2 d}\right)^{\frac{p}{2}} \leq \frac{m}{\log m} \\ \log m < R\left(\frac{n}{\sigma^2 \log m}\right)^{\frac{p}{2}} \leq \frac{m}{\log m} \\ c_1 d < R^{\frac{q}{\sigma^2 p}}\left(\frac{n}{d}\right)^{\frac{q}{2}} < C_1 d. \end{cases} \tag{19}$$

**Theorem C.4** (Minimax Rates for $\ell_p(S_q)$-balls)**.** *Suppose $p, q \in (0,1]$ and condition Eq. (19) holds to avoid degenerate parameter ranges. Then the minimax risk over the $\ell_p(S_q)$-ball is:*

$$\mathfrak{M}(\mathbf{T}_{p,q}(R)) = \inf_{\hat{\underline{\mathbf{L}}}} \sup_{\underline{\mathbf{L}}^* \in \mathbf{T}_{p,q}(R)} \mathbb{E}\left[ \|\hat{\underline{\mathbf{L}}} - \underline{\mathbf{L}}^*\|_{\mathrm{F}}^2 \right]$$

$$\asymp \begin{cases} R\left(\frac{n}{d\sigma^2}\right)^{\frac{p-2}{2}} + R\left(\frac{n}{\sigma^2 \log m}\right)^{\frac{p-2}{2}}, & p > q, \\ R^{\frac{q}{p}}\left(\frac{n}{d\sigma^2}\right)^{\frac{q-2}{2}} + R\left(\frac{n}{\sigma^2 \log m}\right)^{\frac{p-2}{2}}, & p \leq q, m > d^2, \\ R^{\frac{q}{p}}\left(\frac{n}{d\sigma^2}\right)^{\frac{q-2}{2}}, & p \leq q, m \leq d^2. \end{cases} \tag{20}$$

Examining Eq. (20) reveals three distinct regimes:

- $p > q$. The $\ell_p$-type group sparsity dominates, rendering the Schatten-$q$ penalties secondary.

- $p \leq q$, $m > d^2$. Both frequency-domain $\ell_p$-grouping and intra-slice Schatten-$q$-norm jointly control the risk, leading to a sum of two terms.

- $p \leq q$, $m \leq d^2$. The $S_q$-ball effectively saturates the error, making the risk independent of $m$.

Hence, $\ell_p(S_q)$ quasi-norms admit richer, *soft* sparsity structures beyond the hard $\ell_0$-$S_0$ or $\ell_0$-$S_q$ constraints, but yield analogous minimax phenomena once careful covering-entropy or packing arguments are applied.

In conclusion, these results unify the lower and upper bounds for dual-level sparse tensor estimation under both hard-sparsity ($\ell_0$-$S_0$ or $\ell_0$-$S_q$) and soft-sparsity ($\ell_p(S_q)$) constraints. The key takeaway is that the *minimax error* always balances identifying relevant frequencies with estimating each slice's rank structure. Hard- and soft-sparsity assumptions shift how these two aspects interact, but the big-picture story remains consistent: multi-frequency, low-rank modeling carries a fundamental combinatorial cost (for frequency selection) plus a continuous cost (for matrix parameter estimation). These findings rigorously justify the dual-level sparsity approach and characterize the fundamental limits of any estimator hoping to learn such structured tensors from noisy observations.

**Remark C.5** (On the Practical Validation and Use of Theorem 4.2). While Theorem 4.2 establishes minimax lower bounds for dual-sparse tensor recovery under the $\ell_p$–Schatten-$q$ prior, directly validating these rates through simulation is inherently difficult. This is because minimax guarantees are defined over worst-case estimators, which are often inaccessible—especially under nonconvex regularization. Even for vector-valued estimators with $\ell_p(\ell_q)$ sparsity, constructing such minimax-optimal procedures remains an open challenge when $(p, q)$ are general.

## C.1 Sketch of Minimax Proofs under Dual-Level Sparsity

Our proofs of the minimax bounds proceed by considering three successively more general forms of dual-level sparsity, each requiring distinct technical arguments due to their underlying structural assumptions.

**(1) Hard–Hard Dual-Level Sparsity $\mathbf{T}_{0,0}(s, r)$.** When both the number of active frequency components and the rank of each active slice are bounded by $s$ and $r$, respectively, we construct a *packing set* that simultaneously encodes frequency sparsity and low-rankness:

1. *Support selection:* First, choose which $s$ frequency slices (out of $m$) can be nonzero, ensuring a combinatorial factor $\binom{m}{s}$.

2. *Within-slice low-rank matrices:* Next, for each chosen frequency slice, define a family of low rank $\leq r$ matrices whose entries are set to a suitable scale $\delta$, ensuring the separation properties
$$\|\underline{\mathbf{L}}^i - \underline{\mathbf{L}}^j\|_F^2 \geq c_0 \, s \, r \, d \, \delta^2 \quad \text{and} \quad \|\underline{\mathbf{L}}^i - \underline{\mathbf{L}}^j\|_F^2 \leq 2 \, s \, r \, d \, \delta^2,$$
while also achieving large enough cardinality for the packing set.

Applying Fano's inequality to this well-separated set establishes a lower bound by appropriately choosing $\delta \propto \sqrt{\frac{\sigma^2}{n} \left[ s \log(\frac{em}{s}) + s \, r \, d \right]}$. For the matching upper bound, we employ a constrained least squares estimator and use covering number arguments (based on matrix rank$\leq r$ covers and frequency-support combinatorics) to show it achieves the same rate.

**(2) Hard–Soft Dual-Level Sparsity $\mathbf{T}_{0,q}(s, R)$.** When the frequency sparsity is still hard-constrained by $s$, but each slice now belongs to a *soft low-rank* Schatten-$q$ ball of radius $R$, the proof becomes more intricate:

- We decompose into (i) tensors with *different* frequency supports, incurring a cost of $\frac{\sigma^2}{n} s \log(\frac{em}{s})$, and (ii) tensors with the *same* support but *different* within-slice Schatten-$q$ structures, contributing $s R \left(\frac{\sigma^2}{n} d\right)^{1 - \frac{q}{2}}$.

- By carefully combining these two cases (through union bounds and entropy number estimates for Schatten-$q$ sets), we derive matching lower and upper bounds, again establishing minimax optimality.

**(3) Soft–Soft Dual-Level Sparsity $\mathbf{T}_{p,q}(R)$.** Finally, in the most general setting, both frequency sparsity and low-rankness are relaxed into an $\ell_p(S_q)$ quasi-norm. Here, the analysis must carefully track how $p$ and $q$ govern *group-level* versus *within-group* regularity:

- When $p > q$, the $\ell_p$ penalty dominates, so the rates follow primarily from frequency-sparsity arguments.

- If $p \leq q$, we observe a phase transition depending on whether $m$ exceeds $d^2$. For $m > d^2$, the two penalties are both active, summing their respective complexities; for $m \leq d^2$, the Schatten-$q$ term saturates the error, rendering it independent of $m$.

Technically, this requires sophisticated entropy tools, in particular Schütt's theorem on entropy numbers for vector-valued sequence spaces, and a careful *chaining* analysis that tracks interactions across frequencies and singular values in each slice.

**Unified Upper Bound Approach.** Although the detailed proofs vary, the estimation upper bounds all follow a similar template via empirical process theory:

- For $\mathbf{T}_{0,0}(s, r)$, we rely on discrete packing/covering of finite-support rank$\leq r$ matrices.

- For $\mathbf{T}_{0,q}(s, R)$, we combine discrete frequency selection with known entropy results for Schatten-$q$ balls.

- For $\mathbf{T}_{p,q}(R)$, we perform a chaining argument on $\ell_p(S_q)$ quasi-norms, applying Schütt-type entropy estimates.

In each case, bounding the least squares estimator's risk at the matching lower-bound rate confirms it is optimal up to constants.

Across these three regimes of *hard–hard*, *hard–soft*, and *soft–soft* dual-level sparsity, the main technical challenge is to *precisely quantify the geometric complexity* imposed by the interplay of frequency sparsity and low-rankness (whether hard or soft). Once we derive suitable packing/covering or entropy bounds, standard empirical process arguments transform that complexity into matching lower and upper rates. Consequently, the minimax results show a consistent story: learning multi-frequency structures involves a combinatorial cost from selecting active frequencies plus a continuous cost from estimating each low-rank (or Schatten-$q$) slice, culminating in the optimal rates detailed in our main theorems.

### C.2    PROOF OF THEOREM C.1

#### C.2.1    PROOF OF CASE (A)

*Proof.* For case (a), consider the $\frac{srd}{32}$-packing set $\widetilde{\mathbf{T}}_{0,0}(s, r) = \{\underline{\mathbf{L}}^1, \ldots, \underline{\mathbf{L}}^N\}$ constructed in Lemma C.9, where $N$ is its cardinality. We set all nonzero entries of each $\underline{\mathbf{L}}^i \in \widetilde{\mathbf{T}}_{0,0}(s, r)$ to be 1, and define $\vartheta^i = \underline{\mathbf{L}}^i \delta$, with $\delta$ a parameter to be determined. Since each $M(\underline{\mathbf{L}}^i)$ has at most $s\,r\,d$ nonzero elements, for any $\vartheta^i \neq \vartheta^j$, it follows that

$$\|\vartheta^i - \vartheta^j\|_{\mathrm{F}}^2 = \|M(\vartheta^i) - M(\vartheta^j)\|_{\mathrm{F}}^2 = \delta^2 \|M(\underline{\mathbf{L}}^i) - M(\underline{\mathbf{L}}^j)\|_{\mathrm{F}}^2 \leq 2\,s\,r\,d\,\delta^2, \quad \forall i, j \in [N]. \quad (21)$$

On the other hand, from the construction of $\widetilde{\mathbf{T}}_{0,0}(s, r)$, we also have

$$\|\vartheta^i - \vartheta^j\|_{\mathrm{F}}^2 \geq \tfrac{1}{32}\,s\,r\,d\,\delta^2, \quad \forall i, j \in [N]. \quad (22)$$

Using standard mutual information arguments (Wu, 2017) gives

$$\mathsf{I}(\bar{\underline{\mathbf{Y}}}; \psi) \leq \frac{1}{\binom{N}{2}} \sum_{i \neq j} \mathsf{KL}(\vartheta^i \,\|\, \vartheta^j)$$

$$= \frac{1}{\binom{N}{2}} \sum_{i \neq j} \frac{n}{2\,\sigma^2} \|\vartheta^i - \vartheta^j\|_{\mathrm{F}}^2 \quad (23)$$

$$\leq \frac{n}{\sigma^2}\,s\,r\,d\,\delta^2,$$

where $\mathsf{KL}(\cdot \,\|\, \cdot)$ is the Kullback–Leibler divergence, and the last step uses Eq. (21). By applying Fano's inequality (Cover, 2006), we obtain

$$\mathbb{P}(\hat{\vartheta} \neq \psi) \geq 1 - \frac{\frac{n}{\sigma^2}\, s\, r\, d\, \delta^2 + \log 2}{\log N},$$

where $\psi$ is uniformly distributed over the packing set $\widetilde{\mathbf{T}}_{0,0}(s,r)$. To ensure $\mathbb{P}(\hat{\vartheta} \neq \psi) \geq \frac{1}{2}$, it suffices to choose

$$\delta = \tfrac{1}{2} \sqrt{\left[\left(c_1\, r\, s\, d\right) + \left(c_2\, s\, \log(\tfrac{em}{s})\right)\right] \frac{\sigma^2}{s\, r\, d\, n}}.$$

Substituting this choice into Eq. (22) and invoking Lemma C.9 completes the proof, showing that

$$\inf_{\hat{\mathbf{L}}} \ \sup_{\mathbf{L} \in \mathbf{T}_{0,0}(s,r)} \mathbb{P}\!\left[\|\hat{\mathbf{L}} - \mathbf{L}\|_{\mathsf{F}}^2 \geq \tfrac{c\,\sigma^2}{n}\left(r\, s\, d + s\, \log \tfrac{em}{s}\right)\right] \geq \tfrac{1}{2}.$$

Finally, a Markov's inequality argument yields the same lower bound in expectation. $\qquad\square$

### C.2.2 THEORETICAL TOOLS: GILBERT-VARSHAMOV THEOREMS

The proof of case (a) requires a $\frac{srd}{32}$-packing set $\widetilde{\mathbf{T}}_{0,0}(s,r)$. Before proceeding with the construction of this packing set, we first introduce several versions of the Gilbert-Varshamov theorem that will be crucial for our analysis. These results provide guarantees on the existence of well-separated binary and $K$-ary codes.

The first version deals with binary codes containing the zero vector:

**Lemma C.6** (Gilbert-Varshamov Theorem for Binary Codes (Tsybakov & Schatzen, 2011)). *Consider a length-$m$ code with binary symbols (2-ary coding). There exists a subset $\Omega = \{\omega_0, \ldots, \omega_N\}$ of the code book where $\omega_i \in \{0,1\}^m$ that satisfies:*

- *The zero vector is included: $\omega_0 = (0, \ldots, 0)$.*

- *The minimum Hamming distance is bounded: $d_H(\omega_i, \omega_j) \geq m/8$ for all $0 \leq i < j \leq N$.*

- *The set size is exponential: $N \geq 2^{m/8}$.*

This lemma guarantees the existence of a large set of well-separated binary vectors, which will be used to construct the low-rank matrices $M(\underline{\mathbf{L}})_{:,:,k}$ within the $k$-th ($k \in [m]$) frequency component of $\underline{\mathbf{L}} \in \mathbb{R}^{d_1 \times d_2 \times m}$ in the transformed domain defined by $M$ in Eq. (1).

Then, for more general alphabets beyond $\{0, 1\}$, we have:

**Lemma C.7** (Gilbert-Varshamov Theorem for $K$-ary Codes). *For a length-$m$ code with $K$-ary symbols, there exists a $\varrho$-separated set whose cardinality is at least:*

$$N_K(m, \varrho) \geq \frac{K^m}{\sum_{i=0}^{\varrho-1} \binom{m}{i}(K-1)^i}$$

*where $\varrho$ denotes the minimum Hamming distance between any two codewords.*

Further, for codes restricted to a Hamming sphere, we have:

**Lemma C.8** (Gilbert-Varshamov Theorem for Bounded-Weight Codes, (Wu, 2017)). *Consider the Hamming sphere of radius $s$ for a length-$m$ code with $K$-ary symbols. There exists a $\varrho$-separated set within this sphere with cardinality at least:*

$$N_K(m, s, \varrho) \geq \frac{\binom{m}{s}(K-1)^s}{\sum_{i=0}^{\varrho-1} \binom{m}{i}(K-1)^i} \tag{24}$$

A particularly useful special case occurs when $\varrho = c_1 s$ and we consider binary coding ($K = 2$):

$$N_2(m, s, \varrho) \geq (em/s)^{c_2 s} \tag{25}$$

where $c_1$ and $c_2$ are absolute constants.

**Lemma C.9** (Existence of a Packing Set). *There exists a packing set $\widetilde{\mathbf{T}}_{0,0}(s,r) \subset \mathbf{T}_{0,0}(s,r)$ satisfying the following properties:*

1. ***Frequency sparsity constraint:*** *Each tensor in $\widetilde{\mathbf{T}}_{0,0}(s,r)$ has at most $s$ nonzero frequency components.*

2. ***Low-rank structure:*** *Each frequency component is represented by a matrix of rank at most $r$.*

3. ***Separation property:*** *Any two distinct tensors $\underline{\mathbf{L}}^1, \underline{\mathbf{L}}^2 \in \widetilde{\mathbf{T}}_{0,0}(s,r)$ satisfy:*

$$\left\|\underline{\mathbf{L}}^1 - \underline{\mathbf{L}}^2\right\|_{\mathrm{F}}^2 \geq \frac{srd_1}{32}. \tag{26}$$

4. ***Cardinality:*** *The packing set has size at least:*

$$|\widetilde{\mathbf{T}}_{0,0}(s,r)| \geq (em/s)^{cs} \cdot 2^{rsd_1/32} \geq \exp(c_1 rsd_1 + c_2 s \log(em/s)). \tag{27}$$

*Proof.* We now construct our packing set $\widetilde{\mathbf{T}}_{0,0}(s,r) \subset \mathbf{T}_{0,0}(s,r)$ through a sequence of carefully designed steps. This packing set must be sufficiently rich to capture the essential complexity of $\mathbf{T}_{0,0}(s,r)$. The construction must simultaneously enforce *frequency sparsity bounded by $s$*, maintain the *low-rank structure within each frequency component characterized by $r$*, and ensure *good separation properties* across all constructed sub-sets at each step.

**Step 1: Selection of Non-zero Frequencies.** First, we determine a subset $\tilde{\boldsymbol{\Gamma}} \subset \boldsymbol{\Gamma}$ of the support of the $m$ frequency components. This step establishes the *frequency sparsity pattern* of our tensors $\underline{\mathbf{L}}$ in the transformed domain defined by $M(\cdot)$. We proceed as follows:

1) *Ensure frequency sparsity*: First, we use a 2-ary code of length $m$ on the Hamming sphere of radius $s$ to represent possible support patterns.

2) *Ensure sufficient separation*: Second, by requiring the code to be $s/4$-separated, we can guarantee the existence of a set $\tilde{\boldsymbol{\Gamma}}$ with a minimum number of frequency patterns by Lemma C.8:

$$N_2(m,s,s/4) \geq (em/s)^{cs} \geq \exp(cs\log(em/s)) \tag{28}$$

where $c$ is an absolute constant.

**Step 2: Construction of Low-rank Matrices.** We then consider constructing appropriate low-rank matrices. Without loss of generality, assume $d_1 \geq d_2$. Motivated by Klopp (2015), we first construct the set of matrices $\mathbf{A}_{\text{low-rank}}$ as follows:

For positions $(i,j)$ of a matrix $\mathbf{A} \in \mathbf{A}_{\text{low-rank}}$ with $i \leq r$ and $j \in [d_2]$, we set $\mathbf{A}_{i,j} \in \{0,1\}$, and for all other positions (i.e., $i > r$), we set $\mathbf{A}_{i,j} = 0$.

Then, we can ensure that $\text{rank}(\mathbf{A}) \leq r, \forall \mathbf{A} \in \mathbf{A}_{\text{low-rank}}$. Further by Lemma C.6, this construction yields a $\{0,1\}$-code of length $rd_1$, for which we can find a subset $\{\mathbf{0}, \mathbf{A}^1, \ldots, \mathbf{A}^{N_0}\}$ satisfying:

i) *Sufficiently many low-rank patterns*: $N_0 \geq 2^{rd_1/8}$.

ii) *Sufficient separation*: The Hamming distance $d_H(\mathbf{A}^i, \mathbf{A}^j) \geq rd_1/8$ for all $0 \leq i < j \leq N_0$.

In sumamry, by construction, we can find the matrix set $\tilde{\mathbf{A}}_{\text{low-rank}} := \{\mathbf{A}^1, \ldots, \mathbf{A}^{N_0}\}$ that satisfy:

$$|\tilde{\mathbf{A}}_{\text{low-rank}}| \geq 2^{rd_1/8}, \quad \text{rank}(\mathbf{A}^i) \leq r, \quad \left\|\mathbf{A}^i\right\|_{\mathrm{F}}^2 \leq rd_1, \ \left\|\mathbf{A}^i - \mathbf{A}^j\right\|_{\mathrm{F}}^2 \geq rd_1/8, \quad \text{and} \quad \left\|\mathbf{A}^i\right\|_{\mathrm{F}}^2 \geq rd_1/8. \tag{29}$$

**Step 3: Assign Low-rank Patterns for a Fixed Frequency Pattern.** Now have founded the $s$-sparsity frequency support patterns in $\tilde{\boldsymbol{\Gamma}}$ and the $r$-low-rank frequency matrices in $\tilde{\mathbf{A}}_{\text{low-rank}}$. We then need to assign appropriate low-rank patterns to each selected frequency $\boldsymbol{\gamma} \in \tilde{\boldsymbol{\Gamma}}$. This step requires careful consideration of both the separation properties and the cardinality of our construction.

Motivated by the proof of Lemma 3 in Li et al. (2024), we consider the following analysis:

1) From **Step 2**, we know that each frequency component can take $|\tilde{\mathbf{A}}_{\text{low-rank}}| \geq 2^{rd_1/8}$ different low-rank patterns. This gives us a large alphabet size for coding each frequency component.

2) We can view this as a $|\tilde{\mathbf{A}}_{\text{low-rank}}|$-ary coding problem for each selected frequency, where we also need to ensure the resulting codes are $s/2$-separated in Hamming distance.

3) For a fixed frequency sparsity pattern $\gamma \in \tilde{\mathbf{\Gamma}}$, we can lower bound the **cardinality of the resulting set $\tilde{\mathbf{T}}_\gamma$** according to Lemma C.7 as follows:

$$
\begin{aligned}
N_{|\tilde{\mathbf{A}}_{\text{low-rank}}|}(s, s/2) &\geq \frac{(2^{rd_1/8})^s}{\sum_{i=0}^{s/2-1} \binom{s}{i}(2^{rd_1/8}-1)^i} \\
&\geq \frac{(2^{rd_1/8})^s}{\sum_{i=0}^{s/2-1} \binom{s}{i}(2^{rd_1/8})^{s/2-1}} \\
&\geq \frac{(2^{rd_1/8})^s}{2^s \cdot (2^{rd_1/8})^{s/2-1}} \\
&= \frac{(2^{rd_1/8})^{s/2+1}}{2^s} \\
&= 2^{rsd_1/16-s} \geq 2^{rsd_1/32}
\end{aligned}
\tag{30}
$$

The inequalities above are derived through the following steps. The first inequality follows from the application of the Gilbert-Varshamov theorem for $K$-ary codes. The second inequality is obtained by upper bounding $(2^{rd_1/8})^i - 1$ with $(2^{rd_1/8})^{s/2-1}$. The third inequality results from bounding the sum of binomial coefficients by $2^s$. Finally, the last inequality holds under the assumption that $rd_1$ is sufficiently large, specifically $rd_1 \geq 32$.

**Step 4: Integration of Frequency Sparsity and Low-rankness.** Now, for each $\gamma \in \tilde{\mathbf{\Gamma}}$, we have found a set of tensors $\tilde{\mathbf{T}}_\gamma$ specified by $\gamma$. Then, we totally found a set $\tilde{\mathbf{T}}$ of at least $|\tilde{\mathbf{\Gamma}}| \cdot 2^{rsd_1/32}$ tensors. Next, we will show that $\tilde{\mathbf{T}}$ the ideal packing set $\tilde{\mathbf{T}}_{0,0}(s,r)$ of $\mathbf{T}_{0,0}(s,r)$ we are looking for. We consider two cases:

*Case 4.1: Different Frequency Sparsity Patterns* When any two tensors $\underline{\mathbf{L}}^1, \underline{\mathbf{L}}^2 \in \tilde{\mathbf{T}}$ in our construction have different frequency supports $\gamma^1, \gamma^2 \in \tilde{\mathbf{\Gamma}}$, then according to the construction of $\tilde{\mathbf{\Gamma}}$, the frequency supports $\gamma^1$ and $\gamma^2$ are $s/2$-separated. That means the tensors $\underline{\mathbf{L}}^1, \underline{\mathbf{L}}^2$ have at least $s/4$ different frequency positions, thus the distance

$$
\left\| \underline{\mathbf{L}}^1 - \underline{\mathbf{L}}^2 \right\|_F^2 = \left\| M(\underline{\mathbf{L}}^1) - M(\underline{\mathbf{L}}^2) \right\|_F^2 \geq \frac{s}{4} \cdot \frac{rd_1}{8} = \frac{srd_1}{32}
$$

.

*Case 4.2: Same Frequency Sparsity Pattern* When any two tensors $\underline{\mathbf{L}}^1, \underline{\mathbf{L}}^2 \in \tilde{\mathbf{T}}$ in our construction have the same frequency support $\gamma \in \tilde{\mathbf{\Gamma}}$, then according to the construction of $\tilde{\mathbf{T}}_\gamma$, there are at least $s/2$ different low-rank patterns of $\underline{\mathbf{L}}^1, \underline{\mathbf{L}}^2$. That means the tensors $\underline{\mathbf{L}}^1, \underline{\mathbf{L}}^2$ have at least $s/2$ frequency positions that are $rd_1/8$ separated, thus the distance

$$
\left\| \underline{\mathbf{L}}^1 - \underline{\mathbf{L}}^2 \right\|_F^2 = \frac{s}{2} \cdot \frac{rd_1}{8} = \frac{srd_1}{16}
$$

Combining both cases, we can conclude that the cardinality of our constructed set is at least:

$$
|\tilde{\mathbf{T}}| = \prod_{\gamma \in \tilde{\mathbf{\Gamma}}} |\tilde{\mathbf{T}}_\gamma| \geq (em/s)^{cs} \cdot 2^{rsd_1/32} \geq \exp(c_1 rsd_1 + c_2 s \log(em/s)),
\tag{31}
$$

which completes the proof. $\qquad\square$

### C.2.3 PROOF OF CASE (B)

*Proof.* Here, we assume the tensor $\underline{\mathbf{L}}$ has *hard* frequency sparsity (at most $s$ nonzero frequency components) but a *soft* low-rank constraint on each active slice, enforced by a Schatten-$q$ ball of radius $R$. Proving a minimax lower bound under this mixed constraint is more subtle than in case (a). Specifically, we split the argument into two complementary subcases:

- *Subcase 1*: The *location* of the $s$ nonzero frequency slices varies among tensors, but *within each active slice* we use the same low-rank matrix (scaled by some $\delta$). This isolates the combinatorial complexity of deciding which frequency slices are nonzero.

- *Subcase 2*: The *set of active frequency slices is fixed*, but *the matrix in each slice* can differ (still subject to the Schatten-$q$ ball). This reveals the complexity arising from "soft" rank variations within each frequency component.

By analyzing each subcase and then applying a union bound, we show that any estimator must fail on at least one scenario with nontrivial probability, yielding the desired lower bound of order

$$\frac{\sigma^2}{n}\left[s\,\log\!\left(\tfrac{em}{s}\right) + s\,R\left(\tfrac{\sigma^2}{n}\,d\right)^{1-\frac{q}{2}}\right].$$

**Step 1: Constructing A Suitable Matrix Family A.** Before detailing subcases, we define a set of rank$\le r$ matrices $\mathbf{A}$ that, when properly scaled by $\delta$, remain inside the Schatten-$q$ ball of radius $R$. Concretely,

$$\mathbf{A} := \big\{\mathbf{A} : \mathbf{A} \in \delta \times \{0,1\}^{d\times d}, \|\mathbf{A}\|_0 \le r\,d, \mathrm{rank}(\mathbf{A}) \le r\big\},$$

where $\delta$ satisfies:

$$\sum_{i=1}^{r}\sigma_i^2(\mathbf{A}) \le \delta^2\,r\,d \quad\text{and}\quad \sum_{i=1}^{r}\sigma_i^q(\mathbf{A}) \le R.$$

From a simple calculation, letting

$$\delta = \left(R/r\right)^{\frac{1}{q}} d^{-\frac{1}{2}}$$

ensures each $\mathbf{A} \in \mathbf{A}$ actually lies in the Schatten-$q$ ball of radius $R$. This "reference family" $\mathbf{A}$ will be used in subcase 1 to examine which frequency slices are activated.

**Subcase 1: Different Frequency Supports But the Same Slice Matrix.** In this subcase, we focus on "which $s$ slices are nonzero?" while fixing the same rank$\le r$ matrix across those slices.

*(i) Fix a baseline matrix $\mathbf{A} \in \mathbf{A}$.* Since $\|\mathbf{A}\|_{S_q} \le R$ under our choice of $\delta$, we can replicate $\mathbf{A}$ across active slices without violating the soft rank constraint.

*(ii) Choose different frequency supports.* By a combinatorial argument similar to Lemma C.9, we know there exists a family $\tilde{\boldsymbol{\Gamma}}$ of cardinality at least

$$c_1\,s\,\log\!\left(\tfrac{em}{s}\right)$$

that enumerates possible subsets $\boldsymbol{\gamma} \subseteq [m]$ of size $s$. Define

$$\widetilde{\mathbf{T}}_{0,q}(s,R,\mathbf{A}) := \{\boldsymbol{\gamma} \otimes \mathbf{A} : \boldsymbol{\gamma} \in \tilde{\boldsymbol{\Gamma}}\}.$$

Each tensor in this set places the same $\mathbf{A}$ (scaled by $\delta$) in different positions $\boldsymbol{\gamma}$, capturing up to $s$ activated slices.

*(iii) Packing separation.* If two different supports $\boldsymbol{\gamma}^i$ and $\boldsymbol{\gamma}^j$ differ in at least $s/4$ coordinates, the corresponding tensors $\underline{\mathbf{L}}^i, \underline{\mathbf{L}}^j$ differ by at least $c_0\,s\,r\,d\,\delta^2$ in Frobenius norm (for some constant $c_0 > 0$), while they differ by at most $2\,s\,r\,d\,\delta^2$ if the supports overlap substantially. Thus, $\widetilde{\mathbf{T}}_{0,q}(s,R,\mathbf{A})$ forms a $(c_0\,s\,r\,d\,\delta^2)$-packing set.

*(iv) Mutual information & Fano.* As in case (a), using $\mathsf{KL}(\underline{\mathbf{L}}^i\|\underline{\mathbf{L}}^j) \approx \frac{n}{2\,\sigma^2}\|\underline{\mathbf{L}}^i - \underline{\mathbf{L}}^j\|_F^2$ and applying Fano's inequality, we show no estimator can distinguish all elements of $\widetilde{\mathbf{T}}_{0,q}(s,R,\mathbf{A})$ with probability above $1/2$ if $\delta \sim \sqrt{\frac{\sigma^2}{n\,r\,d}\,\log\!\left(\tfrac{em}{s}\right)}$. This step isolates the combinatorial cost $s\log\!\left(\tfrac{em}{s}\right)$ in the final bound.

**Subcase 2: A Fixed Frequency Set But Different Slice Entries.** Now we fix which $s$ slices are nonzero (i.e., fix some $\boldsymbol{\gamma} \in \tilde{\boldsymbol{\Gamma}}$), but allow *different* matrices in each of these slices. Since the slices are constrained by $\|\mathbf{B}_i\|_{S_q} \leq R$, we consider a large packing of $s$-tuples $(\mathbf{B}_1, \ldots, \mathbf{B}_s)$, each $\mathbf{B}_i$ rank-$\leq r$, lying in the Schatten-$q$ ball. Formally,

$$\widetilde{\mathbf{T}}_{0,q}(s, R, \boldsymbol{\gamma}) := \big\{ \underline{\mathbf{L}} : M(\underline{\mathbf{L}})_{:,:,i} = \mathbf{B}_i \text{ for } i \in [s], M(\underline{\mathbf{L}})_{:,:,i} = \mathbf{0} \text{ otherwise}; \mathbf{B} \in (\mathbb{B}^d_{S_q}(R))^s \big\}.$$

Repeating the packing-based argument (analogous to steps 2–4 in Lemma C.9), we build a family in which every pair $\underline{\mathbf{L}}^i, \underline{\mathbf{L}}^j$ differs by at least $\frac{1}{16} s\, r\, d\, \delta^2$ in $\|\cdot\|_{\mathrm{F}}$. An information-theoretic (Fano) calculation then shows no estimator can identify all such slice-tuples simultaneously with high probability (e.g. above 7/8). This yields a second component in the lower bound tied to $s\, R\, (\frac{\sigma^2}{n} d)^{1-\frac{q}{2}}$, reflecting the *intra-slice* degrees of freedom from the Schatten-$q$ ball.

**Combining the Two Subcases Via a Union Bound.** An intuitive way to see the final bound is to note that any estimator $\hat{\underline{\mathbf{L}}}$ must work *both* when the $s$ nonzero slices vary in location (subcase 1) *and* when they are fixed but each slice can vary in a Schatten-$q$ manner (subcase 2). By a union bound, the probability of success in both subcases is at most the sum of success probabilities, ensuring that *some* scenario fails with positive probability. Thus we obtain a probability statement of the form

$$\inf_{\hat{\underline{\mathbf{L}}}} \sup_{\underline{\mathbf{L}} \in \mathbf{T}_{0,q}(s,R)} \mathbb{P}\big[ \|\hat{\underline{\mathbf{L}}} - \underline{\mathbf{L}}\|_{\mathrm{F}}^2 \geq c \big( s \log(\tfrac{em}{s}) + s\, R\, (\tfrac{\sigma^2}{n} d)^{1-\frac{q}{2}} \big) \big] > 0$$

for some constant $c > 0$. A final Markov or Chebyshev step then converts this probability bound to an expectation form, concluding the lower bound in the minimax sense.

More explicitly, one shows that subcase 1 alone forces a risk at least

$$\frac{\sigma^2}{n} \big[ s \log(\tfrac{em}{s}) \big], \quad \text{and subcase 2 alone forces a risk of at least } s\, R\, (\tfrac{\sigma^2}{n} d)^{1-\frac{q}{2}}.$$

So the event of "$\hat{\underline{\mathbf{L}}}$ having small error on *both* subcases" is bounded by the sum of their separate probabilities (i.e. a union bound), leading to the conclusion that *any* estimator must fail on at least one scenario with probability at least a fixed positive constant. From this, we deduce

$$\inf_{\hat{\underline{\mathbf{L}}}} \sup_{\underline{\mathbf{L}} \in \mathbf{T}_{0,q}(s,R)} \mathbb{E}\big[ \|\hat{\underline{\mathbf{L}}} - \underline{\mathbf{L}}\|_{\mathrm{F}}^2 \big] \gtrsim \frac{\sigma^2}{n} \Big[ s\, \log(\tfrac{em}{s}) + s\, R\, \big(\tfrac{\sigma^2}{n} d\big)^{1-\frac{q}{2}} \Big],$$

which completes the proof for case (b). $\qquad \square$

### C.3 Proofs of Theorem C.2

Before formally providing the proof of upper bounds, we provide some useful technical lemmas.

#### C.3.1 Technical Lemmas for Upper Bounds

We first prove the upper bounds for the covering number of the parameter spaces.

**Lemma C.10** (Upper bounds for the covering number). *Denote $N(\mathbf{T}, \|\cdot\|_{\mathrm{F}}, \varepsilon)$ as the $\varepsilon$-covering number of parameter set $\mathbf{T}$.*

*(a) For $q = 0$ and $\varepsilon \in (0, 1]$, let $\mathbb{S}^{md^2-1} := \{\underline{\mathbf{L}} \in \mathbb{R}^{d \times d \times m} : \|\underline{\mathbf{L}}\|_{\mathrm{F}} = 1\}$, then we have*

$$\log N(\varepsilon; \mathbf{T}_{0,0}(s, r) \cap \mathbb{S}^{d^2m-1}, \|\cdot\|_{\mathrm{F}}) \leq s \log \frac{em}{s} + 2srd \log \frac{1}{\varepsilon/\sqrt{s}}.$$

*(b) For $q \in (0, 1]$ and for all $\varepsilon \in [c_q \sqrt{s} R^{\frac{1}{q}} d^{\frac{1}{2}-\frac{1}{q}}, c_q \sqrt{s} R^{\frac{1}{q}}]$,*

$$\log N(\varepsilon; \mathbf{T}_{0,q}(s, R), \|\cdot\|_{\mathrm{F}}) \lesssim_q s \log \frac{em}{s} + s(C_q \frac{sR^{\frac{2}{q}}}{\varepsilon^2})^{\frac{q}{2-q}} d.$$

*Proof.* **Case (a):** Any tensor $\underline{\mathbf{L}} \in \mathbf{T}_{0,0}(s, r)$ has at most $s$ nonzero frontal slices in the transformed domain $M(\cdot)$. Denote the nonzero slice indices by

$$\Gamma(\underline{\mathbf{L}}) = \{ i \in [m] : M(\underline{\mathbf{L}})_{:,:,i} \neq \mathbf{0} \}.$$

Since $|\Gamma(\mathbf{L})| \leq s$, the total number of ways to pick these supports is bounded by $\binom{m}{s}$. Therefore,

$$\log \binom{m}{s} \leq s \, \log\left(\frac{em}{s}\right).$$

This accounts for selecting *which* frequency slices are potentially nonzero.

For each fixed support $\gamma \subseteq [m]$ with $|\gamma| \leq s$, we need to cover the set of matrices $\{\mathbf{A} \in \mathbb{R}^{d \times d} : \|\mathbf{A}\|_F \leq 1, \text{rank}(\mathbf{A}) \leq r\}$ in the $\|\cdot\|_F$ metric by balls of radius $\varepsilon/\sqrt{s}$. It is a standard fact (see, e.g., Recht et al. (2007)) that the $\delta$-covering number for rank$\leq r$ matrices of Frobenius norm at most 1 is upper-bounded by

$$N\left(\delta; \{\mathbf{A} : \|\mathbf{A}\|_{\mathrm{F}} \leq 1, \text{rank}(\mathbf{A}) \leq r\}, \|\cdot\|_{\mathrm{F}}\right) \leq \exp\left(C \, r \, d \, \log(\tfrac{1}{\delta})\right),$$

for some constant $C > 0$. Substituting $\delta = \varepsilon/\sqrt{s} \leq 1$ gives a covering number of order

$$\exp\left(C \, r \, d \, \log(\tfrac{\sqrt{s}}{\varepsilon})\right).$$

Taking logarithms yields

$$\log N\left(\tfrac{\varepsilon}{\sqrt{s}}; \{\mathbf{A} : \|\mathbf{A}\|_{\mathrm{F}} \leq 1, \text{rank}(\mathbf{A}) \leq r\}, \|\cdot\|_{\mathrm{F}}\right) \lesssim r \, d \, \log\left(\tfrac{\sqrt{s}}{\varepsilon}\right).$$

Given $s$ nonzero frequency slices, each can be approximated by a matrix in the covering set with radius $\varepsilon/\sqrt{s}$. Since there are at most $s$ active slices, the total squared error in Frobenius norm sums up to at most $s \cdot (\varepsilon/\sqrt{s})^2 = \varepsilon^2$. Hence,

$$\text{Total covering number} \leq \binom{m}{s} \times \left[\exp\left(C \, r \, d \, \log(\tfrac{\sqrt{s}}{\varepsilon})\right)\right]^s.$$

Taking logarithms and using $\log \binom{m}{s} \leq s \log(\frac{em}{s})$ completes the argument:

$$\log N\left(\varepsilon; \mathbf{T}_{0,0}(s,r) \cap \mathbb{S}^{md^2-1}, \|\cdot\|_{\mathrm{F}}\right) \leq s \log\left(\tfrac{em}{s}\right) + s \left[r \, d \, \log(\tfrac{\sqrt{s}}{\varepsilon})\right] = s \, \log\left(\tfrac{em}{s}\right) + 2 \, s \, r \, d \, \log\left(\tfrac{1}{\varepsilon/\sqrt{s}}\right),$$

possibly absorbing constants into the notation. This completes the proof of Case (a).

**Case (b):** We now analyze the covering number of $\mathbf{T}_{0,q}(s,R)$ by leveraging the entropy properties of $S_q$-balls.

To construct an $\varepsilon$-covering of $\mathbf{T}_{0,q}(s,R)$, we first define a covering set $\widetilde{\mathbb{B}}^d_{S_q}(R)$ of $\mathbb{B}^d_{S_q}(R)$ with respect to the Frobenius norm, where each covering element approximates a matrix within $\mathbb{B}^d_{S_q}(R)$ with an error at most $\frac{\varepsilon}{\sqrt{s}}$.

For any $\mathbf{L} \in \mathbf{T}_{0,q}(s,R)$, its $j$-th frequency component satisfies $M(\mathbf{L})_{::j} \in \mathbb{B}^d_{S_q}(R)$. By the definition of a covering set, there exists a matrix $\mathbf{A}^j \in \widetilde{\mathbb{B}}^d_{S_q}(R)$ such that: $\|\mathbf{A}^j - M(\mathbf{L})_{::j}\|_{\mathrm{F}}^2 \leq \frac{\varepsilon^2}{s}$.

This guarantees that any element in $\mathbf{T}_{0,q}(s,R)$ can be approximated using a combination of $s$ selected frequency components from $m$ total frequencies, each of which is covered by $\widetilde{\mathbb{B}}^d_{S_q}(R)$. Using this, the covering number of $\mathbf{T}_{0,q}(s,R)$ satisfies:

$$N(\varepsilon; \mathbf{T}_{0,q}(s,R), \|\cdot\|_{\mathrm{F}}) \leq \binom{m}{s} \left(N(\tfrac{\varepsilon}{\sqrt{s}}; \mathbb{B}^d_{S_q}(R), \|\cdot\|_{\mathrm{F}})\right)^s.$$

Here, the first term accounts for the number of ways to select $s$ active frequency components, while the second term reflects the covering number for each selected frequency component.

Next, we employ entropy estimates for $S_q$-balls. From Eq. (15b), the entropy number of $\mathbb{B}^d_{S_q}(1)$ satisfies:

$$\epsilon_k(\mathbb{B}^d_{S_q}(1)) = \frac{\varepsilon}{\sqrt{s}} \lesssim_q \left(\frac{d}{k}\right)^{\frac{1}{q}-\frac{1}{2}}.$$

Inverting this relation to express $k$ in terms of $\varepsilon$, we set:

$$k = \log N\left(\frac{\varepsilon}{\sqrt{s}}; \mathbb{B}^d_{S_q}(1), \|\cdot\|_{\mathrm{F}}\right),$$

which, after allowing for a ball radius $R^{1/q}$, leads to:

$$\log N\left(\frac{\varepsilon}{\sqrt{s}}; \mathbb{B}^d_{S_q}(R), \|\cdot\|_{\mathrm{F}}\right) \asymp \left(C_q \frac{s R^{\frac{2}{q}}}{\varepsilon^2}\right)^{\frac{q}{2-q}} d.$$

The constraint on $\varepsilon$ ensures that $k \in [d, d^2]$, keeping the bounds valid.

Combining these results, we obtain:

$$\log N(\varepsilon; \mathbf{T}_{0,q}(s,R), \|\cdot\|_{\mathrm{F}}) \lesssim_q s \log \frac{em}{s} + s \log N\left(\frac{\varepsilon}{\sqrt{s}}; \mathbb{B}^d_{S_q}(R), \|\cdot\|_{\mathrm{F}}\right)$$

$$\lesssim_q s \log \frac{em}{s} + s \left(C_q \frac{s R^{\frac{2}{q}}}{\varepsilon^2}\right)^{\frac{q}{2-q}} d.$$

This bound quantifies how the covering number of $\mathbf{T}_{0,q}(s,R)$ depends on the sparsity level $s$, the dimensionality $d$, and the spectral decay parameter $q$. □

**Refined Statement and Explanation.** We begin by defining a function $f(\mathbf{T}; \mathcal{X})$ for a tensor $\mathbf{T} \in \mathbb{R}^{d \times d \times m}$ and some data structure (or random tensor) $\mathcal{X}$. We consider a constrained supremum

$$\sup_{\varrho(\mathbf{T}) \le \nu, \mathbf{T} \in \mathsf{T}} f(\mathbf{T}; \mathcal{X}),$$

where $\varrho : \mathbb{R}^{d \times d \times m} \to \mathbb{R}^+$ is an *increasing* constraint function and $\mathsf{T}$ is any nonempty collection of tensors. Let $\nu > 0$ be a fixed threshold, and define the event

$$\mathcal{E} := \left\{\mathcal{X} : \exists \mathbf{T} \in \mathsf{T} \text{ such that } f(\mathbf{T}; \mathcal{X}) \ge 2\, g\big(\varrho(\mathbf{T})\big)\right\},$$

where $g : \mathbb{R} \to \mathbb{R}^+$ is strictly increasing. Our aim is to bound $\mathbb{P}(\mathcal{E})$, i.e. the probability that there is some tensor $\mathbf{T}$ with constraint $\varrho(\mathbf{T}) \le \nu$ for which $f(\mathbf{T}; \mathcal{X})$ exceeds $2\, g\big(\varrho(\mathbf{T})\big)$. This setup is quite general: for example, $f$ might be a residual or cost function in an empirical process framework, $\varrho(\mathbf{T})$ might measure the size or norm of $\mathbf{T}$, and $g$ could be a nondecreasing penalty or bound we wish to enforce.

**Peeling Bound.** Lemma 9 of Raskutti et al. (2011), reproduced below, gives a powerful "peeling"-type argument. It says that if we can control

$$\mathbb{P}\left[\sup_{\mathbf{T}: \varrho(\mathbf{T}) \le \nu} f(\mathbf{T}; \mathcal{X}) \ge g(\nu)\right] \le 2\, \exp\big(-c\, a_n\, g(\nu)\big)$$

for some constants $c > 0$ and $a_n > 0$, then one can derive a stronger tail bound for $\mathbb{P}(\mathcal{E})$. Formally:

**Lemma C.11** (Peeling, Lemma 9 of Raskutti et al. (2011)). *Suppose that for all $\nu \ge 0$, we have $g(\nu) \ge \mu$. Then there exists a constant $c > 0$ such that for all $\nu > 0$,*

$$\mathbb{P}\left[\sup_{\mathbf{T} \in \mathsf{T},\, \varrho(\mathbf{T}) \le \nu} f(\mathbf{T}; \mathcal{X}) \ge g(\nu)\right] \le 2\, \exp\big(-c\, a_n\, g(\nu)\big).$$

*Hence,*

$$\mathbb{P}(\mathcal{E}) = \mathbb{P}\Big(\exists \mathbf{T} \in \mathsf{T} : f(\mathbf{T}; \mathcal{X}) \ge 2\, g(\varrho(\mathbf{T}))\Big) \le \frac{2\, \exp\big(-4\, c\, a_n\, \mu\big)}{1 - \exp\big(-4\, c\, a_n\, \mu\big)}.$$

**Why "Peeling" is Useful.** This lemma effectively "peels" off the largest values of $\varrho(\mathbf{T})$ in layers and bounds the supremum in each layer by $g(\nu)$. One then aggregates or unions over these layers to control the probability that $f(\mathbf{T}; \mathcal{X})$ can exceed $2\, g(\varrho(\mathbf{T}))$ for *any* $\mathbf{T}$. Such arguments often appear in minimax or empirical process proofs, where one partitions the parameter space according to $\varrho(\mathbf{T})$.

**Additional Lemmas for Dual-Level Sparse Spectra.** In our dual-level sparse setting, we require more specialized versions of standard covering or chaining arguments. For instance, Lemma C.12 below extends Lemma 6 of Raskutti et al. (2011) to handle an $\ell_0$-type frequency-sparsity set with $\max(s)$ active slices and $\max(r)$ rank constraints:

$$\mathsf{T}_{0,0}(2s, 2r) = \big\{\underline{\mathbf{L}} : \#\{\text{nonzero freq. slices}\} \leq 2s, \; \text{rank}\big(M(\underline{\mathbf{L}})_{:,:,i}\big) \leq 2r\big\}.$$

We also define

$$\widetilde{\mathsf{S}}\big(\mathsf{T}_{0,0}(2s, 2r), \rho\big) = \Big[\mathsf{T}_{0,0}(2s, 2r)\Big] \cap \Big\{\underline{\mathbf{L}} : \|\underline{\mathbf{L}}\|_{\mathrm{F}} \leq \rho\Big\}.$$

**Lemma C.12.** *There exist positive constants $C_1, C_2 > 0$ such that for any $\rho > 0$,*

$$\sup_{\underline{\mathbf{L}} \in \widetilde{\mathsf{S}}\big(\mathsf{T}_{0,0}(2s,2r), \rho\big)} \Big|\big\langle \bar{\underline{\mathbf{E}}}, \underline{\mathbf{L}} \big\rangle\Big| \leq C_u \, \sigma \, \rho \, \sqrt{\frac{1}{n}\Big[s \log\big(\tfrac{em}{s}\big) + s\,r\,d\Big]},$$

*with probability at least $1 - C_1 \exp\big(-C_2\,[s \log(\tfrac{em}{s}) + s\,r\,d]\big)$, where $\bar{\underline{\mathbf{E}}}$ is the (scaled) noise tensor.*

*Sketch of proof.* This follows from Lemma 6 of Raskutti et al. (2011) if we replace the covering number for a naive $\ell_0$-ball by the more specific covering number of $\mathsf{T}_{0,0}(2s, 2r)$. The detailed estimate of that covering number is provided by part (a) of Lemma C.10. Essentially, one shows that among all frequency-sparse and rank-limited tensors of Frobenius norm up to $\rho$, the uniform covering can be done with cardinality roughly $\exp\big[s \log(\tfrac{em}{s}) + s\,r\,d \log(\tfrac{1}{\varepsilon})\big]$, and then translates that into a tail bound on $\sup_{\underline{\mathbf{L}}} \langle \bar{\underline{\mathbf{E}}}, \underline{\mathbf{L}} \rangle$.

**A Chaining Argument (Lemma C.13) For the Hard–Soft Setting.** When frequency sparsity is still "hard" but the rank constraint is "soft" via a Schatten-$q$ ball, we adapt the standard chaining approach to $\mathsf{T}_{0,q}(2s, 2R)$, i.e. the set of tensors with *up to* $2s$ active frequency slices and *each* slice having Schatten-$q$ norm up to $2R$. Let

$$\widetilde{\mathsf{S}}\big(\mathsf{T}_{0,q}(2s, 2R), \rho\big) = \mathsf{T}_{0,q}(2s, 2R) \cap \{\underline{\mathbf{L}} : \|\underline{\mathbf{L}}\|_{\mathrm{F}} \leq \rho\}.$$

We would like to show a tail bound of the form

$$\sup_{\underline{\mathbf{L}} \in \widetilde{\mathsf{S}}\big(\mathsf{T}_{0,q}(2s,2R), \rho\big)} \big|\langle \bar{\underline{\mathbf{E}}}, \underline{\mathbf{L}} \rangle\big| \leq \Big(\sqrt{\frac{s \log(\tfrac{em}{s})}{n}} + \sqrt{s\,R}\,\big(\tfrac{d}{n}\big)^{\frac{1}{2}-\frac{q}{4}}\Big)\rho$$

with high probability. The chaining lemma from Geer et al. (2000) is invoked, requiring a delicate construction of small $\delta$ and an integral bound on the covering number $N(t; \widetilde{\mathsf{S}}(\mathsf{T}_{0,q}(2s, 2R), \rho), \|\cdot\|_{\mathrm{F}})$.

**Lemma C.13** (Chaining bound)**.** *Assume*

$$\log\big(\tfrac{em}{s}\big) \leq C_2 \, n \, R^{\frac{2}{q}}. \tag{32}$$

*Then there exist constants $C_3, C_4 > 0$ such that for any*

$$\rho \geq c \left(\sqrt{\frac{s \log(\tfrac{em}{s})}{n}} + \sqrt{s\,R}\,\big(\tfrac{d}{n}\big)^{\frac{1}{2}-\frac{q}{4}}\right),$$

*we have*

$$\sup_{\underline{\mathbf{L}} \in \widetilde{\mathsf{S}}\big(\mathsf{T}_{0,q}(2s,2R), \rho\big)} \big|\langle \bar{\underline{\mathbf{E}}}, \underline{\mathbf{L}} \rangle\big| \leq \left(\sqrt{\frac{s \log(\tfrac{em}{s})}{n}} + \sqrt{s\,R}\,\big(\tfrac{d}{n}\big)^{\frac{1}{2}-\frac{q}{4}}\right)\rho$$

*with probability at least*

$$1 - C_3 \, \exp\left\{-C_4\,n\Big(\frac{s \log(\tfrac{em}{s})}{n} + s\,R\,\big(\tfrac{d}{n}\big)^{1-\frac{q}{2}}\Big)\right\}.$$

*Proof.* Following the approach of Lemma 7 in Raskutti et al. (2011), we aim to construct a constant $\delta$ that satisfies the conditions

$$\sqrt{n}\delta \geq C_1\rho, \tag{33}$$

and

$$C_2\sqrt{n}\delta \geq \int_{\frac{\delta}{16}}^{\rho} \sqrt{\log N\left(t; \widetilde{\mathbf{S}}(\mathbf{T}_{0,q}(2s, 2R), \rho), \|\cdot\|_{\mathrm{F}}\right)} \mathrm{d}t \coloneqq J(\rho, \delta), \tag{34}$$

where $N\left(t; \widetilde{\mathbf{S}}(\mathbf{T}_{0,q}(2s, 2R), \rho), \|\cdot\|_{\mathrm{F}}\right)$ denotes the $t$-covering number of $\widetilde{\mathbf{S}}(\mathbf{T}_{0,q}(2s, 2R), \rho)$.

Applying Lemma 3.2 in Geer et al. (2000), we obtain that for $\|\bar{\mathbf{E}}\|_{\mathrm{F}}^2 \leq 16$,

$$\mathbb{P}\left[\sup_{\mathbf{L} \in \widetilde{\mathbf{S}}(\mathbf{T}_{0,q}(2s, 2R), \rho)} |\langle \bar{\mathbf{E}}, \mathbf{L}\rangle| \geq \delta, \quad \|\bar{\mathbf{E}}\|_{\mathrm{F}}^2 \leq 16\right] \leq C_3 \exp(-C_4 \frac{n\delta^2}{\rho^2}).$$

Since each entry of $\bar{\mathbf{E}}$ is drawn from $N(0, \frac{\sigma^2}{n})$, applying standard tail bounds for $\chi^2$ random variables (Raskutti et al., 2011) yields

$$\mathbb{P}[\|\bar{\mathbf{E}}\|_{\mathrm{F}}^2 \geq 16] \leq C_5 \exp(-C_6 n).$$

Consequently, we obtain the bound

$$\mathbb{P}\left[\sup_{\mathbf{L} \in \widetilde{\mathbf{S}}(\mathbf{T}_{0,q}(2s, 2R), \rho)} |\langle \bar{\mathbf{E}}, \mathbf{L}\rangle| \geq \delta\right] \leq C_3 \exp(-C_4 \frac{n\delta^2}{\rho^2}) + C_5 \exp(-C_6 n).$$

Next, we construct $\delta$ to satisfy conditions Eq. (33) and Eq. (34). Define

$$\delta = \rho\left(\sqrt{\frac{s \log \frac{em}{s}}{n}} + \omega\right),$$

where $\omega > 0$ is a constant to be determined later. This choice immediately satisfies Eq. (33). For Eq. (34), using condition Eq. (32), we set

$$\rho = \Omega\left(\sqrt{\frac{s \log \frac{em}{s}}{n}} + \sqrt{s}R(\frac{d}{n})^{\frac{1}{2}-\frac{q}{4}}\right) \wedge \sqrt{s}R^{\frac{1}{q}}.$$

It follows that $(\frac{\delta}{16}, \rho)$ lies within the valid range of $\varepsilon$ in Lemma C.10. By applying part (b) of Lemma C.10, we obtain

$$J(\rho, \delta) = \int_{\frac{\delta}{16}}^{\rho} \sqrt{\log N\left(t; \widetilde{\mathbf{S}}(\mathbf{T}_{0,q}(2s, 2R), \rho)\right)} \, \mathrm{d}t$$

$$\leq \int_0^{\rho} \sqrt{2s \log \frac{em}{s} + 2s\left(\frac{s}{t^2}R^{\frac{2}{q}}\right)^{\frac{q}{2-q}} d} \, \mathrm{d}t$$

$$\leq \sqrt{2s \log \frac{em}{s}}\rho + \sqrt{2}(sR)^{\frac{1}{2-q}}\sqrt{d}\rho^{1-\frac{q}{2-q}}.$$

Dividing both sides by $\sqrt{n}\delta$ gives

$$\frac{J(\rho, \delta)}{\sqrt{n}\delta} \leq \frac{\sqrt{2s \log \frac{em}{s}}\rho + \sqrt{2}(sR)^{\frac{1}{2-q}}\sqrt{d}\rho^{1-\frac{q}{2-q}}}{\rho\sqrt{s \log \frac{em}{s}} + \rho\sqrt{n}\omega}.$$

Setting $\omega = \sqrt{2}(sR)^{\frac{1}{2-q}}\sqrt{\frac{d}{n}}\rho^{1-\frac{q}{2-q}}$ ensures that

$$\frac{J(\rho, \delta)}{\sqrt{n}\delta} \leq \sqrt{2}.$$

Thus, condition Eq. (34) holds.

Finally, we conclude that

$$\delta = \rho \left( \sqrt{\frac{s \log \frac{em}{s}}{n}} + \sqrt{sR}(\frac{d}{n})^{\frac{1}{2} - \frac{q}{4}} \right).$$

Substituting into our probability bound, we obtain

$$\mathbb{P} \left[ \sup_{\underline{\mathbf{L}} \in \widetilde{\mathbf{S}}(\mathbf{T}_{0,q}(2s,2R),\rho)} |\langle \bar{\underline{\mathbf{E}}}, \underline{\mathbf{L}} \rangle| \geq \left( \sqrt{\frac{s \log \frac{em}{s}}{n}} + \sqrt{sR}(\frac{d}{n})^{\frac{1}{2} - \frac{q}{4}} \right) \rho \right]$$

$$\leq C_3 \exp \left( -C_4 n \left( \frac{s \log \frac{em}{s}}{n} + sR \left( \frac{d}{n} \right)^{1 - \frac{q}{2}} \right) \right),$$

which completes the proof of Lemma C.13. $\qquad \square$

### C.3.2 PROOF OF THEOREM:UPPERBOUNDSZEROS

We analyze the constrained MLE estimator in Eq. (16). Since for any $q \in [0,1]$, the estimator satisfies

$$\|\bar{\underline{\mathbf{Y}}} - \hat{\underline{\mathbf{L}}}_q\|_{\mathrm{F}}^2 \leq \|\bar{\underline{\mathbf{Y}}} - \underline{\mathbf{L}}^\star\|_{\mathrm{F}}^2,$$

rearranging terms gives

$$\|\hat{\underline{\mathbf{L}}}_q - \underline{\mathbf{L}}^\star\|_{\mathrm{F}}^2 \leq 2|\langle \bar{\underline{\mathbf{E}}}, \hat{\underline{\mathbf{L}}}_q - \underline{\mathbf{L}}^\star \rangle|. \qquad (35)$$

**Proof of Case (a)**

*Proof.* For case (a), since both $\hat{\underline{\mathbf{L}}}_0$ and $\underline{\mathbf{L}}^\star$ belong to $\mathbf{T}_{0,0}(s,r)$, their difference satisfies

$$\hat{\underline{\mathbf{L}}}_0 - \underline{\mathbf{L}}^\star \in \mathbf{T}_{0,0}(2s,2r).$$

Applying Lemma C.12, for any $\rho > 0$, we obtain

$$\sup_{\underline{\mathbf{L}} \in \widetilde{\mathbf{S}}(\mathbf{T}_{0,0}(2s,2r),\rho)} |\langle \bar{\underline{\mathbf{E}}}, \underline{\mathbf{L}} \rangle| \leq C_u \sigma \rho \sqrt{\frac{1}{n}(s \log \frac{em}{s} + srd)}$$

with probability at least $1 - C_1 \exp\{-C_2(s \log \frac{em}{s} + srd)\}$.

Next, consider the event $\mathcal{E}$ where there exists some $\underline{\mathbf{L}} \in \mathbf{T}_{0,0}(2s,2r)$ such that

$$|\langle \bar{\underline{\mathbf{E}}}, \underline{\mathbf{L}} \rangle| \geq C_u \sigma \|\underline{\mathbf{L}}\|_{\mathrm{F}} \sqrt{\frac{1}{n}(s \log \frac{em}{s} + srd)}. \qquad (36)$$

By Lemma C.11, the probability of this event satisfies

$$\mathbb{P}[\mathcal{E}] \leq \frac{2 \exp(-C_3(s \log \frac{em}{s} + srd))}{1 - \exp(-C_3(s \log \frac{em}{s} + srd))}.$$

This follows by applying Lemma C.11 with function $f(\mathbf{T}; \mathcal{X}) = \langle \bar{\underline{\mathbf{E}}}, \underline{\mathbf{L}} \rangle$, set $\mathbf{T} = \mathbf{T}_{0,0}(2s,2r)$, sequence $a_n = n/\sigma^2$, function $\varrho(\mathbf{T}) = \|\mathbf{T}\|_{\mathrm{F}}$, and threshold function $g(\nu) = C_u \sigma \nu \sqrt{\frac{1}{n}(s \log \frac{em}{s} + srd)}$. For any $\nu \geq \sigma \sqrt{\frac{1}{n}(s \log \frac{em}{s} + srd)}$, we ensure that $g(\nu) \geq \frac{\sigma^2}{n}(s \log \frac{em}{s} + srd)$, allowing us to apply the lemma.

Combining Eq. (35) and Eq. (36) yields

$$\|\hat{\underline{\mathbf{L}}}_0 - \underline{\mathbf{L}}^\star\|_{\mathrm{F}}^2 \leq C_u \frac{\sigma^2}{n}(s \log \frac{em}{s} + srd).$$

This bound holds with probability at least $1 - C_1 \exp\{-C_2(s \log \frac{em}{s} + srd)\}$, completing the proof of Eq. (17). $\qquad \square$

**Proof of case(b)**

*Proof.* For case (b), since $\hat{\underline{\mathbf{L}}}_q, \underline{\mathbf{L}}^\star \in \mathbf{T}_{0,q}(s, R)$, it follows that their difference satisfies $\hat{\underline{\mathbf{L}}}_q - \underline{\mathbf{L}}^\star \in \mathbf{T}_{0,q}(2s, 2R)$.

We define the event $\mathcal{E}$ as the existence of some $\underline{\mathbf{L}} \in \mathbf{T}_{0,q}(s, R)$ such that, by Lemma C.13, the following holds with probability at least $1 - C_3 \exp\{-C_4 n(\frac{s \log \frac{em}{s}}{n} + sR(\frac{d}{n})^{1-\frac{q}{2}})\}$:

$$|\langle \bar{\underline{\mathbf{E}}}, \underline{\mathbf{L}} \rangle| \geq C_u \|\underline{\mathbf{L}}\|_{\mathrm{F}} \left( \sqrt{\frac{s \log \frac{em}{s}}{n}} + \sqrt{sR}(\frac{d}{n})^{\frac{1}{2}-\frac{q}{4}} \right).$$

Applying Lemma C.11, we further obtain the probability bound:

$$\mathbb{P}[\mathcal{E}] \leq \frac{2 \exp(-C_3 n(\frac{s \log \frac{em}{s}}{n} + sR(\frac{d}{n})^{1-\frac{q}{2}}))}{1 - \exp(-C_3 n(\frac{s \log \frac{em}{s}}{n} + sR(\frac{d}{n})^{1-\frac{q}{2}}))}. \tag{37}$$

This follows from Lemma C.11 by setting:

$$f(\underline{\mathbf{T}}; \mathcal{X}) = \langle \bar{\underline{\mathbf{E}}}, \underline{\mathbf{L}} \rangle, \quad \mathbf{T} = \mathbf{T}_{0,q}(2s, 2R), \quad a_n = n, \quad \varrho(\underline{\mathbf{T}}) = \|\underline{\mathbf{T}}\|_{\mathrm{F}}, \quad g(\nu) = \nu \left( \sqrt{\frac{s \log \frac{em}{s}}{n}} + \sqrt{sR}(\frac{d}{n})^{\frac{1}{2}-\frac{q}{4}} \right).$$

Combining Eq. (35) with these results, we conclude that:

$$\|\hat{\underline{\mathbf{L}}} - \underline{\mathbf{L}}^\star\|_{\mathrm{F}}^2 \leq C_u \left( \sqrt{\frac{s \log \frac{em}{s}}{n}} + \sqrt{sR}(\frac{d}{n})^{\frac{1}{2}-\frac{q}{4}} \right),$$

which holds with probability at least

$$1 - C_3 \exp\{-C_4 n(\frac{s \log \frac{em}{s}}{n} + sR(\frac{\sigma^2(1-v)}{4n}d)^{1-\frac{q}{2}})\},$$

thus completing the proof of Eq. (18). $\qquad\square$

### C.4 Proof of Theorem C.4

#### C.4.1 Lower bound for $\ell_p(S_q)$

*Proof.* We aim to prove a minimax lower bound under the dual-level sparse structure imposed by an $\ell_p(S_q)$ quasi-norm. Our proof strategy considers two key parameters that govern dual-level sparsity:

- *Frequency sparsity* $1 \leq s \leq m$, controlling how many frequency components can be nonzero.

- *Within-frequency low-rankness* $1 \leq r \leq d$, limiting the rank of each active frequency component.

*Subspace construction and parameter setting.* We begin with the subspace $\mathbf{T}_{0,0}(s, r)$ of tensors, as introduced in earlier sections, where both frequency indices and within-frequency ranks are hard-constrained. For any tensor in this subspace, suppose the absolute value of each nonzero entry is set to $\delta > 0$. This $\delta$ is chosen so that the $(p, q)$-quasi-norm constraint is satisfied, i.e.,

$$s \cdot \left( r \cdot (\delta \sqrt{d})^q \right)^{\frac{p}{q}} = R.$$

Solving for $\delta$ yields

$$\delta = R^{\frac{1}{p}} s^{-\frac{1}{p}} r^{-\frac{1}{q}} d^{-\frac{1}{2}}.$$

Thus, $\delta$ encodes how large each nonzero entry must be so that the tensor simultaneously satisfies frequency-sparsity and low-rankness in a consistent manner for the $\ell_p(S_q)$-norm. Notice that $\delta$ scales inversely with $s$, $r$, and $\sqrt{d}$, reflecting the interplay among frequency selection, rank constraints, and the Frobenius norm.

*Applying generalized Fano's inequality.* Let us express the probability that any estimator $\hat{\underline{L}}$ incurs significant estimation error. Using a generalized Fano argument, we obtain

$$\inf_{\hat{\underline{L}}} \sup_{\underline{L}^* \in \mathsf{T}_{p,q}(R)} \mathbb{P}\Big[\|\hat{\underline{L}} - \underline{L}^*\|_F^2 \geq \tfrac{1}{16} R^{\frac{2}{p}} s^{1-\frac{2}{p}} r^{1-\frac{2}{q}}\Big] \geq 1 - \frac{\frac{n}{\sigma^2} R^{\frac{2}{p}} s^{1-\frac{2}{p}} r^{1-\frac{2}{q}} + \log 2}{\frac{1}{4}\left(s\,r\,d + s\,\log(\frac{m}{s})\right)}. \quad (38)$$

The denominator $s\,r\,d + s\,\log(\frac{m}{s})$ captures (1) the cost of identifying $s$ nonzero frequency slices each of rank at most $r$, plus (2) the combinatorial complexity $s\,\log(\frac{m}{s})$ for subset selection among $m$ frequencies.

*Reformulating via linear programming.* To analyze Eq. (38) precisely, we adopt a linear programming approach similar to Li et al. (2024) for the $\ell_u(\ell_q)$-ball. Let $y = \log s$ and $x = \log r$. Then

$$R^{\frac{2}{p}} s^{1-\frac{2}{p}} r^{1-\frac{2}{q}} = \exp\Big[\log R^{\frac{2}{p}} + \big(1 - \tfrac{2}{p}\big)\log s + \big(1 - \tfrac{2}{q}\big)\log r\Big].$$

Hence, maximizing $R^{\frac{2}{p}} s^{1-\frac{2}{p}} r^{1-\frac{2}{q}}$ is equivalent to maximizing

$$z = \Big(1 - \tfrac{2}{p}\Big) y + \Big(1 - \tfrac{2}{q}\Big) x,$$

subject to constraints bounding $x$ and $y$ (i.e. $0 \leq x \leq \log d$ and $0 \leq y \leq \log m$), plus an additional constraint from balancing numerator and denominator in Eq. (38). Concretely:

$$\begin{cases} 0 \leq x \leq \log d, \quad 0 \leq y \leq \log m, \\ y \geq \min\Big\{-\tfrac{p}{q} x + \log R + \tfrac{p}{2}\big(\log n - \log(\sigma^2) - \log d\big), -\big(\tfrac{p}{q} - \tfrac{p}{2}\big)x + \log R + \tfrac{p}{2}\big(\log n - \log(\sigma^2) - \log(\log m)\big)\Big\}. \end{cases}$$

The first two lines capture $s \leq m$, $r \leq d$, while the last line encodes how $\frac{n}{\sigma^2} R^{\frac{2}{p}} s^{1-\frac{2}{p}} r^{1-\frac{2}{q}}$ compares with $s\,r\,d + s\,\log(\frac{m}{s})$ to ensure a valid Fano-type bound.

*Analyzing slopes and boundary points.* For convenience, define:

$$x_1 = \log R + \tfrac{p}{2}\log\Big(\tfrac{n}{\sigma^2 \log m}\Big), \quad x_2 = \log R + \tfrac{p}{2}\log\Big(\tfrac{n}{\sigma^2 d}\Big),$$

$$y_1 = \tfrac{q}{p}\log R + \tfrac{q}{2}\log\Big(\tfrac{n}{\sigma^2 \log m}\Big), \quad y_2 = \tfrac{q}{p}\log R + \tfrac{q}{2}\log\Big(\tfrac{n}{\sigma^2 d}\Big).$$

We then consider the lines:

- *Line A:* $y = -\tfrac{p}{q} x + \log R + \tfrac{p}{2}\big(\log n - \log(\sigma^2) - \log d\big)$, with slope $-\tfrac{p}{q}$ in the $(x, y)$-plane.

- *Line B:* $y = -\big(\tfrac{p}{q} - \tfrac{p}{2}\big) x + \log R + \tfrac{p}{2}\big(\log n - \log(\sigma^2) - \log(\log m)\big)$, whose slope is $\tfrac{p}{2} - \tfrac{p}{q}$.

- *Objective slope:* The slope of $z = (1 - \tfrac{2}{p}) y + (1 - \tfrac{2}{q}) x$ is $\tfrac{p(q-2)}{q(p-2)}$ in the $(x, y)$-plane.

A standard slope comparison yields these observations:

1. If $p > q$, the slope of $z$ is larger than slopes of Lines A and B, so the maximum of $z$ is attained at boundary points like $(0, y_2)$, $(0, y_1)$, or $(x_1, 0)$.

2. If $p \leq q$, the slope of $z$ is smaller than the slope of A but larger than that of B, so maxima can occur at $(x_1, 0)$, $(x_2, 0)$, or intersections of A and B, depending on $m$ vs. $d$.

Evaluating $z$ at these boundary points, one then obtains the resulting minimax lower bounds:

$$\begin{cases} R\Big(\tfrac{n}{d\sigma^2}\Big)^{\frac{p-2}{2}} + R\Big(\tfrac{n}{\sigma^2 \log m}\Big)^{\frac{p-2}{2}}, & \text{if } p > q, \\[2mm] R^{\frac{q}{p}}\Big(\tfrac{n}{d\sigma^2}\Big)^{\frac{q-2}{2}} + R\Big(\tfrac{n}{\sigma^2 \log m}\Big)^{\frac{p-2}{2}}, & \text{if } p \leq q, m \geq d^2, \\[2mm] R^{\frac{q}{p}}\Big(\tfrac{n}{d\sigma^2}\Big)^{\frac{q-2}{2}}, & \text{if } p \leq q, m \leq d^2. \end{cases}$$

These match precisely the piecewise expressions for the lower bound in the $\ell_p(S_q)$ setting. Hence, combining with the initial Fano-based argument Eq. (38) concludes the minimax lower bound proof.

$\square$

### C.4.2 Covering number of $\mathbb{B}_{\ell_p(S_q)}(R)$

Before deriving the upper bounds for $\ell_p(S_q)$, we first need to derive the covering number of $\mathbb{B}_{\ell_p(S_q)}(R)$ equipped with the $\ell_p(S_q)$-norm. To this end, we generalize Schütt's theorem for vector-valued sequence spaces (Edmunds & Netrusov, 2014). The analysis relies on entropy numbers and their relationships under different parameter ranges. We introduce several key lemmas and derive the upper bound for $e_k$.

**Lemma C.14** (Schütt's Theorem for Vector-valued Sequence Spaces (Edmunds & Netrusov, 2014)). *Let $X$ and $Y$ be $r$-normed quasi-Banach spaces, and let $0 < q < r \leq \infty$. The unit ball $\mathbb{B}_{\ell_q^m(X)}$ is defined as:*

$$\mathbb{B}_{\ell_q^m(X)} = v_1 \mathbb{B}_X \times v_2 \mathbb{B}_X \times \cdots \times v_m \mathbb{B}_X,$$

*where $\mathbb{B}_X$ is the unit ball with $X$-norm, and $v \in \mathbb{B}_q$. For $k, k_0 \in \mathbb{N}$ such that $k_0 \leq k$, let:*

$$D(k_0, k) = \max_{l \in \mathbb{N}, k_0 \leq l \leq k} \left( \frac{l}{k} \right)^{\frac{1}{q} - \frac{1}{r}} e_l(id : X \to Y),$$

$$A(k, m) = \max \left\{ \|id : X \to Y\| \left( \frac{\log(em/k)}{k} \right)^{\frac{1}{q} - \frac{1}{r}}, D(1, k) \right\},$$

*where $\|id : X \to Y\|$ denotes the operator norm, and $e_l(id : X \to Y)$ denotes the $l$-th entropy number. For $k \geq \log_2(m)$, the entropy numbers satisfy:*

- *If $k \leq m$, then*

$$e_k \left( id : \ell_q^m(X) \to \ell_r^m(Y) \right) \simeq A(k, m).$$

- *If $k \geq m$, then there exist constants $C_1, C_2 > 0$ such that:*

$$D(C_1 k/m, k) \leq e_k \left( id : \ell_q^m(X) \to \ell_r^m(Y) \right) \leq D(C_2 k/m, k).$$

Let $q = p$, $r = 2$, $X = S_q^d$, or $Y = \ell_2^d$ for our problem, so that $\|id : X \to Y\| = 1$ and $e_l(id : X \to Y)$ is given by Eq. (15a), Eq. (15b) and Eq. (15c). Using the results of this lemma, we define the function $\phi(l)$ to analyze the behavior of entropy numbers for $\ell_p(S_q)$-balls. Specifically, we let:

$$\phi(l) := \left( \frac{l}{k} \right)^{\frac{1}{q} - \frac{1}{r}} e_l(id : X \to Y) = \begin{cases} \left( \frac{l}{k} \right)^{\frac{1}{p} - \frac{1}{2}} & 1 \leq l \leq d, \\ \left( \frac{l}{k} \right)^{\frac{1}{p} - \frac{1}{2}} \left( \frac{d}{l} \right)^{\frac{1}{q} - \frac{1}{2}} & d \leq l \leq d^2, \\ \left( \frac{l}{k} \right)^{\frac{1}{p} - \frac{1}{2}} 2^{-\frac{l}{d^2}} d^{\frac{1}{2} - \frac{1}{q}} & l \geq d^2. \end{cases}$$

The monotonicity behavior of $\phi(l)$ across different ranges of $l$ is summarized in Table 2.

Table 2: Monotonicity of $\phi(l)$ for $p \leq q$ and $p > q$ in different ranges of $l$ when $p \leq 1$

| Range of $l$ | Expression for $\phi(l)$ | Monotonicity (if $p \leq q$) | Monotonicity (if $p > q$) | Critical Point |
|---|---|---|---|---|
| $1 \leq l \leq d$ | $\left( \frac{l}{k} \right)^{\frac{1}{p} - \frac{1}{2}}$ | Increasing | | None |
| $d \leq l \leq d^2$ | $\left( \frac{l}{k} \right)^{\frac{1}{p} - \frac{1}{2}} \left( \frac{d}{l} \right)^{\frac{1}{q} - \frac{1}{2}}$ | Increasing | Decreasing | None |
| $l \geq d^2$ | $\left( \frac{l}{k} \right)^{\frac{1}{p} - \frac{1}{2}} 2^{-\frac{l}{d^2}} d^{\frac{1}{2} - \frac{1}{q}}$ | Increasing then Decreasing with maximum at $l^* = \frac{\left( \frac{1}{p} - \frac{1}{2} \right) d^2}{\ln 2}$ | | $l^* > d^2$ |
| | | Decreasing | | $l^* \leq d^2$ |

**Lemma C.15** (Entropy Number for $\ell_p(S_q) \hookrightarrow \ell_2(S_2)$)**.** *For* $k \geq \max\{\log m, d\}$, *the entropy numbers* $e_k$ *for* $\mathbb{B}_{\ell_p(S_q)}(R)$ *satisfy:*

$$
e_k \simeq_{p,q}
\begin{cases}
\left(\frac{d}{k}\right)^{\frac{1}{q}-\frac{1}{2}} & p \leq q, \ m \leq d^2 \\[2mm]
\max\left\{ \left(\frac{d}{k}\right)^{\frac{1}{q}-\frac{1}{2}}, \left(\frac{\log(em)}{k}\right)^{\frac{1}{p}-\frac{1}{2}} \right\} & p \leq q, \ m \geq d^2 \\[2mm]
\left(\frac{\max\{d, \log(em)\}}{k}\right)^{\frac{1}{p}-\frac{1}{2}} & q \leq p.
\end{cases}
\tag{39}
$$

*Proof of Lemma C.15.* Let us prove this lemma by carefully analyzing different cases based on the relationships between $p$, $q$, $m$, and $d$. Our analysis will heavily rely on the behavior of $\phi(l)$ as shown in Table 2 and the application of Lemma C.14.

**Case (a):** For $p \leq q, m \leq d^2$, we consider the range $d \leq k \leq d^2$ and divide our analysis into two subcases based on the relationship between $k$ and $m$.

(i) First, consider $k \leq m$: According to Lemma C.14, we have:

$$
e_k \simeq A(k, m) = \max\left\{ \left(\frac{\log(em/k)}{k}\right)^{\frac{1}{p}-\frac{1}{2}}, D(1, k) \right\}.
$$

To evaluate $D(1, k)$, we need to analyze $\phi(l)$ in different ranges:

- For $1 \leq l \leq d$:

$$
\phi(l) = \left(\frac{l}{k}\right)^{\frac{1}{p}-\frac{1}{2}}.
$$

  This function is strictly increasing as $\frac{1}{p} - \frac{1}{2} > 0$ for $p \leq 1$. The maximum in this range occurs at $l = d$.

- For $d \leq l \leq d^2$:

$$
\phi(l) = \left(\frac{l}{k}\right)^{\frac{1}{p}-\frac{1}{2}} \left(\frac{d}{l}\right)^{\frac{1}{q}-\frac{1}{2}}.
$$

  Since $p \leq q$, we have $\frac{1}{p} - \frac{1}{q} \geq 0$, making this function increasing. The maximum in this range occurs at $l = k$ since $k \leq d^2$.

Combining these results, we find:

$$
D(1, k) = \max_{1 \leq l \leq k} \phi(l) = \left(\frac{d}{k}\right)^{\frac{1}{q}-\frac{1}{2}}.
$$

Now, since $m \leq d^2$, we can show:

$$
\left(\frac{\log(em/k)}{k}\right)^{\frac{1}{p}-\frac{1}{2}} \lesssim \left(\frac{d}{k}\right)^{\frac{1}{q}-\frac{1}{2}}.
$$

Therefore:

$$
e_k \simeq \left(\frac{d}{k}\right)^{\frac{1}{q}-\frac{1}{2}}.
$$

(ii) Next, consider $m \leq k \leq d^2$: By Lemma C.14, when $k \geq m$, we have:

$$
D(C_1 k/m, k) \leq e_k \leq D(C_2 k/m, k).
$$

For each $C \in \{C_1, C_2\}$, we need to evaluate:

$$
D(Ck/m, k) = \max_{Ck/m \leq l \leq k} \phi(l).
$$

Let's analyze $\phi(l)$ in the relevant ranges:

- When $Ck/m \le l \le d$:

$$\phi(l) = \left(\frac{l}{k}\right)^{\frac{1}{p}-\frac{1}{2}}.$$

This is increasing, reaching its maximum at $l = d$ if $d$ is in this range.

- When $d \le l \le k \le d^2$:

$$\phi(l) = \left(\frac{l}{k}\right)^{\frac{1}{p}-\frac{1}{2}} \left(\frac{d}{l}\right)^{\frac{1}{q}-\frac{1}{2}}.$$

Since $p \le q$, this is increasing and reaches its maximum at $l = k$.

The monotonicity of $\phi(l)$ implies that both bounds achieve their maximum at $l = k$:

$$D(Ck/m, k) = \left(\frac{k}{k}\right)^{\frac{1}{p}-\frac{1}{2}} \left(\frac{d}{k}\right)^{\frac{1}{q}-\frac{1}{2}} = \left(\frac{d}{k}\right)^{\frac{1}{q}-\frac{1}{2}}.$$

Therefore:

$$D(C_1 k/m, k) = D(C_2 k/m, k) = \left(\frac{d}{k}\right)^{\frac{1}{q}-\frac{1}{2}}.$$

This gives us:

$$e_k \simeq \left(\frac{d}{k}\right)^{\frac{1}{q}-\frac{1}{2}}.$$

**Case (b)**: For $p \le q, m \ge d^2$, when $d \le k \le m$, Lemma C.14 gives:

$$e_k \simeq \max\left\{\left(\frac{\log(em/k)}{k}\right)^{\frac{1}{p}-\frac{1}{2}}, D(1, k)\right\}.$$

We analyze this in two subcases:

(i) For $d \le k \le d^2$: Similar to Case (a), analyzing $\phi(l)$ in three ranges:

- $1 \le l \le d$: $\phi(l) = \left(\frac{l}{k}\right)^{\frac{1}{p}-\frac{1}{2}}$ is increasing;

- $d \le l \le d^2$: $\phi(l) = \left(\frac{l}{k}\right)^{\frac{1}{p}-\frac{1}{2}} \left(\frac{d}{l}\right)^{\frac{1}{q}-\frac{1}{2}}$ is increasing since $p \le q$;

- $l \ge d^2$: $\phi(l)$ is decreasing from $l = d^2$.

Therefore:

$$D(1, k) = \left(\frac{d}{k}\right)^{\frac{1}{q}-\frac{1}{2}}.$$

(ii) For $d^2 \le k \le m$: In this range:

$$D(1, k) = \max_{1 \le l \le k} \phi(l) = \left(\frac{d^2}{k}\right)^{\frac{1}{p}-\frac{1}{2}} d^{\frac{1}{2}-\frac{1}{q}}.$$

When $k \ge d^2$, we can show:

$$\left(\frac{d}{k}\right)^{\frac{1}{q}-\frac{1}{2}} \ge \left(\frac{d^2}{k}\right)^{\frac{1}{p}-\frac{1}{2}} d^{\frac{1}{2}-\frac{1}{q}} = \left(\frac{d}{k}\right)^{\frac{1}{p}-\frac{1}{2}} d^{\frac{1}{p}-\frac{1}{q}}.$$

Therefore, for Case (b):

$$e_k \simeq \max\left\{\left(\frac{\log(em/k)}{k}\right)^{\frac{1}{p}-\frac{1}{2}}, \left(\frac{d}{k}\right)^{\frac{1}{q}-\frac{1}{2}}\right\}.$$

**Case (c)**: For $p > q$, we divide this case into two subcases based on the range of $k$.

(i) For $\max\{d, \log m\} \le k \le md$: When $p > q$, we know $\phi(l)$ is decreasing in $[d, d^2]$. Analyzing $\phi(l)$:

- For $1 \le l \le d$: Maximum occurs at $l = \max\{d, \log(em)\}$;

- For $d \le l \le d^2$: Monotonically decreasing;

- For $l \ge d^2$: Strictly decreasing;

Therefore:

$$D(1, k) = \left( \frac{\max\{d, \log(em)\}}{k} \right)^{\frac{1}{p} - \frac{1}{2}}.$$

(ii) For $md \le k \le md^2$: By similar analysis and considering $p > q$:

$$e_k \simeq m^{\frac{1}{q} - \frac{1}{p}} \left( \frac{d}{k} \right)^{\frac{1}{q} - \frac{1}{2}}.$$

Under the assumption $k \ge m \cdot \max\{d, \log(em)\}$:

$$\left( \frac{\max\{d, \log(em)\}}{k} \right)^{\frac{1}{p} - \frac{1}{2}} \ge m^{\frac{1}{q} - \frac{1}{p}} \left( \frac{d}{k} \right)^{\frac{1}{q} - \frac{1}{2}} \iff \left( \frac{\max\{d, \log(em)\}}{d} \right)^{\frac{1}{p} - \frac{1}{2}} \ge m^{\frac{1}{q} - \frac{1}{p}}.$$

This inequality holds under our assumptions.

Combining all three cases yields the result in Eq. (39). $\qquad\square$

### C.4.3 PROOF OF UPPER BOUNDS FOR $\ell_p(S_q)$

Recall that we define

$$\widetilde{\mathsf{S}}(\mathbf{T}_{p,q}(R), \rho) = \left\{ \underline{\mathbf{L}} \in \mathbb{R}^{d \times m} : \|\underline{\mathbf{L}}\|_F \le \rho \right\} \cap \mathbf{T}_{p,q}(R).$$

*Proof.* We prove the theorem by separating into three distinct cases based on the relationships among $p$, $q$, and $m$. In each case, we construct appropriate constants and verify the necessary conditions.

**Case 1:** $p \le q$ **and** $m \ge d^2$. We begin by adding a radius factor $R^{\frac{1}{p}}$ to the entropy number bounds in Eq. (39), yielding

$$\epsilon \le R^{\frac{1}{p}} \left( \frac{d}{k} \right)^{\frac{1}{q} - \frac{1}{2}} + R^{\frac{1}{p}} \left( \frac{\log(em)}{k} \right)^{\frac{1}{p} - \frac{1}{2}}.$$

Solving for $k$ provides an upper bound on the covering number, giving

$$\log N\left(\epsilon \,;\, \widetilde{\mathsf{S}}(\mathbf{T}_{p,q}(R), r)\right) \le d \left( \epsilon^{-1} R^{\frac{1}{p}} \right)^{\frac{2q}{2-q}} + \log(em) \left( \epsilon^{-1} R^{\frac{1}{p}} \right)^{\frac{2p}{2-p}}. \tag{40}$$

To apply Lemma 3.2 from (Geer et al., 2000), we need to construct constants $(\delta, \rho)$ satisfying two key conditions:

$$\sqrt{n}\,\delta \ge C_1\,\rho \qquad\qquad \text{(Condition 1)} \tag{41}$$
$$C_2\,\sqrt{n}\,\delta \ge J(\rho, \delta) \qquad\qquad \text{(Condition 2)} \tag{42}$$

where

$$J(\rho, \delta) = \int_{\frac{\delta}{16}}^{\rho} \sqrt{\log N\left(t \,;\, \widetilde{\mathsf{S}}(\mathbf{T}_{p,q}(R), \rho)\right)}\, \mathrm{d}t \le \int_0^{\rho} \sqrt{d \left(t^{-1} R^{\frac{1}{p}}\right)^{\frac{2q}{2-q}} + \log(em) \left(t^{-1} R^{\frac{1}{p}}\right)^{\frac{2p}{2-p}}}\, \mathrm{d}t.$$

A direct calculation yields:

$$J(\rho, \delta) \le \sqrt{d}\, R^{\frac{q}{p(2-q)}}\, \rho^{1 - \frac{q}{2-q}} + \sqrt{\log(em)}\, R^{\frac{1}{2-p}}\, \rho^{1 - \frac{p}{2-p}}.$$

**Choice of constants $\rho, \delta$.**   Let

$$\rho = \Omega\Big(R^{\frac{q}{2p}}\big(\tfrac{n}{d}\big)^{\frac{q-2}{4}} + R^{\frac{1}{2}}\big(\tfrac{n}{\log m}\big)^{\frac{p-2}{4}}\Big) \wedge R^{\frac{1}{p}}, \qquad\qquad (\rho \text{ definition})$$

$$\delta = C\,\rho\Big(R^{\frac{q}{2p}}\big(\tfrac{n}{d}\big)^{\frac{q-2}{4}} + R^{\frac{1}{2}}\big(\tfrac{n}{\log m}\big)^{\frac{p-2}{4}}\Big). \qquad\qquad (\delta \text{ definition})$$

*Verifying Condition Eq.* (41).

$$\frac{\sqrt{n}\,\delta}{\rho} \geq C\left[R^{\frac{q}{p}}\,n^{\frac{q}{2}}\big(\tfrac{1}{d}\big)^{\frac{q-2}{2}} + R\,n^{\frac{p}{2}}\big(\tfrac{1}{\log m}\big)^{\frac{p-2}{2}}\right] \geq C_1,$$

where the second inequality follows from Eq. (19).

*Verifying Condition Eq.* (42). Given that $\rho$ is chosen as in Eq. ($\rho$ definition), an analogous ratio bound shows

$$\frac{J(\rho, \delta)}{\sqrt{n}\,\delta} = \frac{\sqrt{d}\,R^{\frac{q}{p(2-q)}}\rho^{-\frac{q}{2-q}} + \sqrt{\log(m)}\,R^{\frac{1}{2-p}}\rho^{-\frac{p}{2-p}}}{C\,\sqrt{n}\Big(R^{\frac{q}{2p}}\big(\tfrac{n}{d}\big)^{\frac{q-2}{4}} + R^{\frac{1}{2}}\big(\tfrac{n}{\log m}\big)^{\frac{p-2}{4}}\Big)} = \frac{\sqrt{2}}{C},$$

implying $C_2\sqrt{n}\,\delta \geq J(\rho, \delta)$. Additionally, we must check that $(\tfrac{\delta}{16}, \rho)$ is a valid interval for covering; an argument similar to Eq. (40) and $\rho < R^{\frac{1}{p}}$ ensures

$$\log N\big(\delta\,;\,\widetilde{\mathsf{S}}(\mathbf{T}_{p,q}(R), \rho)\big) \geq \max\{d, \log m\},$$

so the condition for Lemma C.11 is met.

Hence, applying Lemma C.11 gives that, with probability at least $1 - C_5\,\exp\Big\{-C_6\,n\big[R^{\frac{q}{p}}\big(\tfrac{n}{d}\big)^{\frac{q-2}{2}} + R\big(\tfrac{n}{\log m}\big)^{\frac{p-2}{2}}\big]\Big\}$, we have

$$\sup_{\underline{\mathbf{L}} \in \widetilde{\mathsf{S}}(\mathbf{T}_{p,q}(R), \rho)} \big|\langle \bar{\underline{\mathbf{E}}}, \underline{\mathbf{L}}\rangle\big| \leq \left[R^{\frac{q}{2p}}\big(\tfrac{n}{d\sigma^2}\big)^{\frac{q-2}{4}} + R^{\frac{1}{2}}\big(\tfrac{n}{\sigma^2 \log m}\big)^{\frac{p-2}{4}}\right]\rho. \qquad (43)$$

(*The other two cases for $p \leq q, m \leq d^2$ and $p \geq q$ follow the same procedure, so we omit full detail here.*)

Combining the arguments for all three regimes and applying Lemma C.11 completes the proof of the upper bounds for $\ell_p(S_q)$. $\qquad\square$

## C.5   PRELIMINARY NUMERICAL ILLUSTRATION OF THEOREM 4.2

To complement the theoretical analysis, we conduct preliminary simulations to illustrate how the recovery error obtained by our algorithm behaves under different structural parameters in the hard dual sparsity setting. We vary the spectral rank $r$, the frequency-domain sparsity $s$, the sample size $n$, and the noise-to-signal ratio (NSR). These experiments are not intended to establish minimax optimality but rather to provide an empirical illustration of the trends predicted by theory.

In particular, Theorem 4.2 provides an information-theoretic minimax lower bound on the estimation error under dual spectral sparsity, characterizing the fundamental statistical difficulty of the problem. Such results apply to any estimator, efficient or not. We do not claim that our algorithm achieves this minimax rate. Attaining such bounds algorithmically is highly challenging because the $\ell_p$-Schatten-$q$ regularizer is jointly nonconvex in both parameters and introduces cross-frequency coupling, preventing separable optimization. To the best of our knowledge, no known polynomial-time algorithm achieves the exact minimax rate in this tensor setting, reflecting a statistical–computational gap. Our algorithm therefore serves as a practical and interpretable approximation rather than an exact minimax solution.

Table 3: Preliminary numerical illustration of Theorem 4.2 (Part I). Squared error (SE) under different settings.

| Varying Factor | $r$ | $s$ | NSR | SE |
|---|---|---|---|---|
| NSR ($r = 1, s = 2$) | 1 | 2 | 0.05 | 2.09e-05 |
| NSR ($r = 1, s = 2$) | 1 | 2 | 0.10 | 3.44e-05 |
| NSR ($r = 1, s = 2$) | 1 | 2 | 0.15 | 4.92e-05 |
| NSR ($r = 1, s = 2$) | 1 | 2 | 0.20 | 7.47e-05 |
| Rank $r$ ($s = 2$, NSR=0.1) | 1 | 2 | 0.10 | 3.44e-05 |
| Rank $r$ ($s = 2$, NSR=0.1) | 2 | 2 | 0.10 | 8.05e-05 |
| Rank $r$ ($s = 2$, NSR=0.1) | 3 | 2 | 0.10 | 1.47e-04 |
| Rank $r$ ($s = 2$, NSR=0.1) | 4 | 2 | 0.10 | 2.29e-04 |
| Sparsity $s$ ($r = 2$, NSR=0.1) | 2 | 2 | 0.10 | 8.05e-05 |
| Sparsity $s$ ($r = 2$, NSR=0.1) | 2 | 4 | 0.10 | 2.16e-04 |
| Sparsity $s$ ($r = 2$, NSR=0.1) | 2 | 6 | 0.10 | 3.58e-04 |
| Sparsity $s$ ($r = 2$, NSR=0.1) | 2 | 8 | 0.10 | 5.45e-04 |

**Experimental Setting.** We aim to construct a third-order tensor $\underline{\mathbf{L}} \in \mathbb{R}^{d \times d \times m}$ that exhibits low-rank structure in only a few frequency slices while being zero in the others. To this end, we first define an auxiliary tensor $\tilde{\underline{\mathbf{L}}} \in \mathbb{R}^{d \times d \times m}$ in the frequency domain:

(1) select $s$ active slices; for each slice $i$, generate a rank-$r$ matrix $\tilde{\underline{\mathbf{L}}}^{(i)} = \mathbf{A}_i \mathbf{B}_i$, where $\mathbf{A}_i \in \mathbb{R}^{d \times r}$ and $\mathbf{B}_i \in \mathbb{R}^{r \times d}$ are drawn from Gaussian ensembles;

(2) set the remaining $m - s$ slices to zero, thereby inducing spectral sparsity;

(3) apply an inverse DCT along the third mode of $\tilde{\underline{\mathbf{L}}}$ to obtain the spatial-domain tensor $\underline{\mathbf{L}}$.

We then normalize $\|\underline{\mathbf{L}}\|_{\mathrm{F}} = 1$ and add isotropic Gaussian noise $\underline{\mathbf{E}} \sim \mathcal{N}(0, \sigma^2 \mathbf{I})$ with variance

$$\sigma^2 = \frac{\mathrm{NSR}^2}{d^2 m}.$$

The observed tensor is $\underline{\mathbf{Y}} = \underline{\mathbf{L}} + \underline{\mathbf{E}}$. For $n$ i.i.d. observations, the sample mean has effective noise variance $\sigma^2/n$, consistent with our paper's formulation.

We report the squared error

$$\mathrm{SE} = \|\hat{\underline{\mathbf{L}}} - \underline{\mathbf{L}}\|_{\mathrm{F}}^2,$$

where $\hat{\underline{\mathbf{L}}}$ is the recovered tensor. Since $\|\underline{\mathbf{L}}\|_{\mathrm{F}} = 1$, the expected SE scales with the total noise energy $\sigma^2 d^2 m$. Each SE value is averaged over 10 independent trials with independently generated $\underline{\mathbf{L}}$ and noise.

**Results.** Table 3 summarizes the results when varying NSR, rank $r$, and sparsity $s$.

The observed trends in Table 3 are consistent with theoretical predictions:

(1) recovery error grows with NSR;

(2) higher rank $r$ increases the error, as more components must be recovered;

(3) larger sparsity $s$ also worsens recovery, since more active frequencies increase complexity.

These results qualitatively match Theorem 4.2 (Part I), providing preliminary empirical support. Moreover, they also serve as an empirical reflection of the statistical–computational gap: while the minimax bound characterizes the best possible rate in principle, our practical algorithm demonstrates consistent yet sub-optimal behavior under increasing complexity. A more comprehensive ablation study is left for future work.

# D    EXPERIMENTAL DETAILS AND ADDITIVE RESULTS

This appendix compiles the full experimental and algorithmic details supporting our tensor $\ell_p(S_q)$ framework. We first outline the noisy tensor completion setting and then describe the ADMM-based solver used for optimization, including its update rules, computational cost, and observed convergence behavior. The remaining sections provide extended empirical studies, including ablations, robustness analyses, validation of the dual spectral structure, experiments on additional 3D and 4D tensors, Poisson noise settings, clustering benchmarks, and a comprehensive parameter sensitivity analysis.

## D.1    EXPERIMENTAL SETUP

**Noisy Tensor Completion Task Formulation.**    The noisy tensor completion problem aims to recover a structured tensor $\underline{\mathbf{L}}^\star$ from a set of noisy and incomplete observations. This problem is particularly relevant in applications such as hyperspectral image restoration, video inpainting, and remote sensing data reconstruction, where missing and corrupted data are common due to sensor limitations or transmission errors.

We consider a third-order tensor $\underline{\mathbf{L}}^\star \in \mathbb{R}^{d_1 \times d_2 \times d_3}$ that represents a clean, fully observed data source. However, due to data corruption and missing values, we only have access to a partially observed noisy tensor $\underline{\mathbf{Y}}$, which is generated as:

$$\underline{\mathbf{Y}} = \underline{\mathbf{B}} \odot (\underline{\mathbf{L}}^\star + \underline{\mathbf{E}}),$$

where:

- $\underline{\mathbf{B}}$ *(Binary Mask)*: A binary tensor of the same size as $\underline{\mathbf{L}}^\star$, where each entry $\underline{\mathbf{B}}_{i,j,k} \in \{0,1\}$ indicates whether the corresponding entry in $\underline{\mathbf{L}}^\star$ is observed ($\underline{\mathbf{B}}_{i,j,k} = 1$) or missing ($\underline{\mathbf{B}}_{i,j,k} = 0$).

- $\odot$ *(Hadamard Product)*: The element-wise product operator ensures that only the observed entries are retained, while unobserved entries are set to zero.

- $\underline{\mathbf{E}}$ *(Noise Tensor)*: Represents random additive noise introduced in the observed entries. Each entry of $\underline{\mathbf{E}}$ is sampled independently from a Gaussian distribution:

$$\underline{\mathbf{E}}_{i,j,k} \sim \mathcal{N}(0, \sigma^2),$$

where the noise level $\sigma$ is set as:

$$\sigma = c\sigma_0, \quad \text{with} \quad c = 0.05, \quad \sigma_0 = \frac{\|\underline{\mathbf{L}}^\star\|_F}{\sqrt{d_1 d_2 d_3}}.$$

Here, $\sigma_0$ represents a normalized noise scale based on the Frobenius norm of the clean tensor.

**Sampling Strategy and Experimental Settings.**    We apply a uniform random sampling strategy, where each entry of $\underline{\mathbf{L}}^\star$ is independently observed with probability $p$, meaning that a fraction $1 - p$ of the entries is missing. We consider three different missing ratios: $p \in \{0.05, 0.1, 0.15\}$, which correspond to scenarios where 95%, 90%, and 85% of the entries are missing, respectively. Each experiment is conducted over 10 independent trials to ensure statistical reliability, and the averaged Peak Signal-to-Noise Ratio (PSNR) and Structural Similarity Index (SSIM) are reported to evaluate reconstruction performance.

**Evaluation Metrics.**    To assess the quality of tensor reconstruction, we use the following two widely adopted metrics:

- *Peak Signal-to-Noise Ratio (PSNR)*:

$$\text{PSNR} = 10 \log_{10} \left( \frac{\max(\underline{\mathbf{L}}^\star)^2}{\frac{1}{d_1 d_2 d_3} \|\hat{\underline{\mathbf{L}}} - \underline{\mathbf{L}}^\star\|_F^2} \right).$$

A higher PSNR value indicates better reconstruction quality.

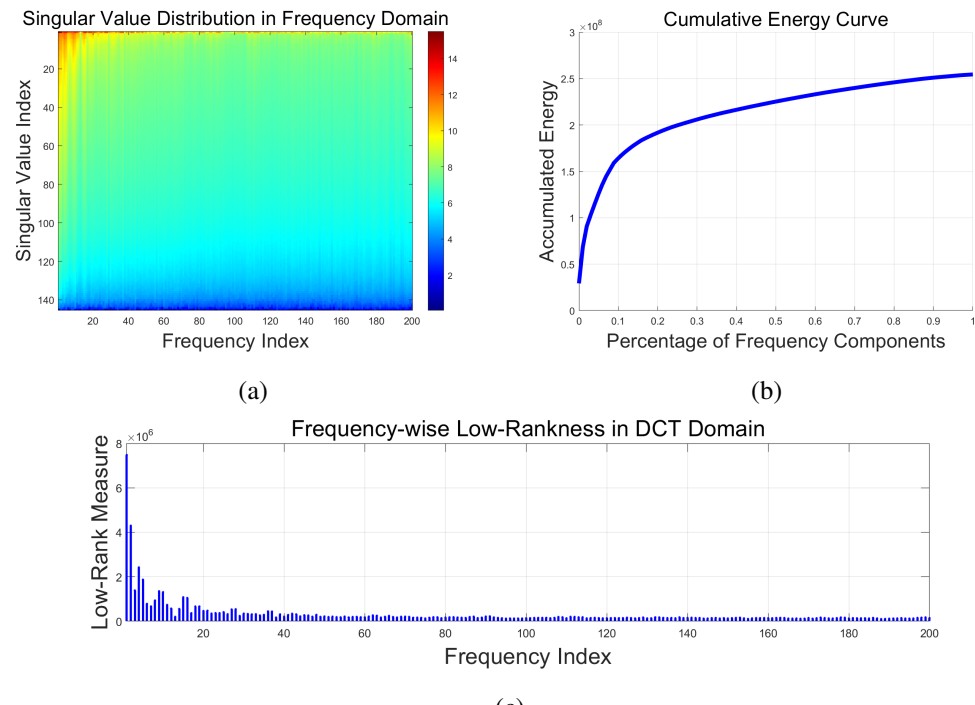

Figure 4: Visualization of dual-level sparsity structure using Indian Pines dataset. (a) The singular value heatmap exhibits both inter-frequency sparsity (horizontal variation) and intra-frequency low-rankness (vertical variation). (b)The cumulative energy curve reveals a majority of energy concentration in first 20% frequencies. (c) The frequency-wise low-rank measure $\|\boldsymbol{\sigma}(M(\mathbf{T})^{(i)})\|_1$ shows significant peaks in low frequencies and rapid decay afterwards.

- *Structural Similarity Index (SSIM)*:

$$\text{SSIM}(\hat{\underline{\mathbf{L}}}, \underline{\mathbf{L}}^\star) = \frac{(2\mu_{\hat{\underline{\mathbf{L}}}}\mu_{\underline{\mathbf{L}}^\star} + c_1)(2\sigma_{\hat{\underline{\mathbf{L}}}\underline{\mathbf{L}}^\star} + c_2)}{(\mu_{\hat{\underline{\mathbf{L}}}}^2 + \mu_{\underline{\mathbf{L}}^\star}^2 + c_1)(\sigma_{\hat{\underline{\mathbf{L}}}}^2 + \sigma_{\underline{\mathbf{L}}^\star}^2 + c_2)}.$$

This metric measures perceptual similarity between the recovered tensor $\hat{\underline{\mathbf{L}}}$ and the ground truth $\underline{\mathbf{L}}^\star$, where $\mu_{\hat{\underline{\mathbf{L}}}}, \mu_{\underline{\mathbf{L}}^\star}$ denote mean values, $\sigma_{\hat{\underline{\mathbf{L}}}}, \sigma_{\underline{\mathbf{L}}^\star}$ denote standard deviations, and $\sigma_{\hat{\underline{\mathbf{L}}}\underline{\mathbf{L}}^\star}$ represents cross-covariance. Parameters $c_1$ and $c_2$ are small constants to stabilize the division.

These metrics together provide a comprehensive evaluation of the reconstruction performance, ensuring that both numerical fidelity and structural integrity are preserved.

**Benchmark Methods.** We compare the proposed $\ell_p(S_q)$-quasi-norm against several existing low-rank tensor regularization techniques:

- *NN*: Matrix nuclear norm (Candès & Tao, 2010)
- *SNN*: Tucker-based tensor nuclear norm (Liu et al., 2013)
- *TNN-DFT*: Tensor nuclear norm with Discrete Fourier Transform (Zhang & Aeron, 2017)
- *TNN-DCT*: Tensor nuclear norm with Discrete Cosine Transform (Lu et al., 2019b)
- *k-Supp*: Tensor $k$-Support norm ($k = 2$) (Wang et al., 2021a)
- *$\ell_{1-2}$-norm*: Tensor $\ell_{1-2}$-norm (Tan et al., 2023)
- *Schatten-p-norm*: Tensor Schatten-$p$-norm ($p = 1/2$) (Kong et al., 2018)
- *LpSq (Proposed)*: The proposed $\ell_p(S_q)$-norm with parameters $(p, q) = (0.8961, 0.8966)$ (Setting I) and $(p, q) = (0.7, 0.71)$ (Setting II). We employ DCT as the transform operator $M(\cdot)$.

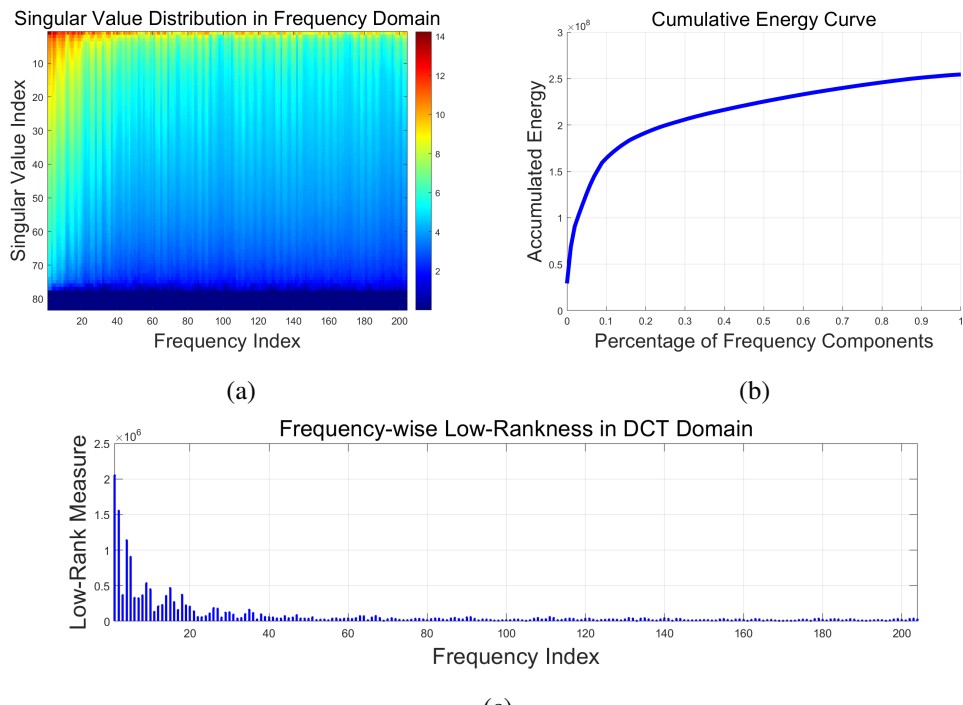

Figure 5: Visualization of dual-level sparsity structure using SalinasA dataset. (a) The singular value heatmap exhibits both inter-frequency sparsity (horizontal variation) and intra-frequency low-rankness (vertical variation). (b) The cumulative energy curve reveals a majority of energy concentration in first 20% frequencies. (c) The frequency-wise low-rank measure $\|\boldsymbol{\sigma}(M(\mathbf{\underline{T}})^{(i)})\|_1$ shows significant peaks in low frequencies and rapid decay afterwards.

For further comparison with different fixed transforms, see Appendix D.5; for comparison with adaptive (learnable) transforms, see Appendix D.7.

**Datasets.** The evaluation is conducted across multiple remote sensing datasets, encompassing hyperspectral, multispectral, and thermal imaging data.

1. *Hyperspectral Data.* We conduct noisy tensor completion on subsets of two representative hyperspectral datasets:

   - *Indian Pines*: This dataset was collected by the AVIRIS sensor in 1992 over the Indian Pines[3] test site in North-western Indiana and consists of $145 \times 145$ pixels and 200 corrected spectral reflectance bands. We use the first 30 bands in the experiments due to computational constraints and parameter tuning.
   - *Salinas A*: Acquired by the AVIRIS sensor over the Salinas Valley, California, in 1998, this dataset consists of 224 bands over a spectral range of 400–2500 nm, with a spatial extent of $86 \times 83$ pixels at a resolution of 3.7m. We use the first 30 bands in the experiments[4].

2. *Multispectral Images.* Multispectral imaging captures image data within specific wavelength ranges across the electromagnetic spectrum and is one of the most widely used modalities in remote sensing. This section presents simulated experiments on multispectral images. The original data consists of three multispectral images: *Cloth*, *Hair*, and *Jelly Beans*, from the

---

[3]The dataset is publicly available at https://www.ehu.eus/ccwintco/index.php?title=Hyperspectral_Remote_Sensing_Scenes.

[4]The dataset is available at https://www.ehu.eus/ccwintco/index.php?title=Hyperspectral_Remote_Sensing_Scenes.

Columbia MSI Database[5]. These images contain scenes of a variety of real-world objects, each with a resolution of $512 \times 512 \times 31$, with intensity values scaled to [0,1].

3. *Thermal Imaging Data.* Thermal infrared data provide crucial measurements of surface energy fluxes and temperatures for various remote sensing applications. We conduct experiments on an infrared dataset: *OSU Thermal Database*[6]. The sequences were recorded on the Ohio State University campus during February and March 2005, capturing multiple individuals moving in groups through the scene. We use the first 30 frames of Sequence 1, forming a tensor of size $320 \times 240 \times 30$.

### D.1.1 Parameter Selection for $(p, q)$

Selecting $(p, q)$ in quasi-norm regularization is fundamentally more challenging than tuning conventional scalar hyperparameters. These parameters jointly determine the *inductive geometry* of our model:

- $p$ modulates *inter-frequency sparsity*, and
- $q$ governs *intra-frequency low-rankness*.

Unlike scalar weights such as $\lambda$, the pair $(p, q)$ defines the structural bias of the regularizer itself and cannot be adjusted post hoc or absorbed into optimization scaling. Their effect is closer to *model specification* than typical hyperparameter tuning.

This challenge is further compounded by two key obstacles:

- **Nonconvexity**: Even in vector or matrix cases, quasi-norms (e.g., $\ell_{1/2}$) induce highly nonconvex landscapes.
- **Identifiability**: As emphasized in Xu (2010), no provably consistent or statistically efficient selection method exists for such parameters under limited data. This difficulty is amplified in the tensor setting due to the inter-band coupling introduced by the dual spectral regularizer.

**Our Strategy.** Given these challenges, we adopt a *simple and reproducible* two-stage procedure on a representative dataset (*Indian Pines*):

- *Coarse exploration:* We sweep $(p, q)$ over the grid $\{0.1, 0.2, \ldots, 1.0\}^2$, evaluating PSNR under 5% uniformly random observations. This avoids artificial validation splits that would further weaken the already sparse signal.
- *Refinement:* The coarse search consistently yields peaks near $p = q \approx 0.9$. We then refine within $[0.88, 0.92]^2$, and select $(p, q) = (0.8961, 0.8966)$ as the final configuration.

This choice, denoted **S0**, is used across all datasets and sampling ratios without further tuning. To study robustness, we additionally evaluate three further configurations spanning a wider region of the parameter space:

$$\textbf{S1} : (0.60, 0.61), \qquad \textbf{S2} : (0.70, 0.71), \qquad \textbf{S3} : (0.80, 0.81).$$

The results reported in Table 1 show that **S0–S3** yield comparable reconstruction quality across hyperspectral, multispectral, and thermal datasets. This consistency indicates that the performance of $\ell_p(S_q)$ does not hinge on finely tuning $(p, q)$, and that the framework remains effective across a contiguous set of parameter values. Further comparisons are presented in Appendix D.4. A detailed and dataset-level analysis of the sensitivity of $(p, q)$, including extended tables and heatmaps, is provided in Appendix D.15.

**Remark.** While adaptive or bilevel tuning strategies may offer further improvements, they remain computationally expensive and statistically unstable under nonconvex objectives and limited data. Developing scalable and principled methods for quasi-norm parameter selection is an important open direction, but is *beyond the scope of this work*, which focuses on the formulation, theoretical properties, and empirical benefits of the proposed $\ell_p(S_q)$ regularizer.

---

[5]Available at https://cave.cs.columbia.edu/repository/Multispectral.
[6]The dataset is available at http://vcipl-okstate.org/pbvs/bench/Data/03/download.html.

## D.2 TENSOR COMPLETION VIA AN ADMM-BASED ALGORITHM

We aim to estimate a structured tensor $\underline{\mathbf{L}}$ from noisy and incomplete observations $\underline{\mathbf{Y}}$. The optimization problem is formulated as:

$$\min_{\underline{\mathbf{L}}} \frac{1}{2}\|\underline{\mathbf{Y}} - \underline{\mathbf{B}} \odot \underline{\mathbf{X}}\|_{\mathrm{F}}^2 + \lambda\|\underline{\mathbf{L}}\|_{\ell_p(S_q)}^p,$$

We introduce an auxiliary variable $\underline{\mathbf{K}}$ and reformulate the problem as:

$$\min_{\underline{\mathbf{L}},\underline{\mathbf{K}}} \frac{1}{2}\|\underline{\mathbf{Y}} - \underline{\mathbf{B}} \odot \underline{\mathbf{K}}\|_{\mathrm{F}}^2 + \lambda\|\underline{\mathbf{L}}\|_{\ell_p(S_q)}^p, \quad \text{s.t.} \quad \underline{\mathbf{K}} = \underline{\mathbf{L}}.$$

This reformulation decouples the data fidelity term from the regularization, making it amenable to optimization via the Alternating Direction Method of Multipliers (ADMM).

We solve this problem using ADMM by iteratively updating $\underline{\mathbf{L}}$, $\underline{\mathbf{K}}$, and the dual variable $\underline{\mathbf{W}}$.

**Update of $\underline{\mathbf{L}}$:** The $\underline{\mathbf{L}}$-subproblem is:

$$\underline{\mathbf{L}}^{t+1} \in \arg\min_{\underline{\mathbf{L}}} \lambda\|\underline{\mathbf{L}}\|_{\ell_p(S_q)}^p + \frac{\mu^t}{2}\left\|\underline{\mathbf{L}} - \underline{\mathbf{K}}^t - \frac{\underline{\mathbf{W}}^t}{\mu^t}\right\|_{\mathrm{F}}^2.$$

Let $\underline{\mathbf{Z}} = \underline{\mathbf{K}}^t - \frac{\underline{\mathbf{W}}^t}{\mu^t}$, reducing the above problem to:

$$\underline{\mathbf{L}}^{t+1} \in \arg\min_{\underline{\mathbf{L}}} \lambda\|\underline{\mathbf{L}}\|_{\ell_p(S_q)}^p + \frac{\mu^t}{2}\|\underline{\mathbf{L}} - \underline{\mathbf{Z}}\|_{\mathrm{F}}^2.$$

Since $M(\underline{\mathbf{L}})$ allows separable updates across frequency components, this problem decomposes into $m$ independent subproblems:

$$\min \frac{1}{2}\|M(\underline{\mathbf{L}})_k - M(\underline{\mathbf{Z}})_k\|_{\mathrm{F}}^2 + \frac{\lambda}{\mu^t}\|M(\underline{\mathbf{L}})_k\|_{S_q}^p, \quad \forall k \in [m]. \tag{44}$$

To efficiently handle the Schatten-$q$ term, we employ *a weighted $\ell_{1/2}$-norm approximation* (see Section D.3 for comparison with $\ell_1$-norm approximation):

$$\sum_{i=1}^d w_{i,k} \cdot \sigma_i(M(\underline{\mathbf{L}})_k)^{1/2},$$

where the weight

$$w_{i,k} = \left(\sum_{j=1}^d \varsigma_{j,k}^q + \epsilon\right)^{\frac{p}{q}-1}\left(\varsigma_{j,k}^{1/2} + \epsilon\right)^{2q-1},$$

with $\varsigma_{j,k} = \sigma_j(M(\underline{\mathbf{L}}^t)_k)$, where $\underline{\mathbf{L}}^t$ denotes the tensor $\underline{\mathbf{L}}$ at the $t$-th iteration.

This formulation leads to a closed-form $\ell_{1/2}$-soft-thresholding update for each singular value:

$$\sigma_i^{(t+1)}(M(\underline{\mathbf{L}})_k) = \mathcal{S}_\theta^{\ell_{1/2}}(\sigma_i(M(\underline{\mathbf{Z}})_k)),$$

where $\theta = \frac{\lambda}{\mu^t}w_{i,k}$ and the $\ell_{1/2}$-soft-thresholding operator (Xu, 2010) is defined as:

$$\mathcal{S}_\theta^{\ell_{1/2}}(\sigma) = \begin{cases} \phi_\theta(\sigma), & |\sigma| > \frac{3\sqrt[3]{54}}{4}\theta^{2/3}, \\ \{\phi_\theta(\sigma), 0\}, & |\sigma| = \frac{3\sqrt[3]{54}}{4}\theta^{2/3}, \\ 0, & |\sigma| < \frac{3\sqrt[3]{54}}{4}\theta^{2/3}. \end{cases} \tag{45}$$

Here, the function $\phi_\theta(\sigma)$ is given by:

$$\phi_\theta(\sigma) = \frac{2}{3}\sigma\left(1 + \cos\left(\frac{2\pi}{3} - \frac{2}{3}\arccos\left(\frac{\theta}{8}|\sigma|^{-3/2}\right)\right)\right).$$

After updating singular values, the frequency component is reconstructed as:

$$M(\underline{\mathbf{L}}^{t+1})_k = \mathbf{U}_k \cdot \texttt{diag}(\boldsymbol{\sigma}^{(t+1)}(M(\underline{\mathbf{Z}})_k)) \cdot \mathbf{V}_k^\top, \quad \forall k \in [m],$$

where $\mathbf{U}_k$ and $\mathbf{V}_k$ are the left and right singular matrices of $M(\underline{\mathbf{Z}})_k$. Finally, applying the inverse $M$-transform yields the updated tensor $\underline{\mathbf{L}}^{t+1}$.

**Update of $\underline{\mathbf{K}}$:**   The auxiliary variable $\underline{\mathbf{K}}$ is updated by solving:

$$\underline{\mathbf{K}}^{t+1} = \arg\min_{\underline{\mathbf{K}}} \frac{1}{2}\|\underline{\mathbf{Y}} - \mathbf{B}\odot\underline{\mathbf{K}}\|_{\mathrm{F}}^2 + \frac{\mu^t}{2}\left\|\underline{\mathbf{K}} - \underline{\mathbf{L}}^{t+1} + \frac{\mathbf{W}^t}{\mu^t}\right\|_{\mathrm{F}}^2.$$

This step ensures that the solution remains within the feasible constraint region.

**Dual Variable Update:**   The Lagrange multiplier is updated as:

$$\underline{\mathbf{W}}^{t+1} = \underline{\mathbf{W}}^t + \mu^t(\underline{\mathbf{K}}^{t+1} - \underline{\mathbf{L}}^{t+1}).$$

The penalty parameter $\mu^t$ is dynamically updated to ensure convergence:

$$\mu^{t+1} = \min\{\gamma\mu^t, \mu_{\max}\}.$$

where $\gamma > 1$ is a predefined scaling factor.

This ADMM-based algorithm (summarized in Algorithm 1) efficiently solves the dual-level sparse tensor completion problem by iteratively enforcing structured sparsity through proximal updates while maintaining computational efficiency. The proposed weighted $\ell_{1/2}$-soft-thresholding mechanism ensures that the non-convex Schatten-$q$ regularization is effectively handled in each iteration.

---

**Algorithm 1** ADMM for dual-sparse tensor completion ($\ell_p$–Schatten-$q$)

---

**Input:** Observed tensor $\underline{\mathbf{Y}}$; binary mask $\mathbf{B}$; transform $M$ (with inverse $M^{-1}$); regulariser $\lambda$; parameters $p, q$; penalty $\mu_0 > 0$, growth factor $\gamma > 1$, maximum $\mu_{\max}$; tolerance $\varepsilon$.
**Output:** Completed tensor $\widehat{\underline{\mathbf{L}}}$.
  1: **Initialise:** $\underline{\mathbf{L}}^0 \leftarrow 0$, $\underline{\mathbf{K}}^0 \leftarrow \underline{\mathbf{Y}}$, $\underline{\mathbf{W}}^0 \leftarrow 0$, $\mu \leftarrow \mu_0$, $t \leftarrow 0$.
  2: **repeat**
    **1. $\underline{\mathbf{L}}$–update**
  3:     $\underline{\mathbf{Z}} \leftarrow \underline{\mathbf{K}}^t - \underline{\mathbf{W}}^t/\mu$                                                  $\triangleright$ dual correction
  4:     **for** each frequency slice $k = 1, \ldots, m$ **do**
  5:         $\mathbf{U}_k\,\mathrm{diag}(\boldsymbol{\sigma}_k)\,\mathbf{V}_k^\top \leftarrow \mathrm{SVD}\big(M(\underline{\mathbf{Z}})_k\big)$
  6:         Compute weights $w_{i,k} = \big(\sum_j \varsigma_{j,k}^q + \epsilon\big)^{\frac{p}{q}-1}\big(\varsigma_{i,k}^{1/2} + \epsilon\big)^{2q-1}$, where $\varsigma_{i,k} = \sigma_i(M(\underline{\mathbf{L}}^t)_k)$
  7:         $\theta_{i,k} \leftarrow (\lambda/\mu)\,w_{i,k}$
  8:         **Half-threshold:** $\sigma_{i,k}^{\mathrm{new}} = \mathcal{S}_{\theta_{i,k}}^{\ell_{1/2}}\big(\sigma_{i,k}\big)$                        $\triangleright$ Eq. (45)
  9:         $M(\underline{\mathbf{L}}^{t+1})_k \leftarrow U_k\,\mathrm{diag}(\boldsymbol{\sigma}_k^{\mathrm{new}})\,V_k^\top$
 10:     **end for**
 11:     $\underline{\mathbf{L}}^{t+1} \leftarrow M^{-1}\big(\{M(\underline{\mathbf{L}}^{t+1})_k\}_{k=1}^m\big)$
    **2. $\underline{\mathbf{K}}$–update**
 12:     $\underline{\mathbf{K}}^{t+1} \leftarrow \big(\mathbf{B}\odot\underline{\mathbf{Y}} + \mu\,(\underline{\mathbf{L}}^{t+1} - \underline{\mathbf{W}}^t/\mu)\big)\oslash\big(\mathbf{B} + \mu\mathbf{1}\big)$         $\triangleright$ element-wise $\oslash$ division
    **3. Dual update**
 13:     $\underline{\mathbf{W}}^{t+1} \leftarrow \underline{\mathbf{W}}^t + \mu\,(\underline{\mathbf{K}}^{t+1} - \underline{\mathbf{L}}^{t+1})$
    **4. Penalty update**
 14:     $\mu \leftarrow \min(\gamma\,\mu,\,\mu_{\max})$
 15:     $t \leftarrow t + 1$
 16: **until** $\|\underline{\mathbf{K}}^t - \underline{\mathbf{L}}^t\|_{\mathrm{F}}/\|\underline{\mathbf{Y}}\|_{\mathrm{F}} < \varepsilon$
 17: **return** $\widehat{\underline{\mathbf{L}}} \leftarrow \underline{\mathbf{L}}^t$

---

**Complexity Analysis:**   Each iteration of our algorithm involves (i) a linear transform on $d_1 d_2$ tubes of length $m$ (reducible to $O(d_1 d_2 m \log m)$ using DCT or FFT), and (ii) $m$ SVDs of $d_1 \times d_2$ matrices, yielding $O(m d_1 d_2 \min(d_1, d_2))$ complexity. These operations are the same basic operations used in standard TNN solvers, so the overall iteration cost remains within the same order of magnitude. Despite the nonconvexity, our algorithm converges efficiently in practice.

D.2.1   CONVERGENCE OF ALGORITHM 1

Our ADMM-based algorithm exhibits stable empirical convergence across all experiments. As illustrated in Figure 6, the objective value typically stabilizes within 200 iterations. This behavior is consistent across different sampling ratios and datasets.

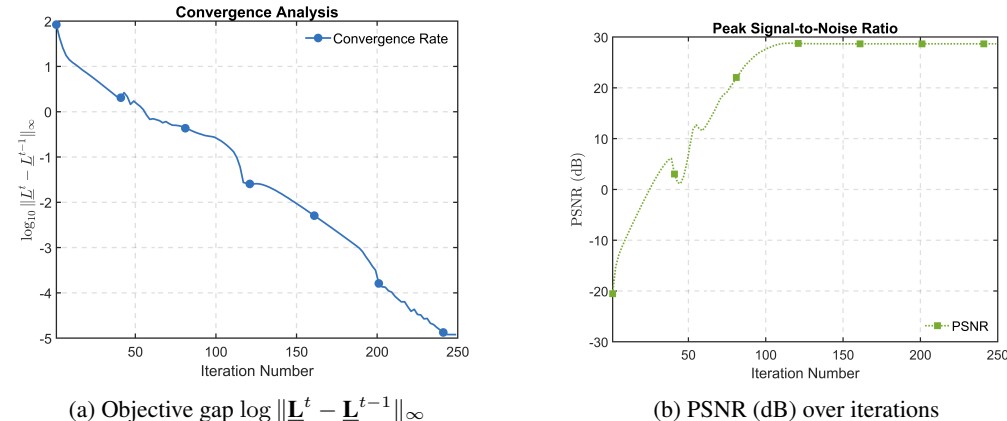

(a) Objective gap $\log \|\underline{\mathbf{L}}^t - \underline{\mathbf{L}}^{t-1}\|_\infty$        (b) PSNR (dB) over iterations

Figure 6: Empirical convergence of the proposed ADMM algorithm on the *Salinas A* dataset. Left: the variable gap between consecutive iterates. Right: the evolution of PSNR across iterations.

From a theoretical perspective, establishing convergence guarantees is challenging due to the nonconvex and nonsmooth nature of the objective, which involves transform-domain operations, SVD, and a weighted $\ell_{1/2}$-quasi-norm. These components render standard ADMM theory inapplicable. While recent advances in nonconvex ADMM provide promising tools, they require problem-specific adaptations to accommodate the spectral structure and inter-frequency coupling present in our model. At present, there is no well-established convergence theory that directly covers such bilayer nonconvex regularizers, making this an open problem for the broader community. We therefore view our empirical convergence as supportive evidence and consider the development of tailored theoretical guarantees a promising direction for future research.

### D.3   ABLATION STUDY ON WEIGHTED $\ell_{1/2}$ VS. WEIGHTED $\ell_1$ SURROGATES

To approximate the nonconvex Schatten-$q$ regularizer in Eq. (44) in a computationally efficient manner, we also considered weighted $\ell_1$ surrogates, in addition to the weighted $\ell_{1/2}$ formulation adopted in Appendix D.2. While the weighted $\ell_1$ formulation is convex and widely used, it proved empirically less effective in preserving the underlying structure of transformed tensor slices. As shown in Table 4, the weighted $\ell_{1/2}$ surrogate consistently achieves higher PSNR and SSIM scores across different datasets and sampling rates, including Salinas A and Indian Pines. This improvement suggests that the nonconvex $\ell_{1/2}$ penalty better captures the spectral decay and low-rank characteristics inherent in each transformed slice. Based on these results, we adopt the weighted $\ell_{1/2}$ approximation throughout our method.

Table 4: Comparison of weighted $\ell_1$ and weighted $\ell_{1/2}$ surrogates for approximating the proposed regularizer on Salinas A and Indian Pines datasets under different sampling rates (SR).

| Dataset | SR | Metric | Weighted $\ell_1$ | Weighted $\ell_{1/2}$ |
|---|---|---|---|---|
| Salinas A | 5% | PSNR | 28.03 | 28.43 |
| | | SSIM | 0.7297 | 0.7374 |
| | 10% | PSNR | 30.38 | 31.81 |
| | | SSIM | 0.8126 | 0.8484 |
| | 15% | PSNR | 32.38 | 33.23 |
| | | SSIM | 0.8588 | 0.8830 |
| Indian Pines | 5% | PSNR | 25.73 | 27.05 |
| | | SSIM | 0.6600 | 0.6740 |
| | 10% | PSNR | 26.88 | 28.92 |
| | | SSIM | 0.6680 | 0.7617 |
| | 15% | PSNR | 27.04 | 29.89 |
| | | SSIM | 0.6731 | 0.7997 |

### D.4 ABLATION STUDY ON THE EFFECT OF DUAL SPECTRAL SPARSITY VS. ITERATIVE REWEIGHTING

To assess whether the empirical gain of our method stems from dual spectral sparsity or simply from iterative reweighting, we first compared against the classical reweighted tensor nuclear norm (RWTNN) method (Baburaj & George, 2016) for tensor Schatten-$q$ minimization. In this baseline, the weighting factor is tuned so that the objective exactly coincides with a Schatten-$q$ quasi-norm when $q = 0.8966$. If the improvement came solely from reweighting singular values, RWTNN should match our performance. Table 5 reports PSNR and SSIM values on Cloth (downsampled to $256 \times 256 \times 31$) and OSU Thermal under noiseless tensor completion, directly comparing our method with RWTNN.

Table 5: Comparison in PSNR/SSIM values with RWTNN-based tensor Schatten-$q$ ($q = 0.8966$) minimization on Cloth and OSU Thermal.

| Dataset | Method | SR=5% | SR=10% | SR=15% |
|---|---|---|---|---|
| Cloth | RWTNN | 24.30 / 0.6804 | 28.40 / 0.8489 | 31.11 / 0.9104 |
| | Ours | **25.29 / 0.7240** | **29.25 / 0.8700** | **31.96 / 0.9273** |
| OSU Thermal | RWTNN | 29.94 / 0.8992 | 34.55 / 0.9643 | 36.62 / 0.9758 |
| | Ours | **31.58 / 0.9346** | **35.70 / 0.9699** | **37.54 / 0.9782** |

The results in Table 5 show that our method consistently outperforms RWTNN. However, this alone does not prove that the improvement stems from frequency–low-rank coupling. Part of the gain can also be attributed to our use of weighted $\ell_{1/2}$ minimization, which induces stronger low-rankness than the $\ell_1$ weighting in standard nuclear norm reweighting. Yet the key question remains: how much of the benefit arises from the coupling itself? To probe this, we next fix $q = 0.8966$ and vary the ratio $p/q$ to directly examine the effect of cross-band coupling.

**Effect of the $p/q$ Ratio on Coupling.** To further isolate the role of cross-band coupling, we fixed $q = 0.8966$ and varied the ratio $p/q$. As $p \to q$, the coupling effect is expected to vanish, so the empirical trend directly reflects how much improvement can be attributed to frequency–low-rank interaction.

Table 6: Effect of varying $p/q$ on Cloth and OSU Thermal (fixed $q = 0.8966$).

| Dataset | Setting | SR=5% | SR=10% | SR=15% |
|---|---|---|---|---|
| Cloth | RWTNN ($p = 0.8966$) | 24.30 / 0.6804 | 28.40 / 0.8489 | 31.11 / 0.9104 |
| | Ours ($p/q = 1$) | 25.14 / 0.7168 | 29.11 / 0.8700 | 31.71 / 0.9235 |
| | Ours ($p/q = 0.999$) | 25.29 / 0.7240 | 29.25 / 0.8700 | 31.96 / 0.9273 |
| | Ours ($p/q = 0.99$) | **25.49 / 0.7366** | **29.53 / 0.8789** | **32.27 / 0.9318** |
| | Ours ($p/q = 0.98$) | 25.32 / 0.7299 | 29.28 / 0.8748 | 32.26 / 0.9312 |
| | Ours ($p/q = 0.97$) | 24.83 / 0.7064 | 28.90 / 0.8596 | 32.11 / 0.9288 |
| OSU Thermal | RWTNN ($p = 0.8966$) | 29.94 / 0.8992 | 34.55 / 0.9643 | 36.62 / 0.9758 |
| | Ours ($p/q = 1$) | 31.32 / 0.9298 | 35.65 / 0.9699 | 37.40 / 0.9778 |
| | Ours ($p/q = 0.999$) | 31.58 / 0.9346 | 35.70 / 0.9699 | 37.54 / 0.9782 |
| | Ours ($p/q = 0.99$) | **31.86 / 0.9354** | **35.84 / 0.9707** | **37.65 / 0.9787** |
| | Ours ($p/q = 0.98$) | 31.42 / 0.9277 | 35.75 / 0.9700 | 37.42 / 0.9780 |
| | Ours ($p/q = 0.97$) | 31.03 / 0.9181 | 35.62 / 0.9696 | 37.37 / 0.9779 |

According to the results in Table 6, we have the following observations:

- Even mild frequency sparsity ($p/q = 0.99$) yields consistent improvements over the $p = q$ case.

- As $p \to q$, the coupling effect diminishes, explaining the modest gains in near-decoupled regimes.

Thus, frequency sparsity is a meaningful modeling ingredient, but its empirical advantage depends on how strongly it couples with the low-rank component.

Based on these observations, our interpretation and discussion are as follows:

1. *Effectiveness with room for improvement*
   The gain at $p/q = 0.99$ suggests that frequency sparsity provides tangible benefits. While the observed margin is moderate, it indicates conservative performance under the current solver, leaving open the possibility that stronger algorithms could further amplify these gains.

2. *Why the performance gain fades as $p \to q$*
   In our formulation, the per-band regularization term is typically

$$F_{q,\tau}^{(k)} = \left( \sum_i (\sigma_i^{(k)})^q \right)^\tau, \quad q < 1, \ \tau = p/q < 1.$$

   When $\tau < 1$, the problem becomes *substantially more non-convex* than the widely studied $\tau \geq 1$ cases in sparse optimization (Hu et al., 2017). To our knowledge, most existing bilayer non-convex sparse optimization methods focus on $q < 1, \ \tau \geq 1$ and are not applicable here. The added difficulty for $\tau < 1$ lies in the stronger non-convexity, which increases optimization challenges.

   To make the problem tractable, let the current iterate be $\{\sigma_{i,0}^{(k)}\}_{i=1}^d$. We adopt the heuristic decomposition $x^\tau = x^1 \cdot x^{\tau-1}$, leading to the surrogate

$$H_{q,\tau}^{(k)} = \left( \sum_i (\sigma_i^{(k)})^q \right) \cdot \left( \sum_i (\sigma_{i,0}^{(k)})^q + \varepsilon \right)^{\tau-1} \approx F_{q,\tau}^{(k)},$$

   which enables separable weighted-$\ell_{1/2}$ updates. This approximation is *tightest when $\tau \approx 1$*, and the gap from $F_{q,\tau}^{(k)}$ widens as $\tau$ moves away from 1, potentially diminishing the expected benefits of frequency–low-rank coupling.

3. *Does the modest performance margin imply limited potential for frequency sparsity?*
   Not necessarily. In the near-decoupled regime ($p \approx q$), the coupling signal is intrinsically weak. Moreover, when $\tau < 1$ deviates from 1, the surrogate becomes a looser approximation of the true objective. Both effects can compress the observable gains. Therefore, the moderate improvement at $p/q = 0.99$ should be interpreted as *a conservative lower bound on what frequency sparsity could achieve with tighter surrogates or stronger coupling*.

4. *Special case $p = q < 1$*
   In this limit, our method reduces to a weighted-$\ell_{1/2}$-based Schatten-$q$ minimization. As shown in Table 4 and Table 5, this variant exhibits improvements over classical weighted nuclear norms. This provides supportive evidence that even in a simplified setting, our weighted-$\ell_{1/2}$-based method offers potential advantages deserving further study.

The optimization of our two-layer nonconvex $\ell_p(S_q)$ regularization is *inherently more challenging than that of the classical single-layer $\ell_p$ regularization* (Xu, 2010). At present, the understanding of such bilayer nonconvex formulations remains limited, and our solver should be viewed as a preliminary attempt. More broadly, the results indicate that frequency–low-rank coupling is a promising modeling component, while underscoring the need for stronger algorithmic tools and deeper theoretical analyses to fully exploit its potential in tensor recovery.

## D.5 ABLATION STUDY ON THE EFFECT OF DIFFERENT TRANSFORMS

Our main experiments employ the Discrete Cosine Transform (DCT) due to its ability to capture smooth spectral structures. To examine whether the observed gains stem primarily from the transform itself or from the proposed regularizer, we conducted preliminary experiments using alternative invertible transforms: the Discrete Fourier Transform (DFT), random orthogonal transforms (Lu et al., 2019b), and an oracle transform constructed from the left singular vectors of the mode-3 unfolding of the ground-truth tensor (Zhang & Ng, 2021). Results on Salinas A are summarized in Table 7.

Table 7: PSNR/SSIM results on Salinas A under different transforms.

| Method | SR = 5% | SR = 10% | SR = 15% |
|---|---|---|---|
| TNN-DFT | 22.55 / 0.5667 | 25.72 / 0.7027 | 28.06 / 0.7804 |
| Ours-DFT $(p, q) = (0.8961, 0.8966)$ | 23.53 / 0.5469 | 27.04 / 0.6807 | 28.58 / 0.7321 |
| Ours-DFT $(p, q) = (0.75, 0.75)$ | 24.10 / 0.5917 | 27.85 / 0.7261 | 29.91 / 0.7969 |
| TNN-Random | 16.28 / 0.2331 | 22.21 / 0.4996 | 24.31 / 0.5692 |
| Ours-Random $(p, q) = (0.8961, 0.8966)$ | 17.25 / 0.2430 | 23.63 / 0.5041 | 25.46 / 0.5863 |
| TNN-Oracle | 29.06 / 0.8362 | 32.04 / 0.8924 | 33.77 / 0.9169 |
| Ours-Oracle $(p, q) = (0.8961, 0.8966)$ | **31.37 / 0.8391** | **34.05 / 0.9070** | **35.83 / 0.9324** |

These results suggest that the proposed regularizer remains effective across different invertible transforms, with consistent advantages over TNN even in the absence of DCT preprocessing. This indicates that the observed improvements stem primarily from the dual spectral modeling itself rather than from a particular transform. At the same time, the oracle case highlights the potential of combining our framework with adaptive or learned transforms.

### D.6 ROBUSTNESS TO HIGHER SAMPLING RATIOS

To further examine the stability of parameter settings at higher observation ratios, we evaluated the fixed configuration $(p, q) = (0.8961, 0.8966)$ with a constant regularization parameter $\lambda = 0.4$ on the Salinas A and OSU Thermal datasets. The corresponding PSNR and SSIM results are reported in Table 8.

These results indicate that the fixed configuration generalizes well and continues to improve as more data become available, with no sign of performance collapse or overfitting at higher sampling ratios.

Table 8: Performance of the fixed parameter setting $(p, q) = (0.8961, 0.8966)$ under higher sampling ratios.

| Sampling Ratio | Salinas A | | OSU Thermal | |
|---|---|---|---|---|
| | PSNR | SSIM | PSNR | SSIM |
| 30% | 35.27 | 0.9275 | 37.84 | 0.9698 |
| 40% | 36.52 | 0.9437 | 38.71 | 0.9737 |
| 50% | 37.58 | 0.9555 | 39.26 | 0.9758 |

### D.7 COMPARISON WITH STRONGER BASELINES

To further assess the effectiveness of the proposed framework, we compare against more advanced tensor nuclear norm variants, including the Framelet TNN (Jiang et al., 2020) and Adaptive TNN (Kong et al., 2021). Following Zhang & Ng (2021), we use a variant of the Adaptive TNN which constructs the transform matrix at each iteration using the left singular vectors of the tensor's mode-3 unfolding, thereby avoiding explicit rank selection in Kong et al. (2021). For fairness, we adopt the same adaptive scheme for our method. Results on Salinas A are reported in Table 9.

Table 9: Comparison (PSNR/SSIM) with Framelet TNN and Adaptive TNN on Salinas A with $(p, q) = (0.8961, 0.8966)$.

| Method | SR = 5% | SR = 10% | SR = 15% |
|---|---|---|---|
| Framelet TNN | 26.01 / 0.6382 | 27.76 / 0.7159 | 28.55 / 0.7461 |
| Adaptive TNN | 27.76 / 0.8245 | 30.91 / 0.8846 | 32.90 / 0.9173 |
| Ours-DCT | 28.43 / 0.7374 | 31.81 / 0.8484 | 33.23 / 0.8830 |
| Ours-Adaptive | **29.77 / 0.8093** | **33.38 / 0.8987** | **35.34 / 0.9288** |

These results show that our method achieves competitive or superior performance compared to both Framelet and Adaptive TNN. The gain arises not only from transform selection but also from the joint spectral modeling introduced by our regularizer. Moreover, the framework naturally unifies fixed and adaptive transforms, as well as classical low-rank models, within a single formulation.

### D.8 Noiseless Tensor Completion

To further examine the generality of the proposed framework, we conducted experiments on standard noiseless tensor completion. Results on Salinas A with $(p, q) = (0.8961, 0.8966)$ are reported in Table 10. These results confirm that our method maintains promising performance in the noiseless setting, complementing its robustness under noisy observations. This suggests that the benefits of dual spectral sparsity are not limited to noise suppression, but also extend to enhancing recovery quality in idealized conditions.

Table 10: Tensor completion (noiseless) results (PSNR/SSIM) on Salinas A.

| Method | SR = 5% | SR = 10% | SR = 15% |
|---|---|---|---|
| TNN-DFT | 22.95 / 0.5987 | 27.60 / 0.7874 | 28.28 / 0.7985 |
| TNN-DCT | 28.29 / 0.7976 | 33.24 / 0.9159 | 35.89 / 0.9506 |
| $k$-Supp | 22.85 / 0.5976 | 27.49 / 0.7708 | 28.19 / 0.7951 |
| $\ell_{1-2}$ | 23.17 / 0.6129 | 27.88 / 0.7942 | 30.75 / 0.8735 |
| Schatten-$1/2$ | 23.41 / 0.4937 | 28.77 / 0.7571 | 31.40 / 0.8475 |
| Framelet TNN | 28.56 / 0.7060 | 33.44 / 0.8673 | 35.75 / 0.9117 |
| Adaptive TNN | 31.54 / 0.8962 | 35.93 / 0.9538 | 38.45 / 0.9733 |
| Ours-DCT | 31.25 / 0.8425 | 36.01 / 0.9403 | 38.96 / 0.9704 |
| Ours-Adaptive | **33.90 / 0.9148** | **38.57 / 0.9675** | **40.81 / 0.9807** |

### D.9 Robustness under Non-Gaussian Noise

Our theoretical analysis focuses on the Gaussian setting, which provides a clean statistical model for establishing minimax bounds. Nevertheless, the proposed framework is not inherently restricted to Gaussian noise. To assess robustness in practice, we conducted experiments with uniform noise, where each entry is sampled from $[0, u]$ with $u = 0.05 \cdot \|\mathbf{L}\|_{\mathrm{F}} / \sqrt{d_1 d_2 d_3}$. Results on Salinas A are reported in Table 11. These results demonstrate resilience to mild non-Gaussian perturbations.

Table 11: Performance of the proposed method on Salinas A under uniform noise.

| Sampling Ratio | PSNR | SSIM |
|---|---|---|
| 30% | 33.98 | 0.9461 |
| 40% | 34.80 | 0.9612 |
| 50% | 35.22 | 0.9667 |

### D.10 Statistical Validation of the Dual Spectral Structure

Dual spectral sparsity, i.e., the simultaneous presence of inter-frequency sparsity and intra-frequency low-rankness in the transform domain, is a central modeling assumption of this work. While we have already included examples from medical and hyperspectral data (e.g., *MRI, CT, Salinas A*, and *Indian Pines*), we provide here a systematic and statistically grounded validation that goes well beyond illustrative evidence. Our goal is to determine *whether dual spectral sparsity is an inherent structural pattern rather than an artifact of specific examples*. To assess its generality, we quantitatively compare real tensors with principled baselines such as *Gaussian random tensors*, evaluating whether inter-frequency sparsity and intra-frequency low-rankness exhibit statistically significant differences between real tensors and their random counterparts.

Specifically, given a tensor $\mathbf{T} \in \mathbb{R}^{d_1 \times d_2 \times m}$ and a linear transform $M$ applied along the third mode, we seek to evaluate two questions:

(i) *Inter-frequency sparsity: Is the energy across frequency slices highly concentrated?*

(ii) *Intra-frequency low-rankness: Do the singular values within each slice decay rapidly, indicating low effective rank?*

To answer these questions in a statistically rigorous manner, we first construct quantitative metrics for the two phenomena and then perform hypothesis tests against matched Gaussian random tensors.

**Quantitative Metrics for Dual Spectral Sparsity.** We introduce the following quantitative metrics for the two phenomena:

(i) *Quantifying intra-frequency low-rankness.* For each transformed slice $M(\mathbf{T})_{:,:,i}$, we adopt the *stable rank* (Ipsen & Saibaba, 2025):

$$\text{s-rank}(\mathbf{A}) := \frac{\|\mathbf{A}\|_{\mathrm{F}}^2}{\|\mathbf{A}\|_{\mathrm{spec}}^2} = \frac{\sum_j \sigma_j(\mathbf{A})^2}{\max_j \sigma_j(\mathbf{A})^2},$$

which captures the "effective dimension" of a spectrum: a rank-one matrix has stable rank 1, whereas a full-rank matrix with equal singular values achieves the maximum possible stable rank. We then define the *average frequency-wise stable rank:*

$$\text{F-rank}_{\text{avg}}(\mathbf{T}) := \frac{1}{m} \sum_{i=1}^m \text{s-rank}(M(\mathbf{T})_{:,:,i}), \qquad \mathfrak{r}(\mathbf{T}) := \frac{\text{F-rank}_{\text{avg}}(\mathbf{T})}{\min(d_1, d_2)}.$$

Here $\mathfrak{r}(\mathbf{T}) \in (0, 1]$ serves as a normalized indicator of slice-level low-rankness: values closer to 0 indicate strong intra-frequency low-rankness, and values close to 1 indicate nearly full-rank behavior.

(ii) *Quantifying inter-frequency sparsity.* Let $E_i := \|M(\mathbf{T})_{:,:,i}\|_{\mathrm{F}}^2$ denote the energy of the $i$-th frequency slice. We define

$$\text{F-sparsity}(\mathbf{T}) := \frac{\sum_{i=1}^m E_i}{\max_i E_i}, \qquad \mathfrak{s}(\mathbf{T}) := \frac{\text{F-sparsity}(\mathbf{T})}{m}.$$

The ratio $\mathfrak{s}(\mathbf{T})$ equals 1 if all frequencies carry similar energy and becomes smaller as energy becomes concentrated in a few slices. Thus $\mathfrak{s}(\mathbf{T})$ acts as a normalized measure of inter-frequency sparsity.

**Hypothesis Testing on Typical Tensor Datasets.** To quantitatively assess the universality of dual spectral sparsity, we apply the proposed metrics to two typical tensor datasets: 22 grayscale YUV video tensors[7] and 31 multispectral datasets[8]. Specifically, for each tensor $\mathbf{T}$, we compute:

- $\mathfrak{s}(\mathbf{T})$: the *frequency sparsity ratio*, measuring how concentrated the transform-domain energy is across frequency slices;

- $\mathfrak{r}(\mathbf{T})$: the *stable-rank ratio*, capturing the effective intra-slice low-rankness of the transformed slices;

- $\mathfrak{s}(\text{rand})$ and $\mathfrak{r}(\text{rand})$: the same quantities computed for Gaussian random tensors with identical size and mean energy.

We expect real tensors to exhibit a clear gap from random baselines, reflecting strong inter-frequency sparsity and intra-frequency low-rankness.

As reported in Table 12, both quantities $\mathfrak{s}(\mathbf{T})$ and $\mathfrak{r}(\mathbf{T})$ for video tensors are consistently several orders of magnitude smaller than those of matched Gaussian random tensors, indicating that transform-domain energy concentrates among a few frequency slices and that the active slices are intrinsically low-rank. These findings suggest that *dual spectral sparsity is not an incidental artifact of the transform but an inherent property shared across video data.*

Similar results are also observed in Table 13 for 31 multispectral images, where $\mathfrak{s}(\mathbf{T})$ remains on the order of $10^{-2}$–$10^{-3}$, while Gaussian random tensors of identical size consistently yield values around 0.9. Similarly, the stable-rank ratios $\mathfrak{r}(\mathbf{T})$ stay between $10^{-3}$ and $10^{-2}$, in sharp contrast to the $\approx 0.25$ ratios observed for the random baselines. This again confirms strong inter-frequency sparsity and pronounced intra-frequency low-rankness, reinforcing the universality of dual spectral sparsity across modalities beyond videos and medical images.

This conclusion is further enhanced by ***formal hypothesis tests*** conducted over all 22 video tensors and 31 multispectral tensors.

---

[7]https://media.xiph.org/video/derf/
[8]https://cave.cs.columbia.edu/repository/Multispectral

Table 12: Dual spectral sparsity statistics across 22 YUV video tensors. $\mathfrak{s}(\mathbf{T})$ and $\mathfrak{r}(\mathbf{T})$ quantify inter-frequency sparsity and intra-frequency low-rankness of real tensors; the last two columns report the corresponding values for Gaussian random tensors of the same size.

| Dataset | $\mathfrak{s}(\mathbf{T})$ | $\mathfrak{r}(\mathbf{T})$ | $\mathfrak{s}(\text{rand})$ | $\mathfrak{r}(\text{rand})$ |
|---|---|---|---|---|
| akiyo | $3.33 \times 10^{-3}$ | $2.99 \times 10^{-2}$ | $9.49 \times 10^{-1}$ | $2.84 \times 10^{-1}$ |
| bridge-far | $4.76 \times 10^{-4}$ | $1.10 \times 10^{-1}$ | $9.45 \times 10^{-1}$ | $2.84 \times 10^{-1}$ |
| bus | $6.73 \times 10^{-3}$ | $1.74 \times 10^{-2}$ | $9.78 \times 10^{-1}$ | $2.80 \times 10^{-1}$ |
| carphone | $2.62 \times 10^{-3}$ | $4.79 \times 10^{-2}$ | $9.54 \times 10^{-1}$ | $2.84 \times 10^{-1}$ |
| claire | $2.02 \times 10^{-3}$ | $4.10 \times 10^{-2}$ | $9.54 \times 10^{-1}$ | $2.84 \times 10^{-1}$ |
| coastguard | $3.33 \times 10^{-3}$ | $5.39 \times 10^{-2}$ | $9.42 \times 10^{-1}$ | $2.83 \times 10^{-1}$ |
| container | $3.33 \times 10^{-3}$ | $3.82 \times 10^{-2}$ | $9.51 \times 10^{-1}$ | $2.84 \times 10^{-1}$ |
| flower | $4.00 \times 10^{-3}$ | $6.39 \times 10^{-2}$ | $9.73 \times 10^{-1}$ | $2.81 \times 10^{-1}$ |
| foreman | $3.34 \times 10^{-3}$ | $6.94 \times 10^{-2}$ | $9.43 \times 10^{-1}$ | $2.84 \times 10^{-1}$ |
| grandma | $1.15 \times 10^{-3}$ | $1.05 \times 10^{-1}$ | $9.50 \times 10^{-1}$ | $2.84 \times 10^{-1}$ |
| hall | $3.33 \times 10^{-3}$ | $5.89 \times 10^{-2}$ | $9.51 \times 10^{-1}$ | $2.84 \times 10^{-1}$ |
| highway | $5.00 \times 10^{-4}$ | $7.20 \times 10^{-2}$ | $9.43 \times 10^{-1}$ | $2.84 \times 10^{-1}$ |
| miss-america | $6.67 \times 10^{-3}$ | $5.42 \times 10^{-2}$ | $9.48 \times 10^{-1}$ | $2.84 \times 10^{-1}$ |
| mobile | $3.34 \times 10^{-3}$ | $9.91 \times 10^{-2}$ | $9.44 \times 10^{-1}$ | $2.84 \times 10^{-1}$ |
| mother-daughter | $3.33 \times 10^{-3}$ | $5.86 \times 10^{-2}$ | $9.40 \times 10^{-1}$ | $2.85 \times 10^{-1}$ |
| news | $3.33 \times 10^{-3}$ | $4.03 \times 10^{-2}$ | $9.31 \times 10^{-1}$ | $2.85 \times 10^{-1}$ |
| salesman | $2.23 \times 10^{-3}$ | $6.69 \times 10^{-2}$ | $9.45 \times 10^{-1}$ | $2.84 \times 10^{-1}$ |
| silent | $3.33 \times 10^{-3}$ | $4.27 \times 10^{-2}$ | $9.39 \times 10^{-1}$ | $2.84 \times 10^{-1}$ |
| stefan | $1.11 \times 10^{-2}$ | $5.59 \times 10^{-2}$ | $9.82 \times 10^{-1}$ | $2.80 \times 10^{-1}$ |
| suzie | $6.67 \times 10^{-3}$ | $4.94 \times 10^{-2}$ | $9.63 \times 10^{-1}$ | $2.85 \times 10^{-1}$ |
| tempete | $3.87 \times 10^{-3}$ | $5.96 \times 10^{-2}$ | $9.76 \times 10^{-1}$ | $2.81 \times 10^{-1}$ |
| waterfall | $3.85 \times 10^{-3}$ | $5.58 \times 10^{-2}$ | $9.74 \times 10^{-1}$ | $2.81 \times 10^{-1}$ |

Table 14 summarizes the statistical testing between video tensors and matched Gaussian random tensors, which gives rise to the following findings:

- **Statistical test on inter-frequency sparsity** $\mathfrak{s}(\mathbf{T})$: the test video data yield extremely small values and show a complete separation in distribution from random tensors, with both Wilcoxon and KS tests reporting $p < 10^{-6}$ and an exceptionally large effect size (Cohen's $d = -91.84$). This confirms that transform-domain energy is highly concentrated among only a few frequency slices in the test video data.

- **Statistical test on intra-frequency low-rankness** $\mathfrak{r}(\mathbf{T})$: the stable-rank ratios are again substantially smaller for real tensors, with both tests reporting $p < 10^{-6}$ and a large effect size (Cohen's $d = -13.88$). This demonstrates that singular values within each active frequency slice decay much more rapidly than in random tensors, revealing strong slice-wise low-rank structure.

Table 15 reports the hypothesis testing results over all 31 multispectral tensors. For both inter-frequency sparsity $\mathfrak{s}(\mathbf{T})$ and intra-frequency stable-rank ratios $\mathfrak{r}(\mathbf{T})$, the Wilcoxon and Kolmogorov–Smirnov tests consistently yield $p < 10^{-6}$, indicating complete distributional separation from their Gaussian random counterparts. The corresponding effect sizes are extremely large (Cohen's $d = -4.26 \times 10^2$ for $\mathfrak{s}$ and $-1.83 \times 10^2$ for $\mathfrak{r}$), reflecting differences of several orders of magnitude. These results confirm that the multispectral datasets exhibit both pronounced frequency sparsity and strong slice-wise low-rankness, further reinforcing the universality of dual spectral sparsity beyond natural videos and medical images.

Together, these results indicate that dual spectral sparsity, understood as energy concentration across slices together with pronounced low-rankness within slices, is not a coincidental pattern but a statistically stable property observed in many forms of real tensor data.

**Consistency of Dual Spectral Sparsity Across Modalities.** The same dual sparsity pattern appears in *natural videos, medical imagery (MRI and CT), hyperspectral cubes such as Salinas A and Indian Pines, and multispectral data*. Across all these modalities, transform-domain energy concentrates

Table 13: Dual spectral sparsity statistics across 31 multispectral images. $\mathfrak{s}(\mathbf{T})$ and $\mathfrak{r}(\mathbf{T})$ denote frequency sparsity and stable-rank ratios, compared against Gaussian random tensors with matched size and energy.

| Dataset | $\mathfrak{s}(\mathbf{T})$ | $\mathfrak{r}(\mathbf{T})$ | $\mathfrak{s}(\text{rand})$ | $\mathfrak{r}(\text{rand})$ |
|---|---|---|---|---|
| balloons | $3.24 \times 10^{-2}$ | $3.45 \times 10^{-3}$ | $9.90 \times 10^{-1}$ | $2.53 \times 10^{-1}$ |
| beads | $3.43 \times 10^{-2}$ | $1.11 \times 10^{-2}$ | $9.81 \times 10^{-1}$ | $2.54 \times 10^{-1}$ |
| cd | $3.30 \times 10^{-2}$ | $7.38 \times 10^{-3}$ | $9.86 \times 10^{-1}$ | $2.53 \times 10^{-1}$ |
| chart_toy | $3.23 \times 10^{-2}$ | $3.19 \times 10^{-3}$ | $9.87 \times 10^{-1}$ | $2.54 \times 10^{-1}$ |
| clay | $4.08 \times 10^{-2}$ | $3.15 \times 10^{-3}$ | $9.89 \times 10^{-1}$ | $2.54 \times 10^{-1}$ |
| cloth | $3.39 \times 10^{-2}$ | $5.20 \times 10^{-3}$ | $9.86 \times 10^{-1}$ | $2.53 \times 10^{-1}$ |
| egyptian_statue | $3.23 \times 10^{-2}$ | $3.64 \times 10^{-3}$ | $9.89 \times 10^{-1}$ | $2.53 \times 10^{-1}$ |
| face | $3.27 \times 10^{-2}$ | $5.25 \times 10^{-3}$ | $9.86 \times 10^{-1}$ | $2.54 \times 10^{-1}$ |
| fake_beers | $3.23 \times 10^{-2}$ | $3.17 \times 10^{-3}$ | $9.92 \times 10^{-1}$ | $2.53 \times 10^{-1}$ |
| fake_food | $3.49 \times 10^{-2}$ | $4.43 \times 10^{-3}$ | $9.87 \times 10^{-1}$ | $2.54 \times 10^{-1}$ |
| fake_lemonslices | $3.24 \times 10^{-2}$ | $6.22 \times 10^{-3}$ | $9.92 \times 10^{-1}$ | $2.52 \times 10^{-1}$ |
| fake_lemons | $3.28 \times 10^{-2}$ | $3.69 \times 10^{-3}$ | $9.93 \times 10^{-1}$ | $2.53 \times 10^{-1}$ |
| fake_peppers | $3.47 \times 10^{-2}$ | $3.43 \times 10^{-3}$ | $9.92 \times 10^{-1}$ | $2.54 \times 10^{-1}$ |
| fake_strawberries | $3.23 \times 10^{-2}$ | $3.57 \times 10^{-3}$ | $9.90 \times 10^{-1}$ | $2.53 \times 10^{-1}$ |
| fake_sushi | $3.27 \times 10^{-2}$ | $3.76 \times 10^{-3}$ | $9.87 \times 10^{-1}$ | $2.53 \times 10^{-1}$ |
| fake_tomatoes | $3.54 \times 10^{-2}$ | $5.49 \times 10^{-3}$ | $9.90 \times 10^{-1}$ | $2.53 \times 10^{-1}$ |
| feathers | $3.25 \times 10^{-2}$ | $3.20 \times 10^{-3}$ | $9.87 \times 10^{-1}$ | $2.53 \times 10^{-1}$ |
| flowers | $3.38 \times 10^{-2}$ | $3.83 \times 10^{-3}$ | $9.88 \times 10^{-1}$ | $2.55 \times 10^{-1}$ |
| glass_tiles | $3.40 \times 10^{-2}$ | $3.84 \times 10^{-3}$ | $9.94 \times 10^{-1}$ | $2.53 \times 10^{-1}$ |
| hairs | $3.24 \times 10^{-2}$ | $3.52 \times 10^{-3}$ | $9.92 \times 10^{-1}$ | $2.54 \times 10^{-1}$ |
| jelly_beans | $3.25 \times 10^{-2}$ | $8.68 \times 10^{-3}$ | $9.91 \times 10^{-1}$ | $2.53 \times 10^{-1}$ |
| oil_painting | $3.24 \times 10^{-2}$ | $4.27 \times 10^{-3}$ | $9.90 \times 10^{-1}$ | $2.53 \times 10^{-1}$ |
| paints | $3.23 \times 10^{-2}$ | $3.04 \times 10^{-3}$ | $9.88 \times 10^{-1}$ | $2.52 \times 10^{-1}$ |
| photo_face | $3.23 \times 10^{-2}$ | $6.02 \times 10^{-3}$ | $9.88 \times 10^{-1}$ | $2.54 \times 10^{-1}$ |
| pompoms | $3.44 \times 10^{-2}$ | $5.32 \times 10^{-3}$ | $9.87 \times 10^{-1}$ | $2.54 \times 10^{-1}$ |
| real_apples | $3.25 \times 10^{-2}$ | $3.73 \times 10^{-3}$ | $9.92 \times 10^{-1}$ | $2.53 \times 10^{-1}$ |
| real_peppers | $3.32 \times 10^{-2}$ | $3.58 \times 10^{-3}$ | $9.91 \times 10^{-1}$ | $2.53 \times 10^{-1}$ |
| sponges | $3.45 \times 10^{-2}$ | $2.51 \times 10^{-3}$ | $9.88 \times 10^{-1}$ | $2.55 \times 10^{-1}$ |
| stuffed_toys | $3.28 \times 10^{-2}$ | $3.42 \times 10^{-3}$ | $9.89 \times 10^{-1}$ | $2.54 \times 10^{-1}$ |
| superballs | $3.50 \times 10^{-2}$ | $4.35 \times 10^{-3}$ | $9.85 \times 10^{-1}$ | $2.53 \times 10^{-1}$ |
| thread_spools | $3.28 \times 10^{-2}$ | $3.97 \times 10^{-3}$ | $9.89 \times 10^{-1}$ | $2.53 \times 10^{-1}$ |

Table 14: Hypothesis testing results for dual spectral sparsity across 22 YUV video tensors. $\mathfrak{s}(\mathbf{T})$ measures inter-frequency sparsity, and $\mathfrak{r}(\mathbf{T})$ measures intra-frequency low-rankness. Both metrics for real tensors are compared against size-matched Gaussian random tensors.

| Metric | Wilcoxon $p$-value | KS $p$-value | Cohen's $d$ |
|---|---|---|---|
| Inter-frequency sparsity $\mathfrak{s}(\mathbf{T})$ | $< 10^{-6}$ | $< 10^{-6}$ | $-91.84$ |
| Intra-frequency low-rankness $\mathfrak{r}(\mathbf{T})$ | $< 10^{-6}$ | $< 10^{-6}$ | $-13.88$ |

in a small subset of frequency slices, the active slices show rapid singular value decay that reflects strong low-rankness, and the statistics of real data differ clearly and consistently from random-tensor baselines.

*These observations indicate that dual spectral sparsity is a statistically stable structural feature present in many real tensor modalities rather than a dataset-specific artifact.*

Table 15: Hypothesis testing results for dual spectral sparsity across 31 multispectral images. The metrics $\mathfrak{s}(\mathbf{T})$ and $\mathfrak{r}(\mathbf{T})$ denote inter-frequency sparsity and intra-frequency stable-rank ratios, compared against Gaussian random tensors with matched size and energy.

| Metric | Wilcoxon $p$-value | KS $p$-value | Cohen's $d$ |
|---|---|---|---|
| Inter-frequency sparsity $\mathfrak{s}(\mathbf{T})$ | $< 10^{-6}$ | $< 10^{-6}$ | $-4.26 \times 10^2$ |
| Intra-frequency low-rankness $\mathfrak{r}(\mathbf{T})$ | $< 10^{-6}$ | $< 10^{-6}$ | $-1.83 \times 10^2$ |

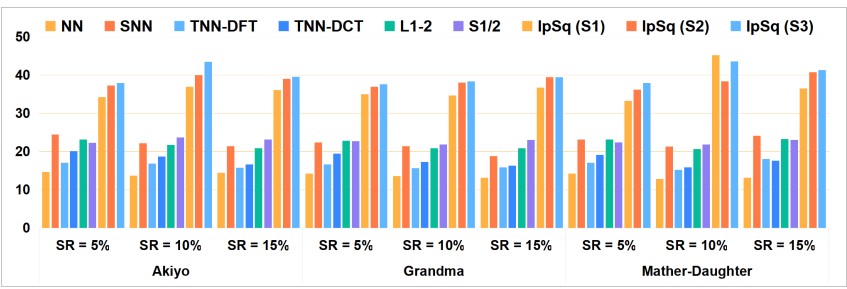

Figure 7: Runtime (in seconds) of different tensor completion methods on three YUV video sequences (*Akiyo*, *Grandma*, and *Mother-Daughter*) under varying sampling rates (5%, 10%, 15%).

### D.11    EXTENDED EXPERIMENTS TO MORE 3D TENSOR DATA

#### D.11.1    EXTENDED EXPERIMENTS ON MORE VISUAL DATA: YUV VIDEOS

We further evaluate the proposed $\ell_p(S_q)$ regularizer on three YUV video sequences (*Akiyo*, *Grandma*, and *Mother-Daughter*), each converted to a $144 \times 176 \times 30$ tensor using only the luminance (Y) channel. These sequences exhibit diverse motion and spatiotemporal complexity, making them suitable for benchmarking real-world completion performance.

**Recovery Performance.**    As summarized in Table 16, classical low-rank models such as NN, SNN, and TNN-based variants provide reasonable reconstructions but tend to saturate as the sampling ratio increases. In contrast, the proposed $\ell_p(S_q)$ configurations (S1–S3) deliver consistently improved recovery quality, particularly in the higher sampling regimes. This behavior is aligned with our observations on other modalities and suggests that the dual spectral sparsity captured by $\ell_p(S_q)$ remains effective for YUV video data, where the frequency content is more structured yet still exhibits significant variability across frames.

**Runtime Efficiency.**    To further assess the practical efficiency, we also compare the runtime of competing methods under varying sampling rates. As shown in Figure 7, the proposed $\ell_p(S_q)$ variants (S1–S3) incur approximately twice the runtime of the TNN-DCT method across all sampling ratios and datasets. Nonetheless, the runtime remains within the same order of magnitude as TNN-based methods. This observation is consistent with our theoretical complexity analysis. Moreover, given the noticeable gains in reconstruction accuracy, particularly in low-sampling regimes, we regard the modest computational overhead as a reasonable and practically acceptable tradeoff.

#### D.11.2    EXTENDED EXPERIMENTS ON NON-VISUAL DATA: SEISMIC DATA

To evaluate the generality of the proposed $\ell_p(S_q)$ regularizer beyond visual tensors, we further evaluate the proposed $\ell_p(S_q)$ regularizer on a *seismic* dataset[9], a representative example of non-visual tensor modalities. Due to computational constraints, we extract the first 30 frontal slices from *original_data_128* (depth 64), yielding a tensor of size $64 \times 128 \times 30$. Seismic data exhibit oscillatory and heterogeneous spectral patterns that differ markedly from the structured textures found in visual datasets. As shown in Table 17, the proposed $\ell_p(S_q)$ family achieves stable and high reconstruction quality across different sampling ratios on seismic tensors. The trend is consistent with what we

---

[9]The data is available at http://pan.baidu.com/s/1qYwI1IG.

Table 16: Noisy tensor completion results on three typical video datasets. The proposed $\ell_p(S_q)$ is reported under four parameter choices: Setting **S1** $(p,q) = (0.60, 0.61)$, Setting **S2** $(p,q) = (0.70, 0.71)$, and Setting **S3** $(p,q) = (0.80, 0.81)$.

| Dataset | SR | Metric | NN | SNN | TNN-DFT | TNN-DCT | $\ell_{1-2}$ | $S_{1/2}$ | S1 | S2 | S3 |
|---|---|---|---|---|---|---|---|---|---|---|---|
| *Akiyo* | 5% | PSNR | 15.37 | 19.49 | 27.04 | 26.58 | 27.00 | 26.89 | 27.63 | 27.97 | **28.09** |
| | | SSIM | 0.1864 | 0.6047 | 0.8019 | **0.8233** | 0.7974 | 0.7313 | 0.7715 | 0.7845 | 0.7875 |
| | 10% | PSNR | 18.01 | 22.54 | 29.18 | 28.91 | 29.39 | 28.95 | 30.53 | 30.57 | **30.59** |
| | | SSIM | 0.2858 | 0.7186 | 0.8556 | **0.8899** | 0.8576 | 0.7853 | 0.8625 | 0.8631 | 0.8580 |
| | 15% | PSNR | 19.64 | 24.37 | 30.60 | 29.96 | 30.70 | 29.98 | 32.30 | 32.33 | **32.39** |
| | | SSIM | 0.3694 | 0.7812 | 0.8791 | **0.9133** | 0.8803 | 0.8011 | 0.8973 | 0.8963 | 0.8945 |
| *Grandma* | 5% | PSNR | 16.52 | 19.13 | 28.53 | 28.26 | 28.68 | 29.12 | 29.96 | 30.01 | **30.14** |
| | | SSIM | 0.1928 | 0.5503 | 0.8135 | 0.8217 | 0.8134 | 0.7874 | 0.8205 | 0.8259 | **0.8263** |
| | 10% | PSNR | 18.34 | 22.52 | 31.44 | 31.15 | 31.67 | 31.77 | 33.50 | 33.56 | **33.68** |
| | | SSIM | 0.2992 | 0.6755 | 0.8822 | 0.8932 | 0.8852 | 0.8563 | 0.9031 | **0.9041** | 0.9037 |
| | 15% | PSNR | 19.66 | 24.88 | 33.02 | 32.55 | 33.23 | 32.95 | 35.43 | 35.49 | **35.52** |
| | | SSIM | 0.3757 | 0.7525 | 0.9088 | 0.9164 | 0.9097 | 0.8784 | **0.9332** | 0.9327 | 0.9307 |
| *Mother-Daughter* | 5% | PSNR | 16.17 | 20.22 | 27.66 | 26.85 | 27.51 | 27.28 | 28.18 | 28.45 | **28.60** |
| | | SSIM | 0.1895 | 0.5824 | 0.7491 | **0.7670** | 0.7441 | 0.6742 | 0.7317 | 0.7445 | 0.7504 |
| | 10% | PSNR | 18.55 | 23.48 | 29.38 | 28.76 | 29.56 | 28.34 | 30.62 | **30.66** | 30.52 |
| | | SSIM | 0.2780 | 0.6840 | 0.8035 | **0.8327** | 0.8058 | 0.7067 | 0.8212 | 0.8226 | 0.8135 |
| | 15% | PSNR | 20.32 | 25.31 | 30.49 | 29.56 | 30.54 | 28.80 | 31.62 | **31.69** | 31.65 |
| | | SSIM | 0.3548 | 0.7405 | 0.8293 | **0.8554** | 0.8305 | 0.7188 | 0.8474 | 0.8504 | 0.8457 |

Table 17: Noisy tensor completion results on the *Seismic* dataset.

| Method | SR = 5% | | SR = 10% | | SR = 15% | |
|---|---|---|---|---|---|---|
| | PSNR | SSIM | PSNR | SSIM | PSNR | SSIM |
| TNN-DFT | 24.46 | 0.8830 | 27.64 | 0.9293 | 30.75 | 0.9558 |
| TNN-DCT | 32.31 | 0.9688 | 37.99 | 0.9878 | 41.25 | 0.9930 |
| $k$-Supp | 24.32 | 0.8785 | 27.72 | 0.9334 | 30.72 | 0.9521 |
| $\ell_{1-2}$ | 23.22 | 0.8563 | 28.38 | 0.9355 | 31.32 | 0.9583 |
| $S_{1/2}$ | 36.72 | 0.9783 | 41.20 | 0.9895 | 43.64 | 0.9935 |
| $\ell_p(S_q)$ (0.50,0.51) | 36.30 | 0.9776 | 42.15 | 0.9912 | 44.72 | 0.9948 |
| $\ell_p(S_q)$ (0.60,0.61) | 36.87 | 0.9765 | 41.99 | 0.9909 | 44.67 | 0.9943 |
| $\ell_p(S_q)$ (0.70,0.71) | 36.33 | 0.9779 | 42.16 | 0.9912 | 44.71 | 0.9942 |
| $\ell_p(S_q)$ (0.80,0.81) | 37.65 | 0.9821 | **43.35** | **0.9922** | 45.74 | **0.9953** |
| $\ell_p(S_q)$ (0.90,0.91) | 37.75 | **0.9830** | 43.04 | 0.9921 | 45.64 | 0.9952 |
| $\ell_p(S_q)$ (0.94,0.95) | **38.46** | 0.9828 | 43.15 | 0.9917 | **46.06** | 0.9951 |

observe on imaging data and suggests that the dual spectral structure captured by $\ell_p(S_q)$ naturally appears in non-visual tensor fields with richer spectral variability.

### D.12 EXTENDED EXPERIMENTS TO HIGHER-ORDER TENSORS

Our regularizer extends directly to higher-order tensors as long as a suitable transform and slice level singular values are well defined. Under this basic condition, the dual spectral sparsity principle carries over naturally to arbitrary tensor orders. This section outlines such an extension and provides preliminary evidence showing that the same modeling behavior persists beyond the third-order setting.

**A Strategy to Extend to Higher-Order Tensors.** For higher-order tensors with order $K > 3$, a natural extension of the proposed $\ell_p(S_q)$ regularizer is to generalize the spectral shrinkage scheme to multiple orientations. Following the spirit of orientation-invariant TNN (Wang et al., 2023b), one can first define $K$ distinct 3-way tensor unfoldings $\underline{\mathbf{T}}_{[k]} := \mathfrak{F}_k(\underline{\mathbf{T}})$ via mode-$(k, k+1)$ 3D-unfolding, each capturing the spectral structure along a different orientation. Then, the $\ell_p(S_q)$ regularization can be applied independently to each $\underline{\mathbf{T}}_{[k]}$ in the transformed domain (e.g., DCT or DFT), and the

total regularizer becomes a weighted sum across all orientations:

$$\mathcal{R}(\underline{\mathbf{T}}) := \sum_{k=1}^{K} \omega_k \|\mathcal{S}_q(\mathfrak{F}_k(\underline{\mathbf{T}}))\|_p^p,$$

where $\omega_k$ are positive weights satisfying $\sum_k \omega_k = 1$. We call this Orientation-Invariant $\ell_p(S_q)$ (OI-$\ell_p(S_q)$). This formulation allows $\ell_p(S_q)$ to capture joint spectral sparsity patterns across multiple dimensions, enhancing the model's flexibility and robustness for high-dimensional data. In practice, a simple yet effective choice is to set all weights equally and restrict to circular unfolding with $k = 1, \ldots, K$.

**Preliminary Results.** We further assess the proposed model on a fourth-order tensor derived from the *Ground* dataset[10], commonly used in small object detection for aerial surveillance. The input consists of the first 30 frames, each of size $216 \times 288$ with 3 RGB channels, forming a $216 \times 288 \times 3 \times 30$ tensor.

To benchmark against matrix-based baselines, NN is applied along each unfolding mode of the tensor; the best result across all four modes is reported. Since TNN-based methods (TNN-DCT and TNN-DFT) are tailored to third-order inputs, we extract three grayscale videos corresponding to each color channel and apply these methods independently to each $216 \times 288 \times 30$ slice. For SqNN (Mu et al., 2014), we adopt the mode partition $\{1, 4\}$ vs. $\{2, 3\}$ and set $\lambda = \lambda_\iota \sigma \sqrt{p d_1 d_4 \log(d_1 d_4 + d_2 d_3)}$. SNN (Tomioka et al., 2011) uses weights $(1, 1, 0.01, 1)$ with global tuning. For OITNN and the proposed OI-$\ell_p(S_q)$, we adopt weights 1:100:1:1 and set $\lambda = \lambda_\iota \sigma \sqrt{p d_1 d_3 d_4 \log(d_1 d_3 d_4 + d_2 d_3)}$. We evaluate the methods under two sampling ratios ($p = 5\%, 10\%$).

Table 18 shows that the OI-$\ell_p(S_q)$ variant attains the strongest PSNR and SSIM at both sampling ratios on the *Ground* data, indicating that dual spectral sparsity remains effective in higher-order settings.

Table 18: The PSNR and SSIM values obtained by seven tensor norms (NN (Klopp et al., 2014), SqNN (Mu et al., 2014), SNN (Tomioka et al., 2011), TNN-DFT (Wang et al., 2019), TNN-DCT (Lu et al., 2019b), OITNN-O (Wang et al., 2023b), and the proposed OI-$\ell_p(S_q)$ with $(p, q) = (0.50, 0.51)$ for noisy tensor completion on the 4D *Ground* data.

| SR | | NN | SqNN | SNN | TNN-DFT | TNN-DCT | OITNN-O | OI-$\ell_p(S_q)$ |
|----|------|-------|--------|--------|---------|---------|---------|------------------|
| 5% | PSNR | 24.83 | 26.80 | 25.03 | 30.90 | 30.94 | 32.85 | **34.35** |
|    | SSIM | 0.7697 | 0.8011 | 0.8055 | 0.8955 | 0.8997 | 0.9306 | **0.9512** |
| 10% | PSNR | 27.03 | 29.10 | 27.36 | 32.27 | 32.49 | 34.25 | **36.47** |
|     | SSIM | 0.8395 | 0.8725 | 0.8559 | 0.9083 | 0.9149 | 0.9382 | **0.9669** |

### D.13  EXTENDED EXPERIMENTS TO POISSON TENSOR COMPLETION

We next extend our framework to the Poisson tensor completion setting, which arises naturally in photon-limited imaging and other count-based sensing modalities. Following the masked Poisson observation model introduced by Zhang & Ng (2021), each observed entry is generated through a signal-dependent Poisson process, leading to heteroscedastic noise and significant challenges in low-intensity regions.

We adopt the Bi-Module Tensor Regularization (BTR) framework of Wang et al. (2025a) and replace the quadratic fidelity term with the Poisson negative log-likelihood. The $\ell_p(S_q)$ spectral regularizer is then applied to both bi-module unfoldings, yielding a Bi–$\ell_p(S_q)$ formulation that preserves bidirectional spectral structure while remaining compatible with count-based measurements.

For experimentation, we follow the non-uniform sampling protocol of Wang et al. (2025a), in which structured dropout is applied along all tensor modes to obtain sampling ratios $\mathbf{p} \in \{20\%, 30\%\}$. Given a ground-truth tensor $\underline{\mathbf{X}}^\star$, we normalize its dynamic range to $\|\underline{\mathbf{X}}^\star\|_\infty = 100$, set the background constant to $c = 5$, and generate Poisson measurements $\underline{\mathbf{Y}}$ via

$$\underline{\mathbf{P}}_{ijk} \sim \mathfrak{P}(\underline{\mathbf{X}}^\star_{ijk} + c), \quad \underline{\mathbf{Y}} = \underline{\mathbf{B}} \circledast \underline{\mathbf{P}},$$

---

[10]http://www.loujing.com/rss-small-target

where $\mathbf{B}$ is the binary mask tensor, and $\circledast$ denotes element-wise product. We evaluate the resulting Bi$-\ell_p(S_q)$ model on hyperspectral, multispectral, LiDAR, and infrared tensors, providing a diverse testbed for assessing robustness under Poisson corruption and highly irregular sampling.

Table 19: Poisson tensor completion results under non-uniform sampling (20% and 30%). The baselines include PoissonNN (Cao & Xie, 2015), SNN/HaLRTC (Liu et al., 2013), TMac (Xu et al., 2015), and TNN-DFT/DCT (Zhang & Ng, 2021). We additionally compare with the two Bi-Module regularizers BTR-DFT and BTR-DCT from Wang et al. (2025a), abbreviated as BTR-F and BTR-C in the table, respectively.

| Dataset | SR | Metric | PoissonNN | SNN | TMac | TNN-F | TNN-C | BTR-F | BTR-C | Bi-$\ell_p(S_q)$ |
|---|---|---|---|---|---|---|---|---|---|---|
| *Indian Pines* | 20% | PSNR | 7.77 | 23.07 | 24.21 | 24.65 | 25.90 | 28.73 | 28.44 | **30.05** |
| | | SSIM | 0.0089 | 0.4081 | 0.5481 | 0.4068 | 0.5044 | 0.7202 | 0.7151 | **0.7358** |
| | 30% | PSNR | 8.54 | 22.91 | 24.65 | 24.82 | 26.59 | 29.38 | 29.65 | **31.15** |
| | | SSIM | 0.0169 | 0.4163 | 0.5819 | 0.4221 | 0.5313 | 0.7273 | 0.7620 | **0.7881** |
| *SalinasA* | 20% | PSNR | 7.62 | 23.13 | 25.01 | 21.74 | 25.75 | 28.72 | 29.73 | **32.28** |
| | | SSIM | 0.0340 | 0.4511 | 0.6725 | 0.3637 | 0.5516 | 0.7589 | 0.7671 | **0.7839** |
| | 30% | PSNR | 8.40 | 23.17 | 25.39 | 22.28 | 26.92 | 30.48 | 31.48 | **34.55** |
| | | SSIM | 0.0487 | 0.4414 | 0.6888 | 0.3857 | 0.5963 | 0.8076 | 0.8161 | **0.8695** |
| *Cloth* | 20% | PSNR | 13.36 | 21.20 | 21.11 | 22.93 | 24.57 | 25.71 | 25.96 | **27.97** |
| | | SSIM | 0.0905 | 0.4450 | 0.4174 | 0.5686 | 0.6299 | 0.7212 | 0.7521 | **0.8005** |
| | 30% | PSNR | 13.36 | 21.50 | 21.83 | 23.36 | 25.24 | 26.63 | 27.16 | **28.51** |
| | | SSIM | 0.1451 | 0.5290 | 0.4915 | 0.6544 | 0.7261 | 0.7548 | 0.7912 | **0.8088** |
| *Oilpainting* | 20% | PSNR | 15.11 | 23.33 | 22.58 | 27.52 | 28.40 | 28.96 | 29.72 | **32.04** |
| | | SSIM | 0.0677 | 0.3391 | 0.3528 | 0.4211 | 0.4656 | 0.5538 | **0.6384** | 0.6262 |
| | 30% | PSNR | 13.95 | 22.30 | 21.38 | 26.33 | 27.23 | 29.57 | 30.66 | **33.07** |
| | | SSIM | 0.1357 | 0.4580 | 0.4525 | 0.5717 | 0.6240 | 0.5525 | 0.6334 | **0.6471** |
| *SenerioBDis* | 20% | PSNR | 13.24 | 17.61 | 18.54 | 20.11 | 20.05 | 20.18 | 20.13 | **21.53** |
| | | SSIM | 0.0767 | 0.2180 | 0.2285 | 0.3550 | 0.3907 | 0.4125 | **0.4492** | 0.4439 |
| | 30% | PSNR | 14.08 | 18.75 | 18.89 | 21.40 | 21.41 | 21.12 | 21.09 | **22.58** |
| | | SSIM | 0.1234 | 0.2628 | 0.2526 | 0.3848 | 0.4207 | 0.4309 | **0.4668** | 0.4572 |
| *SenerioBIntens* | 20% | PSNR | 14.19 | 18.06 | 17.76 | 20.45 | 20.47 | 21.06 | 20.65 | **21.80** |
| | | SSIM | 0.1205 | 0.3269 | 0.2688 | 0.4690 | 0.4769 | 0.5306 | 0.5106 | **0.5560** |
| | 30% | PSNR | 14.89 | 19.01 | 18.10 | 21.36 | 21.41 | 21.93 | 21.58 | **23.14** |
| | | SSIM | 0.1959 | 0.3998 | 0.2967 | 0.5279 | 0.5369 | 0.5664 | 0.5594 | **0.6041** |
| *Thermal* | 20% | PSNR | 10.53 | 19.29 | 15.46 | 25.66 | 26.01 | 26.08 | 26.50 | **29.38** |
| | | SSIM | 0.0724 | 0.3806 | 0.2096 | 0.5788 | 0.6195 | 0.6571 | **0.7083** | 0.7056 |
| | 30% | PSNR | 11.29 | 20.64 | 15.83 | 26.59 | 27.42 | 27.23 | 27.88 | **30.46** |
| | | SSIM | 0.1142 | 0.4391 | 0.2447 | 0.5918 | 0.6392 | 0.6639 | **0.7185** | 0.7039 |

Table 19 summarizes the performance of Bi-$\ell_p(S_q)$ under non-uniform Poisson sampling. We compare against a broad set of representative baselines, including matrix-based PoissonNN (Cao & Xie, 2015), Tucker-based HaLRTC/SNN (Liu et al., 2013), factorization-based TMac (Xu et al., 2015), and transform-based TNN-DFT/DCT (Zhang & Ng, 2021). For completeness, we also include the recent bi-module tensor regularizers BTR-DFT and BTR-DCT proposed by Wang et al. Wang et al. (2025a). Across the datasets and sampling ratios, Bi-$\ell_p(S_q)$ consistently achieves the best or near-best PSNR and SSIM, outperforming both classical low-rank priors and modern transform-based tensor models. These results demonstrate that Bi-$\ell_p(S_q)$ provides a robust and adaptable prior for Poisson tensor completion across diverse sensing modalities and noise conditions.

## D.14 EXTENDED EXPERIMENTS TO CLUSTERING

To examine whether the proposed $\ell_p(S_q)$ prior remains effective beyond completion tasks, we further apply it to *unsupervised image clustering*. In this setting, the goal is to recover the underlying identity or digit category by exploiting structural correlations across samples. Following the self-representation paradigm widely used in subspace clustering, each image is expressed as a combination

of the remaining images, while the coefficient tensor is regularized through the bidirectional t-linear structure imposed by BTR (Wang et al., 2025a). Embedding $\ell_p(S_q)$ into this framework yields a nonconvex spectral shrinkage along both t-modes, producing a more discriminative affinity matrix for clustering.

*Datasets.* We use four benchmark datasets with varying numbers of classes and resolutions:

- *FRDUE*: 3040 facial images from 152 individuals, each of size $25 \times 22$.

- *FRDUE-100*: A subset of FRDUE with 2500 images from 100 subjects, same resolution.

- *PIE-10*: 680 images of 10 people under varying lighting/pose, size $22 \times 22$.

- *USPS1000*: 1000 handwritten digits from USPS dataset, 10 classes, $16 \times 16$ resolution.

*Data Tensor Formation.* All images are normalized to $[0, 1]$ and stacked into a third-order tensor $\underline{\mathbf{X}} \in \mathbb{R}^{h \times n \times w}$, where each lateral slice $\underline{\mathbf{X}}_{:,i,:}$ corresponds to one image. No pretrained features or handcrafted descriptors are used, allowing the clustering performance to reflect purely the modeling ability of the tensor regularizers.

Table 20: Clustering performance comparison on four benchmark datasets (*FRDUE*, *FRDUE-100*, *PIE-10*, and *USPS1000*) in terms of accuracy (ACC), normalized mutual information (NMI), and purity (PUR).

| Dataset | Metric | R-TPCA | | OR-TPCA | R-TLRR | | OR-TLRR | | BTR | | Bi-$\ell_p(S_q)$ | |
|---|---|---|---|---|---|---|---|---|---|---|---|---|
| | | DFT | DCT | DFT | DFT | DCT | DFT | DCT | DFT | DCT | DFT | DCT |
| *FRDUE* | ACC | 0.761 | 0.778 | 0.764 | 0.843 | 0.840 | 0.836 | 0.739 | 0.859 | 0.859 | **0.862** | 0.856 |
| | NMI | 0.909 | 0.914 | 0.913 | 0.951 | 0.951 | 0.947 | 0.904 | 0.958 | 0.957 | **0.960** | 0.956 |
| | PUR | 0.796 | 0.808 | 0.799 | 0.876 | 0.872 | 0.864 | 0.778 | 0.890 | 0.889 | **0.892** | 0.887 |
| *FRDUE-100* | ACC | 0.791 | 0.791 | 0.797 | 0.861 | 0.860 | 0.866 | 0.762 | 0.886 | 0.877 | **0.896** | 0.885 |
| | NMI | 0.913 | 0.914 | 0.922 | 0.953 | 0.953 | 0.956 | 0.906 | 0.964 | 0.961 | **0.966** | 0.963 |
| | PUR | 0.820 | 0.820 | 0.827 | 0.888 | 0.888 | 0.893 | 0.796 | 0.913 | 0.903 | **0.920** | 0.909 |
| *PIE-10* | ACC | 0.428 | 0.427 | 0.540 | 0.590 | 0.583 | 0.459 | 0.198 | 0.600 | 0.611 | 0.603 | **0.638** |
| | NMI | 0.667 | 0.662 | 0.736 | 0.756 | 0.762 | 0.705 | 0.515 | 0.770 | 0.778 | 0.771 | **0.781** |
| | PUR | 0.447 | 0.446 | 0.559 | 0.606 | 0.600 | 0.480 | 0.205 | 0.619 | 0.633 | 0.622 | **0.656** |
| *USPS1000* | ACC | 0.355 | 0.354 | 0.337 | 0.426 | 0.409 | 0.519 | 0.450 | 0.580 | 0.534 | **0.589** | 0.524 |
| | NMI | 0.307 | 0.299 | 0.283 | 0.383 | 0.392 | 0.510 | 0.449 | 0.586 | 0.539 | **0.591** | 0.534 |
| | PUR | 0.447 | 0.454 | 0.443 | 0.526 | 0.508 | 0.633 | 0.591 | 0.686 | 0.666 | **0.687** | 0.665 |

*Model Configuration.* The dictionary is fixed as $\underline{\mathbf{D}} = \underline{\mathbf{X}}$, enforcing a self-expressive relation. BTR regularization weight $\alpha$ is tuned over $\{0.1, 0.5, 0.9\}$. The regularization parameter $\lambda$ is tuned over $\{0.1, 0.5, 1, 2, 5\}$ based on validation. ADMM is used to solve the optimization problem.

*Clustering Protocol.* After optimization, the coefficient tensors from the two t-modules are symmetrized and aggregated to form an affinity matrix $\hat{\mathbf{Z}}$. Spectral clustering with normalized cuts is then applied to obtain cluster assignments. Accuracy (ACC), normalized mutual information (NMI), and purity (PUR) are reported. Each experiment is repeated ten times to mitigate the effect of random initialization. The results, summarized in Table 20, demonstrate that incorporating the nonconvex $\ell_p(S_q)$ regularizer into BTR consistently enhances the discriminability of the learned affinities across all datasets.

### D.15 EXTENDED ANALYSIS OF PARAMETER SENSITIVITY OF $\ell_p(S_q)$

**Why is selecting $(p, q)$ fundamentally difficult?** A natural question is how the parameters $(p, q)$ should be chosen in the proposed $\ell_p(S_q)$ family. These parameters are not ordinary tunable coefficients but structural modeling variables. The parameter $p$ governs sparsity across frequency slices, and $q$ determines low-rankness within each slice. Together they define the inductive bias of the regularizer, and their effects cannot be substituted through simple rescaling.

From a theoretical perspective, the situation is even more restrictive. In vector and matrix settings, nonconvex regularizations such as $\ell_p$ or half-order penalties already lack reliable selection rules. In tensor settings the coupling between inter-frequency sparsity and intra-frequency low-rankness

introduces additional layers of complexity, and current theory does not provide actionable guidance for choosing $(p, q)$. This motivates an empirical investigation to understand whether certain parameter regions behave consistently across modalities.

**Experimental design.** To study the sensitivity of $\ell_p(S_q)$ to $(p, q)$, we evaluate three tensor modalities with distinct spectral characteristics: hyperspectral data (*SalinasA*), video sequences (*Grandma*), and seismic volumes (*Seismic*). For each dataset we form tensors using the first 50, 80, and 120 frontal slices. Increasing the number of slices introduces more frequency components and raises spectral complexity, which helps us evaluate the stability of $\ell_p(S_q)$ under varying tensor sizes.

For each tensor instance we sweep a two-dimensional parameter grid with $p \in \{0.5, 0.6, 0.7, 0.8, 0.9\}$ and $q = p + \Delta q$, where $\Delta q$ spans eight decreasing offsets down to 0. All configurations are evaluated with a fixed sampling ratio of ten percent and a noise level of five percent. Reconstruction quality is measured by PSNR and SSIM.

**Observed trends across datasets.** The quantitative results in Tables 21 -23 reveal consistent trends across all datasets and frequency settings.

- *Effect of small $p$:* Settings with $p = 0.5$ consistently yield lower PSNR across all three modalities. This pattern is visible in the *SalinasA*, *Grandma*, and *Seismic* results. A likely explanation is that very small $p$ imposes overly aggressive inter-frequency shrinkage, which suppresses informative frequency components and leads to under-representation of useful structure.

- *Effect of large $q$ (large $q - p$):* Increasing $q$ beyond a moderate range yields diminishing or slightly reduced SSIM. This behavior appears in the 80-frame *Grandma* results and similarly in the *SalinasA* and *Seismic* experiments, suggesting that excessive intra-slice shrinkage suppresses structural detail.

- *High-performing region:* The highest and most stable PSNR and SSIM values occur when $p$ lies between 0.7 and 0.9 and $q$ slightly exceeds $p$. This pattern is consistent across all datasets. For instance, in the 120-frame *Seismic* results the best PSNR appears near $(p, q - p) \approx (0.8, 0.01)$, with similar behavior observed in *SalinasA* and *Grandma*.

Overall, the tables show that neighboring $(p, q)$ configurations often achieve comparable reconstruction quality, and that this phenomenon appears consistently across modalities and frequency numbers. The performance landscape of $\ell_p(S_q)$ is therefore smooth rather than sharply peaked in the tested range, indicating that the method does not depend on fine-grained parameter tuning.

The heatmaps in Figures 8-10 further support these observations. In the low-frequency-number regime (50 slices), the PSNR surface shows a clear ridge around $\Delta q$ between 0.01 and 0.015, with a visible peak near $(p, q) \approx (0.8, 0.81)$. As the number of slices increases to 80 and 120, the sensitivity decreases and the high-performing region expands, indicating that richer spectral information stabilizes the behavior of the regularizer.

Table 21: Sensitivity analysis of $\ell_p(S_q)$ on the *SalinasA* dataset under different frequency counts (i.e., using the first 50, 80, and 120 frames). Each sub-table corresponds to a fixed frequency count, with rows indexed by $p$ and columns indexed by $q - p$. Each entry reports PSNR/SSIM.

| # Freq. | $p$ | $q - p = 0$ | 0.0012 | 0.0025 | 0.005 | 0.01 | 0.015 | 0.02 | 0.025 | 0.03 |
|---|---|---|---|---|---|---|---|---|---|---|
| | 0.5 | 30.81/0.8128 | 30.89/0.8128 | 30.98/0.8130 | 30.99/0.8135 | 31.13/0.8138 | 31.13/0.8144 | 31.11/0.8174 | 31.02/0.8186 | 30.83/0.8165 |
| | 0.6 | 31.14/0.8235 | 31.14/0.8235 | 31.18/0.8243 | 31.17/0.8234 | 31.15/0.8212 | 31.25/0.8238 | 30.95/0.8125 | 31.11/0.8136 | 30.64/0.7894 |
| 50 | 0.7 | 31.20/0.8270 | 31.23/0.8275 | 31.22/0.8267 | 31.21/0.8256 | 31.32/0.8275 | 31.42/0.8281 | 31.35/0.8224 | 31.26/0.8159 | 30.51/0.7783 |
| | 0.8 | 31.42/0.8335 | 31.43/0.8337 | 31.48/0.8343 | 31.54/0.8348 | 31.52/0.8315 | 31.43/0.8273 | 31.35/0.8200 | 31.10/0.8052 | 30.54/0.7767 |
| | 0.9 | 31.56/0.8366 | 31.59/0.8378 | 31.57/0.8362 | 31.60/0.8367 | 31.57/0.8322 | 31.45/0.8248 | 31.30/0.8161 | 31.10/0.8055 | 30.95/0.7971 |
| | 0.5 | 31.62/0.8401 | 31.65/0.8412 | 31.68/0.8420 | 31.75/0.8409 | 31.62/0.8344 | 31.67/0.8360 | 31.78/0.8404 | 31.68/0.8428 | 31.53/0.8373 |
| | 0.6 | 31.87/0.8511 | 31.87/0.8499 | 31.90/0.8518 | 31.98/0.8528 | 32.01/0.8510 | 32.00/0.8513 | 31.85/0.8452 | 31.83/0.8400 | 31.36/0.8248 |
| 80 | 0.7 | 31.97/0.8545 | 32.05/0.8558 | 32.05/0.8553 | 32.06/0.8544 | 32.11/0.8541 | 32.14/0.8526 | 32.13/0.8488 | 32.02/0.8422 | 30.99/0.8012 |
| | 0.8 | 32.25/0.8614 | 32.26/0.8610 | 32.25/0.8603 | 32.26/0.8598 | 32.29/0.8569 | 32.20/0.8512 | 32.02/0.8419 | 31.58/0.8230 | 30.91/0.7950 |
| | 0.9 | 32.40/0.8636 | 32.41/0.8630 | 32.41/0.8625 | 32.38/0.8607 | 32.32/0.8559 | 32.18/0.8481 | 31.93/0.8379 | 31.72/0.8295 | 31.55/0.8227 |
| | 0.5 | 33.19/0.8730 | 33.26/0.8740 | 33.34/0.8756 | 33.44/0.8761 | 33.63/0.8785 | 33.70/0.8795 | 33.70/0.8804 | 33.52/0.8795 | 33.20/0.8745 |
| | 0.6 | 33.65/0.8840 | 33.62/0.8835 | 33.68/0.8839 | 33.75/0.8854 | 33.78/0.8857 | 33.84/0.8858 | 33.83/0.8853 | 33.61/0.8772 | 33.79/0.8796 |
| 120 | 0.7 | 33.82/0.8884 | 33.83/0.8884 | 33.86/0.8886 | 33.92/0.8891 | 34.00/0.8894 | 34.13/0.8908 | 34.20/0.8905 | 34.21/0.8889 | 33.85/0.8768 |
| | 0.8 | 34.03/0.8928 | 34.07/0.8931 | 34.08/0.8933 | 34.13/0.8934 | 34.19/0.8932 | 34.22/0.8914 | 34.23/0.8892 | 34.13/0.8843 | 33.49/0.8641 |
| | 0.9 | 34.29/0.8976 | 34.32/0.8975 | 34.34/0.8975 | 34.37/0.8968 | 34.36/0.8949 | 34.34/0.8930 | 34.26/0.8899 | 34.21/0.8878 | 34.11/0.8842 |

Table 22: Sensitivity analysis of $\ell_p(S_q)$ on the *grandma* dataset under different frequency counts (i.e., using the first 50, 80, and 120 frames). Each sub-table corresponds to a fixed frequency count, with rows indexed by $p$ and columns indexed by $q - p$. Each entry reports PSNR/SSIM.

| # Freq. | $p$ | $q-p=0$ | 0.0012 | 0.0025 | 0.005 | 0.01 | 0.015 | 0.02 | 0.025 | 0.03 |
|---|---|---|---|---|---|---|---|---|---|---|
| | 0.5 | 32.83/0.9353 | 32.90/0.9354 | 33.03/0.9378 | 33.07/0.9391 | 33.17/0.9404 | 33.27/0.9425 | 33.24/0.9411 | 33.26/0.9433 | 33.03/0.9377 |
| | 0.6 | 33.31/0.9446 | 33.32/0.9444 | 33.33/0.9440 | 33.32/0.9434 | 33.38/0.9439 | 33.36/0.9432 | 33.28/0.9401 | 33.22/0.9382 | 32.79/0.9231 |
| 50 | 0.7 | 33.39/0.9440 | 33.38/0.9441 | 33.37/0.9438 | 33.43/0.9446 | 33.56/0.9448 | 33.55/0.9412 | 33.38/0.9348 | 33.02/0.9239 | 32.01/0.8977 |
| | 0.8 | 33.61/0.9463 | 33.66/0.9468 | 33.65/0.9461 | 33.70/0.9458 | 33.72/0.9440 | 33.62/0.9398 | 33.29/0.9327 | 32.83/0.9201 | 32.20/0.9022 |
| | 0.9 | 33.74/0.9449 | 33.77/0.9447 | 33.82/0.9455 | 33.77/0.9443 | 33.71/0.9420 | 33.58/0.9387 | 33.18/0.9297 | 32.87/0.9219 | 32.41/0.9101 |
| | 0.5 | 31.93/0.9313 | 31.98/0.9312 | 31.95/0.9304 | 32.07/0.9318 | 32.08/0.9320 | 32.14/0.9322 | 32.21/0.9344 | 32.14/0.9358 | 31.81/0.9298 |
| | 0.6 | 32.30/0.9362 | 32.33/0.9372 | 32.31/0.9369 | 32.34/0.9366 | 32.36/0.9356 | 32.27/0.9331 | 32.04/0.9295 | 31.62/0.9216 | 31.33/0.9176 |
| 80 | 0.7 | 32.39/0.9373 | 32.39/0.9369 | 32.42/0.9376 | 32.42/0.9375 | 32.39/0.9355 | 32.38/0.9329 | 32.16/0.9246 | 31.70/0.9105 | 31.01/0.8860 |
| | 0.8 | 32.55/0.9378 | 32.56/0.9375 | 32.55/0.9374 | 32.57/0.9367 | 32.47/0.9334 | 32.33/0.9282 | 32.17/0.9208 | 31.77/0.9115 | 31.29/0.8944 |
| | 0.9 | 32.53/0.9336 | 32.56/0.9345 | 32.54/0.9337 | 32.56/0.9329 | 32.47/0.9297 | 32.33/0.9249 | 32.11/0.9189 | 31.83/0.9106 | 31.38/0.8968 |
| | 0.5 | 32.63/0.9414 | 32.73/0.9436 | 32.76/0.9442 | 32.81/0.9434 | 32.92/0.9447 | 32.93/0.9440 | 32.92/0.9446 | 32.82/0.9438 | 32.60/0.9425 |
| | 0.6 | 32.91/0.9468 | 32.94/0.9467 | 32.97/0.9464 | 32.95/0.9467 | 33.08/0.9473 | 32.98/0.9446 | 32.75/0.9402 | 32.49/0.9356 | 32.17/0.9316 |
| 120 | 0.7 | 33.00/0.9473 | 33.02/0.9471 | 33.05/0.9470 | 33.16/0.9478 | 33.18/0.9468 | 33.14/0.9435 | 32.86/0.9346 | 32.51/0.9244 | 31.85/0.9032 |
| | 0.8 | 33.30/0.9490 | 33.29/0.9485 | 33.30/0.9484 | 33.30/0.9473 | 33.27/0.9452 | 33.11/0.9396 | 32.84/0.9330 | 32.45/0.9227 | 31.95/0.9066 |
| | 0.9 | 33.28/0.9453 | 33.26/0.9447 | 33.27/0.9446 | 33.24/0.9435 | 33.17/0.9404 | 33.05/0.9372 | 32.82/0.9319 | 32.51/0.9234 | 31.94/0.9085 |

Table 23: Sensitivity analysis of $\ell_p(S_q)$ on the *Seismic* dataset under different frequency counts (i.e., using the first 50, 80, and 120 frames). Each sub-table corresponds to a fixed frequency count, with rows indexed by $p$ and columns indexed by $q - p$. Each entry reports PSNR/SSIM.

| # Freq. | $p$ | $q-p=0$ | 0.0012 | 0.0025 | 0.005 | 0.01 | 0.015 | 0.02 | 0.025 | 0.03 |
|---|---|---|---|---|---|---|---|---|---|---|
| | 0.5 | 41.83/0.9882 | 41.83/0.9882 | 42.15/0.9901 | 42.15/0.9901 | 42.52/0.9911 | 42.53/0.9911 | 42.53/0.9911 | 42.54/0.9911 | 42.54/0.9911 |
| | 0.6 | 42.51/0.9911 | 43.07/0.9914 | 42.63/0.9905 | 42.90/0.9905 | 42.82/0.9912 | 43.02/0.9912 | 42.82/0.9912 | 43.06/0.9913 | 43.06/0.9912 |
| 50 | 0.7 | 42.55/0.9911 | 42.17/0.9901 | 42.17/0.9901 | 42.17/0.9901 | 42.17/0.9901 | 42.17/0.9901 | 42.18/0.9901 | 42.72/0.9914 | 43.44/0.9923 |
| | 0.8 | 43.90/0.9920 | 43.90/0.9919 | 43.91/0.9919 | 44.10/0.9925 | 44.12/0.9925 | 44.12/0.9925 | 44.12/0.9924 | 44.33/0.9926 | 44.33/0.9926 |
| | 0.9 | 43.90/0.9919 | 43.91/0.9919 | 43.92/0.9919 | 44.08/0.9924 | 44.11/0.9924 | 44.14/0.9924 | 44.16/0.9925 | 44.15/0.9924 | 44.20/0.9924 |
| | 0.5 | 42.21/0.9864 | 42.22/0.9864 | 42.22/0.9864 | 42.97/0.9886 | 43.14/0.9898 | 43.15/0.9898 | 43.15/0.9898 | 43.16/0.9897 | 43.16/0.9897 |
| | 0.6 | 43.40/0.9897 | 43.41/0.9897 | 43.40/0.9897 | 43.40/0.9897 | 43.68/0.9906 | 43.58/0.9904 | 43.41/0.9896 | 43.41/0.9895 | 43.58/0.9903 |
| 80 | 0.7 | 42.98/0.9886 | 42.99/0.9885 | 42.99/0.9885 | 42.99/0.9885 | 43.16/0.9897 | 43.16/0.9897 | 43.56/0.9904 | 43.40/0.9896 | 44.02/0.9907 |
| | 0.8 | 43.82/0.9905 | 43.92/0.9909 | 43.94/0.9909 | 44.17/0.9913 | 44.42/0.9913 | 44.60/0.9916 | 44.61/0.9916 | 44.61/0.9916 | 44.73/0.9915 |
| | 0.9 | 43.66/0.9905 | 43.84/0.9905 | 43.94/0.9909 | 44.04/0.9911 | 44.40/0.9914 | 44.52/0.9916 | 44.59/0.9916 | 44.61/0.9915 | 44.67/0.9916 |
| | 0.5 | 42.33/0.9852 | 42.65/0.9862 | 42.65/0.9863 | 42.65/0.9863 | 43.11/0.9874 | 43.23/0.9877 | 43.24/0.9876 | 43.08/0.9871 | 43.08/0.9871 |
| | 0.6 | 43.38/0.9876 | 43.29/0.9871 | 43.23/0.9871 | 43.76/0.9885 | 43.63/0.9877 | 43.47/0.9873 | 43.61/0.9877 | 43.65/0.9881 | 43.55/0.9874 |
| 120 | 0.7 | 43.40/0.9873 | 43.29/0.9870 | 43.29/0.9870 | 43.29/0.9870 | 43.29/0.9869 | 43.43/0.9872 | 43.43/0.9872 | 43.71/0.9877 | 44.56/0.9899 |
| | 0.8 | 44.24/0.9890 | 44.33/0.9891 | 44.34/0.9891 | 44.47/0.9893 | 44.61/0.9895 | 44.84/0.9902 | 45.02/0.9904 | 45.04/0.9904 | 45.08/0.9902 |
| | 0.9 | 44.23/0.9891 | 44.24/0.9890 | 44.30/0.9891 | 44.46/0.9894 | 44.60/0.9899 | 44.73/0.9902 | 44.86/0.9902 | 45.03/0.9903 | 45.02/0.9902 |

**Conclusions based on Empirical Evidence.** The collective evidence from Tables 21 -23 and Figures 8-10 suggests that $(p, q)$ does not require fine tuning. A broad region yields near-optimal results, and the high-performing band around $(p, q) \approx (0.8, 0.81)$ appears consistently across modalities and tensor depths. More importantly, the trends observed in the hyperspectral, video, and seismic experiments agree strongly, implying that the geometric effect of the $\ell_p(S_q)$ prior interacts smoothly with spectral complexity.

**Empirical Recommendation of $(p, q)$.** Based on the overall trends observed in Tables 21–23 and the heatmaps in Figures 9–10, a practical choice in many cases is

$$p \in [0.8, 0.9], \qquad q - p \in [0.005, 0.015].$$

These ranges provide stable reconstruction quality across the tested datasets, and may serve as a useful starting point when no task-specific tuning is available.

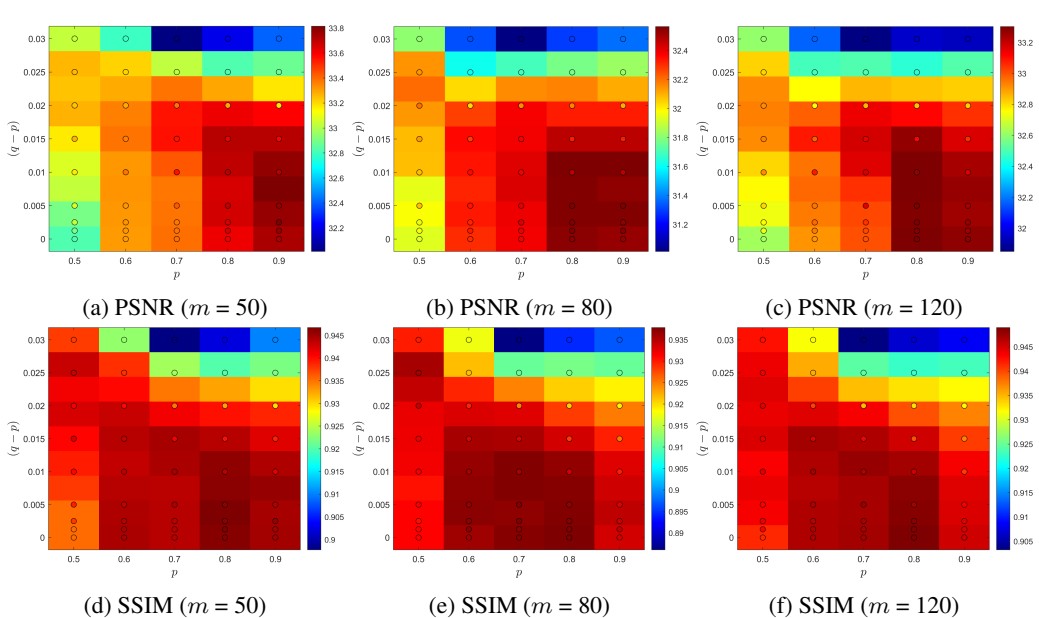

Figure 8: Heatmap visualization of PSNR and SSIM scores for $\ell_p(S_q)$ under varying $(p, q - p)$ configurations. Rows correspond to dataset *Grandma* with metrics (PSNR, SSIM); columns represent frequency counts (50, 80, 120 frames).

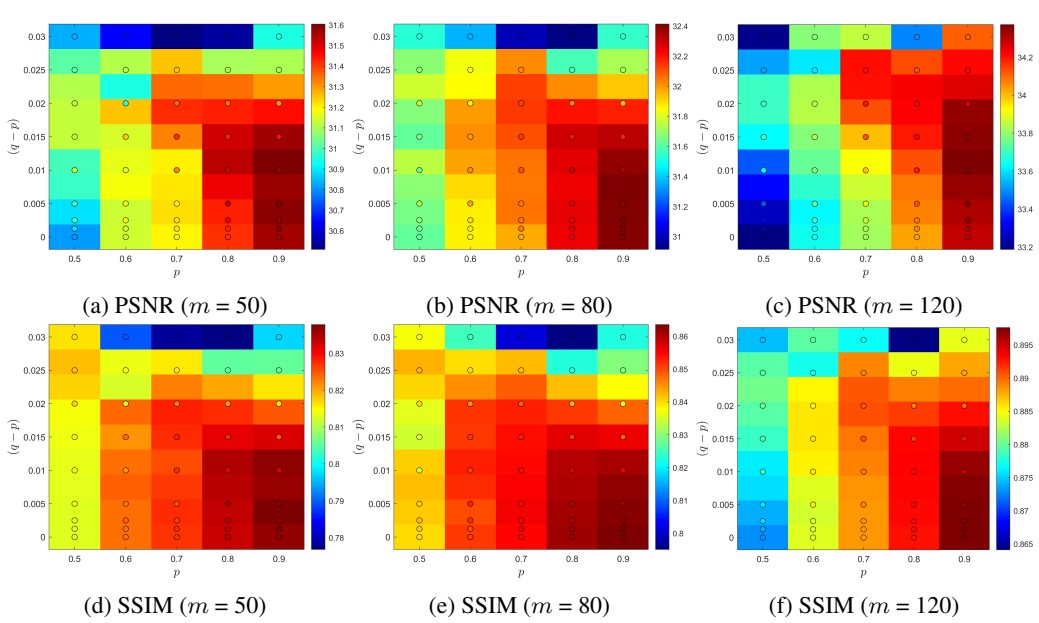

Figure 9: Heatmap visualization of PSNR and SSIM scores for $\ell_p(S_q)$ under varying $(p, q - p)$ configurations. Rows correspond to dataset *SanilasA* with metrics (PSNR, SSIM); columns represent frequency counts (50, 80, 120 frames).

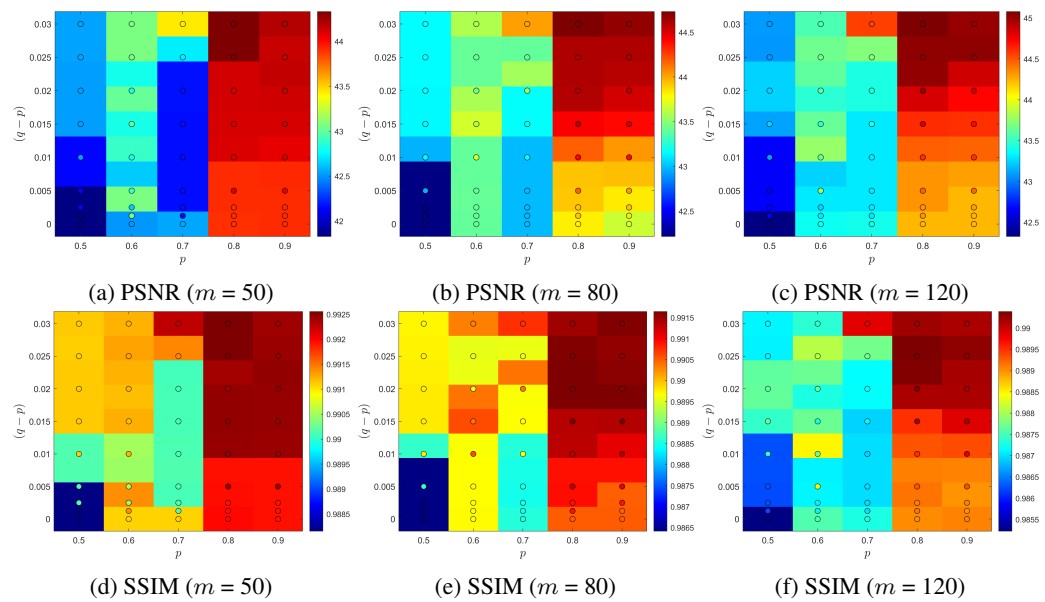

(a) PSNR ($m = 50$)  (b) PSNR ($m = 80$)  (c) PSNR ($m = 120$)

(d) SSIM ($m = 50$)  (e) SSIM ($m = 80$)  (f) SSIM ($m = 120$)

Figure 10: Heatmap visualization of PSNR and SSIM scores for $\ell_p(S_q)$ under varying $(p, q - p)$ configurations. Rows correspond to dataset *Seismic* with metrics (PSNR, SSIM); columns represent frequency counts (50, 80, 120 frames).

## E    DISCUSSION OF LIMITATIONS AND FUTURE DIRECTIONS

The central contribution of this work is the introduction of the $\ell_p$-Schatten-$q$ quasi-norm as a unified tool to capture dual spectral sparsity, together with its statistical characterization via minimax lower bounds and an implementable solver that validates the modeling principle empirically.

This paper represents ***an initial step toward developing a principled foundation for dual spectral regularization in tensor learning***. Our aim is not to resolve every theoretical and algorithmic challenge at once, but to clarify the core ideas and establish a baseline framework. Viewed in this light, several aspects remain beyond the present scope. We outline them below as aspects that clarify the present scope and highlight opportunities for future exploration, rather than as limitations of the proposed approach.

- **Aspect 1: Hard vs. Soft Dual Sparsity.**
  The hard formulation (**A1**) in Section 4 enforces exact sparsity and fixed-rank constraints, which are *overly rigid* for real-world data where spectral contributions and singular values typically *decay gradually* rather than *vanish abruptly*. Thus, the soft formulation (**A3**) is *not a computational convenience but a more realistic model of data structure*.

  ○ From a *modeling* perspective, (**A1**) assumes that only a fixed number of frequency components are active and that each active slice has an exact rank cutoff. In practice, however, real tensors rarely exhibit such abrupt patterns—frequency contributions usually decay gradually, and singular values diminish smoothly rather than dropping to zero. Therefore, (**A3**) is not merely a computational surrogate for (**A1**), but a necessary relaxation that better captures the continuous decay and approximate sparsity observed in real data.

  ○ From a *statistical* perspective, convex relaxations such as the $\ell_1$ norm or the nuclear norm are known to bias large coefficients or singular values, leading to suboptimal recovery rates. In contrast, nonconvex quasi-norms ($q < 1$) reduce this shrinkage bias and more closely approximate the ideal $\ell_0$ or rank constraints. As a result, they are widely shown to achieve sharper minimax recovery guarantees and lower sample complexity in sparse and low-rank estimation, placing our choice within a well-established line of advances in high-dimensional statistics.

  ○ From an *algorithmic* perspective, directly solving (**A3**) is challenging due to its bilayer nonconvex structure, which couples inter-frequency and intra-frequency constraints. To

address this, we design proximal updates that decouple the problem slice by slice in the transform domain and apply closed-form weighted $\ell_{1/2}$ thresholding to singular values. This tailored procedure ensures that each iteration remains computationally tractable while still enforcing the dual-level spectral bias that (**A1**) was meant to capture.

- **Aspect 2: Parameter Selection.**
  The parameters $(p, q)$ define the intrinsic geometry of the regularizer and directly shape its inductive bias. Currently, no principled or statistically consistent theory exists for selecting such quasi-norm parameters—even in simpler vector or matrix cases—making this an important open problem.

  - *Practical strategy:* We adopt a simple and reproducible two-stage procedure on the Indian Pines dataset and then fix $(p, q)$ globally across all datasets and sampling ratios, thereby avoiding per-dataset tuning.
  - *Robustness check:* An alternative configuration, $(p, q) = (0.70, 0.71)$, selected without precise tuning, also delivers competitive performance, suggesting that the method is not overly sensitive to parameter values.
  - *Cross-dataset validation:* Across diverse hyperspectral, multispectral, and thermal datasets, both parameter settings provide consistent performance improvements, suggesting that our framework remains robust without reliance on delicate fine-tuning.

  Developing scalable and theoretically grounded parameter selection strategies remains open, and our empirical results provide preliminary evidence of practical stability and cross-dataset generalization. Toward adaptive selection, learning $(p, q)$ directly from data is a valuable direction. While bilevel optimization offers a potential route, such methods are generally nontrivial under low sampling and nonconvexity, and thus fall beyond the present scope, which focuses on the formulation and theory of our regularizer.

- **Aspect 3: Theoretical Scope.**
  Our theoretical development in Section 4 is carried out under the Gaussian location model with idealized sparsity patterns.

  - *Motivation:* The Gaussian location model serves as a canonical setting in high-dimensional statistics. By abstracting away application-specific complications, it allows us to examine the *intrinsic interplay between inter-frequency sparsity and intra-frequency low-rankness* in its purest form.
  - *Insight:* This abstraction not only clarifies the fundamental estimation difficulty created by the coupled dual structure, but also provides a neutral ground for deriving sharp minimax rates that can serve as benchmarks for more complex models.
  - *Future extensions:* Building upon this baseline, a deeper challenge lies in extending the framework to encompass regression models with covariates, dependent or heavy-tailed noise, and task-heterogeneous structures. Such extensions would connect the present analysis to broader statistical paradigms and bring the theory closer to realistic data-generating processes, which we view as a natural and promising direction for future research.

- **Aspect 4: Algorithmic Guarantees.**
  Our reweighted proximal solver (Section 5) exhibits stable convergence across all experiments, indicating that it provides a workable approach to the problem despite its inherent nonconvexity.

  - *Motivation:* The challenge of $\ell_p(S_q)$ minimization lies in its bilayer nonconvex structure, where group-level frequency selection is intertwined with within-slice spectral shrinkage. This coupling creates optimization difficulties that surpass those in classical single-layer sparse or low-rank problems (Xu, 2010).
  - *Current insight:* To address this, our solver combines proximal updates with weighted $\ell_{1/2}$ thresholding, which effectively disentangles the two levels of structure during iteration. Although a formal global guarantee is still lacking, the observed empirical stability indicates that the method aligns well with the intrinsic geometry of the problem.
  - *Future directions:* Advancing from empirical evidence to rigorous guarantees would require new tools for analyzing bilayer nonconvex objectives. Achieving this would not only strengthen the theoretical foundation of our approach but also contribute more broadly to understanding hierarchical nonconvex optimization.

- **Aspect 5: Computational Aspects.**
  The present implementation in Section 5 relies on full SVDs, which provide a transparent but computationally heavy baseline.

  ○ *Motivation:* Full SVDs ensure that the modeling principle of dual spectral sparsity is validated without approximation artifacts. This choice highlights the intrinsic behavior of the proposed regularizer, separating modeling contributions from implementation-specific accelerations.

  ○ *Current scope:* For the moderate-scale experiments considered here, full SVDs are sufficient and provide a clean reference point. However, their cubic scaling with matrix dimension makes them less suitable for truly large-scale tensor problems, where computational cost can dominate statistical efficiency.

  ○ *Future directions:* A promising line of work is to replace full decompositions with randomized or truncated SVDs, which preserve leading spectral information at substantially lower cost. Beyond randomized methods, integrating adaptive transforms or learned decompositions could further reduce redundancy and tailor the computation to data structure. Progress in this direction would not only expand the scalability of our solver but also contribute to a broader understanding of how dual-level regularization interacts with approximate spectral methods.

In summary, the above aspects are ***not the central focus*** of this work but instead point to natural directions for further development. By clarifying the scope of our contribution, we aim to provide a clear foundation that future studies can build upon in broader theoretical and practical settings.

