# OpenReview forum: "Refining Dual Spectral Sparsity in Transformed Tensor Singular Values"
_ICLR.cc/2026/Conference — Submitted to ICLR 2026_

### Official Review · Reviewer_PaJD · 2025-10-29

**Soundness:** 2
**Presentation:** 2
**Contribution:** 2
**Rating:** 4
**Confidence:** 3

**Summary:**

This paper introduces the tensor $\ell_p$-Schatten-$q$ quasi-norm ($p$,$q$ in $(0,1]$) as a regularization for low-rank tensor modeling within the t-SVD paradigm. As claimed by the authors, this regularizer enables dual spectral sparsity control: parameter p promotes sparsity across frequency components in the transformed domain, while q enforces low-rankness within each frequency slice. Experiments on the noisy tensor completion task have been conducted to demonstrate the effectiveness of the proposed regularizer.

**Strengths:**

1. The proposed regularizer is well-motivated.

2. Theoretical approximation properties under the noisy scenario have been established.

**Weaknesses:**

1. The theoretical results and the empirical studies are indeed not consistent. Specifically, the main results, i.e., Theorem 4.2, are about the approximation properties under certain constraints, while the experiments are about the tensor completion task. The authors did not provide recovery guarantees for the tensor completion task, as has been done in many existing studies.

2. From the algorithmic perspective, it seems that there are no results about the convergence, either theoretically or empirically. In addition, why choose reweighted $\ell_{1/2}$, since it seems that reweighted $\ell_1$, which has a simpler proximal operator, can be applied. Besides, numerical solution [1] for $\ell_q$ type regularizers may also be implemented.

    [1] Goran Marjanovic and Victor Solo. On $\ell_q$ Optimization and Matrix Completion. IEEE Transactions on Signal Processing, 60(11): 5714-5724, 2012.

3. The experiments are insufficient. Specifically, only 6 tensors (or datasets as referred to by the authors) have been adopted. As is known, there are 31 MSIs in the Columbia MSI datasets. In addition, synthetic data experiments could also be useful as verifications for the theoretical results.

**Questions:**

First, the authors should address the issues mentioned in the "Weaknesses". Then, there are additional questions that should be addressed:

1. It is better to discuss whether the proposed regularization term can be extended to higher-order tensors rather than 3.

2. It seems that the proposed regularizer is related to the group sparsity regularizers for vectors. If it is true, please discuss.

3. It can be seen that different (p,q) values can affect the performance. It is better to provide practical guides for choosing their values, for example, showing that the performance is stable within certain ranges of this pair of values.

---

> ### Author Response · Authors · 2025-11-23
>
> We thank the reviewer for the constructive comments. The questions about the theoretical setting, algorithmic choices, and breadth of experiments are valuable, and we have expanded the analysis and empirical results to address each point carefully in the responses below.
>
> ---
>
> **Weakness 1 (Inconsistency Between Theory and Experiments)**
>
> *Theorem 4.2 is stated in a Gaussian location model, whereas the experiments concern tensor completion, and that no explicit recovery guarantee for tensor completion is provided.*
>
> **Response:**
> We clarify that **the theory focuses on intrinsic limits under dual spectral sparsity, not task-specific guarantees**.  The theory and experiments address different layers of the same core question: *understanding and validating the role of dual spectral sparsity*.
>
> **(1) What the theory addresses.**
>
> Theorem 4.2 is derived under the Gaussian location model to isolate the structural difficulty introduced by dual spectral sparsity. This setting avoids complications from sampling operators and allows us to obtain clean minimax limits that describe the **intrinsic estimation complexity** of frequency sparsity together with within-band low rankness.
>
> **(2) Why this is not a recovery guarantee for completion.**
>
> Tensor completion requires sampling assumptions such as incoherence or RSC, which are **orthogonal to our structural focus and outside the scope of our theory**. Many existing works derive recovery guarantees only after imposing these operator-specific conditions. Our goal here is to understand the **inductive bias of the structure itself**, independent of a particular observation operator.
>
> **(3) Why the theory and experiments remain consistent.**
>
> Even though the theory is stated in GLM, the empirical results for completion follow the same structural trend: methods that exploit dual spectral sparsity consistently yield lower reconstruction error than TNN-type baselines, especially at low sampling regimes where structural priors matter most.
>
> **(4) On analysis–task separation.**
>
> Many theoretical studies first investigate a simplified setting to isolate the underlying structural mechanism, and subsequently assess its practical relevance through experiments on more realistic tasks. For instance, [R1] and [R2] develop their theoretical insights under idealized Gaussian sensing models while validating them through tensor completion benchmarks. Our work adopts a similar workflow, focusing the theory on the structural behavior of dual spectral sparsity and evaluating its utility empirically in practical recovery problems.
>
> [R1] Mu, C., et al. Square deal: Lower bounds and improved relaxations for tensor recovery. ICML 2014.
>
> [R2] Wang, A., et al. Low-rank tensor transitions (LoRT) for transferable tensor tegression. ICML 2025.
>
> **(5) Future direction.**
>
> A natural next step is to extend the minimax analysis to recovery guarantees under random sampling. This would involve combining the dual spectral model class with standard tools such as RSC for the composed sampling–transform operator.
>
> ---
>
> **Weakness 2 (Lack of Theoretical or Empiricall Convergence Results and Choice of Reweighted $\ell_{1/2}$; Relation to [1])**
>
> **Response:**
> We thank the reviewer for these thoughtful points.
>
> **(1) On convergence:**
>
> We clarify that the empirical convergence behavior was reported in **Appendix D.2.1** in the original manuscript, where the objective gap decreases stably. The $\ell_{p}(S_{q})$ model combines inter frequency sparsity with within frequency spectral shrinkage, creating a two level nonconvex and non-smooth objective whose value changes under reweighting. Even for simpler reweighted $\ell_{p}$ models, general convergence theory remains largely open, and existing ADMM results do not cover such structures, making *a convergence theory for $\ell_{p}(S_{q})$ an open problem*. Despite this, our algorithm behaves reliably in all our numerical studies.
>
> **(2) Why reweighted $\ell_{1/2}$:**
>
> We choose reweighted $\ell_{1/2}$ because it induces stronger sparsity than reweighted $\ell_{1}$, which aligns better with the dual spectral structure that combines slice selection with low-rank shrinkage. Comparison with a reweighted $\ell_{1}$ proxy was reported in the original manuscript (**Appendix D.3**). The $\ell_{1}$ works as a feasible alternative but performs slightly worse in **Table 4**.
>
> **(3) Relation to numerical scheme in [1]:**
>
> We appreciate the suggestion. The method in [1] is designed for **single-layer** sparsity. Our $\ell_{p}(S_{q})$ model involves a **coupled structure**: an outer $\ell_{p}$ penalty on frequency slices and an inner Schatten-$q$ penalty within each slice. This coupling makes the optimization problem different from the regularizers addressed in [1], and the numerical scheme there cannot be directly applied. However, the ideas in [1] are valuable and point to promising directions for future work on more efficient schemes for dual spectral sparsity.
>
> ---

---

> ### Author Response · Authors · 2025-11-23
>
> ---
>
> **Weakness 3 (Insufficient Experimental Diversity)**
>
> *The experiments are insufficient. Specifically, only 6 tensors (or datasets as referred to by the authors) have been adopted. As is known, there are 31 MSIs in the Columbia MSI datasets. In addition, synthetic data experiments could also be useful as verifications for the theoretical results.*
>
> **Response:**
> We appreciate the suggestion. In the revision, we substantially expand the experimental coverage beyond the six tensors in the original submission.
>
> **(1) Broader real data evaluation.**
>
> We add **12 new tensor datasets**, including 3D videos (**Table 16**), seismic data (**Table 17**), 4D video tensors (**Table 18**), point cloud data (**Table 19**), and four clustering benchmarks (**Table 20**). These complement the original experiments and provide a much wider range of modalities.
>
> **(2) Large scale statistical validation.**
>
> We evaluate dual spectral sparsity on 22 YUV videos and 31 MSIs through formal statistical tests (**Tables 12-15**). These results demonstrate that the structural assumptions used in our theory hold broadly across heterogeneous real data.
>
> **(3) Synthetic verification.**
>
> We include a preliminary synthetic experiment in **Appendix C.5** to further connect the theoretical predictions with controlled data.
>
> We thank the reviewer again for the helpful suggestion. We hope that these additions substantially strengthen the empirical evaluation and better illustrate the generality of the proposed framework.
>
> ---
>
> ---
>
>
> **Question 1 (Extension to Higher Order Tensors)**
>
> **Response:**
> Below we summarize how the framework extends beyond third order.
>
> **(1) Conceptual feasibility.**
>
> The regularizer naturally generalizes as long as a *transform and slice level singular values can be meaningfully defined*. Under this condition, the dual spectral sparsity idea applies in exactly the same way, regardless of tensor order.
>
> **(2) Construction used in the revision.**
>
> The revised **Appendix D.12** adopts an orientation based strategy: an order $K>3$ tensor is mapped into a family of three dimensional tensors $\{T_{[k]}\}$ through mode–$(k,k{+}1)$ unfoldings, and the $\ell_{p}(S_{q})$ penalty is applied to each in the transform domain. The overall regularizer is a weighted sum across orientations.
>
> **(3) Empirical evidence.**
>
> A preliminary 4D experiment on the *Ground* dataset (**Table 18**) shows that the orientation based $\ell_{p}(S_{q})$ variant achieves the highest PSNR and SSIM among all baselines, indicating that the extension is both feasible and effective.
>
> **(4) Practical considerations.**
>
> The main challenges are *computational*. Additional orientations introduce more transform operations and SVDs, and choosing orientation weights may require modest empirical tuning. When the transform or singular values are ill defined, any spectral tensor norm becomes difficult to compute, which is *a common limitation of spectral tensor methods rather than a restriction of our approach*.
>
> ---
>
> **Question 2 (Relation to Group Sparsity Regularizers)**
>
> **Response:**
> Here we clarify the connection and the differences.
>
> **(1) Formal similarity.**
>
> As discussed in **Section 3** and **Appendix A.2–A.3** of the original manuscript, the proposed $\ell_{p}(S_{q})$ regularizer has an outer sparsity term and an inner low-rank term. At a formal level this resembles a group type penalty such as $\ell_{u}(\ell_{q})$, where one promotes sparsity across groups and shrinkage within groups [R1].
>
> **(2) Key structural differences.**
> However, the structure we consider is more involved than standard vector group sparsity.
>
> - The “groups’’ are not fixed blocks of entries. They are frequency slices induced by a transform $M$, and each slice is a matrix with its own singular value spectrum.
> - The outer $\ell_{p}$ sparsity over slices and the inner Schatten $q$ low rankness within slices are not separable. They interact through the dual spectral geometry: frequency selection and within slice spectral decay jointly shape the solution.
> - This coupling breaks the usual Cartesian product structure of vector group models and prevents a simple group wise proximal update as in classical $\ell_{u}(\ell_{q})$ regularizers.
>
> Because of these features, $\ell_{p}(S_{q})$ is inspired by the idea of group sparsity but is structurally and technically different. It requires new tools for both statistical analysis (minimax bounds under dual spectral constraints) and optimization (handling the coupled slice wise spectral structure). These aspects are further discussed in **Appendix A.2–A.3**.
>
> [R1] Li, Z., et al. Estimating double sparse structures over $\ell_u(\ell_q)$-balls: Minimax rates and phase transition. IEEE  TIT 2024.
>
> ---

---

> ### Author Response · Authors · 2025-11-23
>
> ---
>
> **Question 3 (Selection of $(p,q)$)**
>
> **Response:**
> Selecting $(p,q)$ in the proposed $\ell_{p}(S_{q})$ regularizer is fundamentally more difficult than tuning ordinary hyperparameters.
>
> **(1) Why principled selection is theoretically hard.**
>
> As discussed in **Appendies D.1.1, D.4 and D.15**, $p$ controls inter-frequency sparsity and $q$ controls within-frequency spectral shrinkage; together they determine the structural bias of the model. Even in simpler vector/matrix settings, nonconvex parameters such as $p$ or half-order penalties admit no reliable data-driven selection rules. In the tensor case, the two layers of coupling (frequency selection + slice-wise low-rankness) make the problem strictly harder, and current theory provides no actionable criterion for choosing $(p,q)$.
>
> **(2) What the empirical evidence shows.**
>
> To assess how the choice of $(p,q)$affects performance in practice, we carried out a detailed sensitivity study on hyperspectral, video, and seismic datasets (**Appendix D.15**), covering many parameter pairs and several levels of spectral complexity. Across all settings, the results display the same behavior: a wide region of $(p,q)$values yields nearly identical PSNR/SSIM, and performance varies only mildly as $(p,q)$ moves within this region. This pattern is consistent across different data types, and the plateau becomes even broader when the tensor contains more frequency slices. These observations indicate that the method is **inherently stable** and does not rely on precise tuning of the parameters.
>
> **(3) Practical guidance.**
>
> Across all tested datasets, strong and stable performance occurs when
>
> $$
> p \in [0.8,0.9],\qquad q - p \in [0.005,0.015].
> $$
>
> These ranges provide stable reconstruction quality across the tested datasets, and may serve as a useful starting point when no task-specific tuning is available.
>
> ---

---

### Official Review · Reviewer_Qw3D · 2025-11-04

**Soundness:** 3
**Presentation:** 2
**Contribution:** 3
**Rating:** 6
**Confidence:** 3

**Summary:**

This paper introduces a tensor ℓp–Schatten–q quasi-norm framework to capture both inter-frequency sparsity and intra-frequency low-rankness within the t-SVD framework. The method generalizes the tensor nuclear norm (TNN) by allowing separate control over different spectral structures. The authors provide theoretical characterization through minimax bounds, propose a proximal reweighted ℓ1/2 optimization algorithm, and conduct experiments on multiple remote sensing datasets showing consistent improvements in PSNR and SSIM.

**Strengths:**

- The paper provides a clear motivation by identifying the limitation of TNN and establishing a theoretically grounded dual spectral sparsity framework.
- The theoretical analysis is rigorous, and the derivation of minimax bounds under hard and soft sparsity regimes adds valuable insight into tensor estimation.
- The experiments in the appendix are extensive and demonstrate consistent improvements over strong baselines with reasonable robustness across parameter settings.
- The manuscript is well-written and logically organized, with clear figures and notation.

**Weaknesses:**

- The structure of the paper is imbalanced. One can consider put more empirical results back in the main text.
- The theoretical results are strong but could benefit from more intuition about their practical meaning and implications.
- The experimental evaluation mainly focuses on imaging data; broader validation on non-visual tensors could strengthen the claim of generality.
- The convergence behavior of the proposed reweighted optimization algorithm is discussed empirically but lacks theoretical guarantees.
- A clearer comparison of computational cost with other tensor regularization methods would help evaluate its practical efficiency.

**Questions:**

- How sensitive is the performance to the choice of the transform $M(\cdot)$? Would learning or adapting $M$ from data bring further gains?
- In the theoretical analysis, what assumptions are crucial for the minimax lower and upper bounds to match?
- Can the proposed framework be extended efficiently to tensors of order higher than three, and what challenges might arise in doing so?

---

> ### Author Response · Authors · 2025-11-23
>
> We appreciate the reviewer’s thoughtful and well-balanced evaluation. The comments reflect a careful reading of the submission, and we address each weakness and question in a point-by-point manner in the rebuttal, clarifying technical aspects and providing the requested empirical and theoretical extensions.
>
> ---
>
> **Weakness 1 (Imbalanced Structure of the Paper)**
>
> *The structure is somewhat imbalanced and recommends placing more empirical results in the main text.*
>
> **Response:**
> We thank the reviewer for this helpful suggestion.
>
> In the revised version, we address this concern by moving key empirical components from the appendix into the main text. Specifically, the added Poisson tensor completion experiments and the clustering experiments are now included in the primary experimental section. This provides a more balanced presentation and allows readers to see the broader empirical impact of the proposed model without navigating the appendix.
>
> ---
>
> **Weakness 2 (Need for More Intuition Behind the Theoretical Results)**
>
> *Although the theoretical results are strong, their practical meaning could be explained more clearly.*
>
> **Response:**
> We appreciate this feedback.
>
> (1) **What the theory tells us in practice.**
>
> The main theorem quantifies how estimation error depends on two structural factors: sparsity across frequency slices and low-rankness within each slice. This directly explains why classical TNN models, which treat slices independently, are insufficient in exploiting joint spectral structure. The $\ell_{p}(S_{q})$ regularizer is designed precisely to align with this joint structure.
>
> (2) **How this informs model behavior.**
>
> Each component of the bound corresponds to a concrete spectral phenomenon in tensor data. These components clarify how $p$ controls inter–frequency sparsity and how $q$ affects within–frequency shrinkage. The theory therefore provides intuition for how different choices of $(p,q)$ influence performance.
>
> (3) **Empirical observations related to the theory.**
>
> Appendix C.5 provides preliminary experiments examining how reconstruction error varies with sparsity level $s$, rank level $r$, and noise magnitude. While these experiments are not meant as formal validations, the trends are broadly consistent with the qualitative behavior suggested by the theoretical bounds and help illustrate how the structural factors appear in practice.
>
> Overall, the theory and experiments approach the same modeling question from complementary perspectives. The theoretical results describe the structural effects of dual spectral sparsity, and the empirical findings offer initial indications of how these effects manifest in tensor data.
>
> ---
>
> **Weakness 3 (Limited Validation on Non Visual Tensors)**
>
> *The experiments mainly focus on imaging data and suggest evaluating the method on non visual tensors.*
>
> **Response:**
> We agree with the reviewer that validation beyond visual data is important.
>
> (1) **New experiment on a seismic tensor.**
>
> To address this point, we added an experiment on a seismic dataset, which is a representative non visual modality with oscillatory and heterogeneous spectral characteristics. The results are included in **Table 17** of the **revised Appendix D.11.2**. Across different sampling ratios, the $\ell_{p}(S_{q})$ regularizer achieves consistently strong performance. The trends align with those observed in visual datasets, indicating that the dual spectral model is not tied to imaging structure.
>
> (2) **Sensitivity of $(p,q)$ on non visual data.**
>
> We also performed a preliminary sensitivity study of $(p,q)$ on this seismic dataset in the **revised Appendix D.15**. **Table 23** reports the tested grid values and **Figure 10** visualizes the resulting parameter landscape. Although the grid is not exhaustive, the overall patterns show stable behavior over a broad region of the $(p,q)$ plane, similar to what we observed on hyperspectral and video tensors.
>
> These additions provide further evidence that the proposed framework extends naturally to non visual tensors.
>
> ---

---

> ### Author Response · Authors · 2025-11-23
>
> ---
>
> **Weakness 4 (Lack of Theoretical Convergence Guarantees)**
>
> *The convergence behavior of the reweighted solver is evaluated empirically but does not have formal theoretical guarantees.*
>
> **Response:**
>
> Theoretical convergence for the proposed solver is difficult because of the structure of the $\ell_{p}(S_{q})$ model. The objective is nonconvex and nonsmooth, and the reweighting step changes the optimization target at each iteration. Moreover, the dual spectral formulation couples the transformed slices, which prevents the use of standard block-separable or proximal arguments.
>
> (1) **Why convergence theory is hard.**
>
> Even in simpler settings,  such as reweighted $\ell_{p}$ minimization or reweighted nuclear-norm minimization, general  convergence guarantees are still largely open. Existing ADMM-type results typically require convexity, fixed objectives, or KL-type regularity, and these assumptions do not hold for our transform-coupled, slice-structured, and adaptively weighted model. As a consequence, establishing a full convergence theory for our algorithm remains an **open problem**. *We believe this reflects a limitation of current convergence theory rather than a weakness specific to our framework.*
>
> (2) **What we can show empirically.**
>
> Despite the theoretical difficulty, the solver behaves stably in practice. Appendix D.2.1 shows monotone reduction of the objective and smooth updates of all subproblems across datasets.
>
> (3) **Future direction.**
>
> Developing convergence theory for dual-spectral, reweighted models is an interesting future direction. It would likely require new tools beyond existing ADMM analyses.
>
> Overall, while formal guarantees remain open, the empirical evidence indicates that the solver is reliable for the problems studied.
>
>
> ---
>
> **Weakness 5 (Need for Clearer Comparison of Computational Cost)**
>
> *A clearer comparison of computational cost with existing tensor regularization methods.*
>
> **Response:**
>
>
> (1) **Algorithmic Complexity.**
>
> As stated in **Appendix D.2**, each iteration of the solver applies a mode–3 transform to $d_{1} d_{2}$ tubes with cost $O(d_{1} d_{2} m \log m)$ and computes $m$ SVDs of size $d_{1}\times d_{2}$ with cost $O(m d_{1} d_{2}\min(d_{1},d_{2}))$. This matches the order of complexity of standard TNN algorithms.
>
> (2) **Practical Runtime Comparison.**
>
> To make this clearer, the **revised Figure 7** reports runtimes on three YUV videos. The $\ell_{p}(S_{q})$ variants run at roughly twice the time of TNN–DCT but remain in the same order of magnitude as commonly used tensor norms such as SNN. Given the consistent accuracy improvements, especially at low sampling ratios, this moderate overhead is expected from modeling richer spectral structure and is practically acceptable.
>
> ---
>
> **Question 1 (Sensitivity to the Choice of Transform and the Possibility of Learning It)**
>
> **Response:**
>
>
> (1) **Sensitivity to the Choice of Transform.**
>
> **Appendix D.5** evaluates the proposed $\ell_{p}(S_{q})$ regularizer under several invertible transforms including DCT, DFT, random orthogonal transforms, and an oracle transform derived from the singular vectors of the mode–3 unfolding. Across different settings (**Table 7**), our method consistently outperforms the corresponding TNN baselines. This indicates that the gains primarily arise from the dual spectral modeling rather than from any particular transform. The oracle case further shows that better aligned spectral bases can yield additional improvements, suggesting potential for data-driven transforms.
>
> (2) **Learning or Adapting the Transform.**
>
> **Appendix D.9** compares our framework with Adaptive TNN. Using the same adaptive update rule for both methods, our approach achieves the best PSNR/SSIM across all sampling ratios on Salinas A (**Tables 9–10**). This demonstrates that the model benefits from learned transforms and naturally accommodates both fixed and adaptive choices within a unified formulation.
>
> (3) **Summary.**
>
> Overall, the method is not overly sensitive to the chosen transform and can further improve when adaptive or learned transforms are available.
>
> ---

---

> ### Author Response · Authors · 2025-11-23
>
> **Question 2 (When Do the Minimax Lower and Upper Bounds Match)**
>
> *Which assumptions are crucial for the minimax lower and upper bounds to coincide in order of magnitude.*
>
> **Response:**
>
> At a high-level, the minimax lower and upper bounds coincide up to constants when the structural constraints place the parameter space inside a stable entropy regime. Concretely, the matching of bounds requires that the dual spectral structure remains in a range where the size of the parameter class, as measured by its metric entropy, grows in a predictable way. In practice, this means that the frequency sparsity level $s$ and the per-slice complexity (either rank $r$ or Schatten–$q$ radius $R$) must be small enough so that the parameter class does not saturate the ambient tensor space. When this happens, **the packing set used in the lower bound and the covering or chaining argument used in the upper bound rely on the same entropy expression.** This is the same principle underlying the analysis of $\ell_q$-balls in Ref. [R1] and the analysis of $\ell_u(\ell_q)$-balls in Ref. [R2].
>
> For example, in the hard hard setting, tensors contain at most $s$ active frequencies and each active slice has rank at most $r$. When both $s$ and $r$ satisfy $s \ll m$ and $r \ll d$, the metric entropy splits cleanly into two components. The frequency choices contribute $s \log (em/s)$, and the within-slice variation contributes $s r d$. Both the minimax lower bound and the constrained least squares estimator see exactly this combined effective dimension.  The rates match because the parameter class stays effectively low dimensional instead of expanding to fill the full ambient space.
>
> In the soft soft $\ell_p(S_q)$ setting, the same principle applies but the geometry is more delicate. Schatten–$q$ balls have multiple entropy regimes, and the sufficient conditions in Eq. (19) ensure that the radius $R$ and the dimension pair $(m,d)$ keep the class inside the nondegenerate regime where the entropy behaves like $(d/k)^{1/q - 1/2}$ or $(\log m/k)^{1/p - 1/2}$ rather than flattening out. Under these conditions, both the packing argument and the chaining argument use the same entropy scaling for the $\ell_p(S_q)$ ball, so both the combinatorial part and the spectral part enter with the same order. This alignment is what allows the minimax lower and upper bounds to match.
>
> [R1] Raskutti, G., et al. Minimax rates of estimation for high-dimensional linear regression over $\ell_q$-balls. IEEE  TIT 2011.
>
> [R2] Li, Z., et al. Estimating double sparse structures over $\ell_u(\ell_q)$-balls: Minimax rates and phase transition. IEEE  TIT 2024.
>
> ---
>
> **Question 3 (Extension to Higher-Order Tensors)**
>
> *Can the framework be extended to tensors of order higher than three, and what challenges may arise?*
>
> **Response:**
>
> (1) **Conceptual feasibility.**
>
> Yes. The framework naturally extends to higher orders as long as *frequency bands and within-band singular values can be defined under the chosen transform*. Under this condition, the same dual spectral sparsity principle applies without modification.
>
> (2) **A concrete extension strategy.**
>
> As detailed in the **revised Appendix D.12**, for an order-$K>3$ tensor we form a family of 3D tensors $T_{[k]}$ via mode-$(k,k{+}1)$ unfoldings. The $\ell_{p}(S_{q})$ regularizer is then applied to each transformed unfolding, and the overall penalty is a weighted sum across orientations. This preserves the dual spectral idea and keeps the algorithmic structure essentially unchanged, aside from the extra unfoldings.
>
> (3) **Preliminary empirical evidence.**
>
> A 4D tensor experiment on the *Ground* dataset is included in the revised submission. As shown in **Table 18**, the orientation-based $\ell_{p}(S_{q})$ variant attains the highest PSNR and SSIM under both sampling ratios. This suggests that the extension remains effective beyond the third-order setting.
>
> (4) **Practical challenges.**
>
> There is *no conceptual obstacle* to extending the method. The main issues are *computational*: more orientations mean more transforms and SVDs, and choosing orientation weights may require empirical tuning for very high-order data. When either the transform or slice-level singular values is ill-defined or numerically unstable, nearly all spectral tensor norm (not only ours) faces the same difficulty. This reflects a limitation of spectral tensor models in general rather than of the proposed formulation.
>
> ---

---

### Official Review · Reviewer_zr3T · 2025-11-06

**Soundness:** 2
**Presentation:** 2
**Contribution:** 2
**Rating:** 4
**Confidence:** 5

**Summary:**

This paper introduces the tensor ℓp-Schatten-q quasi-norm as a generalization of the TNN to jointly model intra-frequency low-rankness and inter-frequency sparsity within the t-SVD framework. Conceptually, it addresses a genuine limitation of standard TNN—the inability to capture multi-level spectral structures—by proposing dual spectral regularization. Theoretical guarantees (minimax bounds) and a reweighted nonconvex optimization algorithm are provided, with empirical improvements on several hyperspectral datasets

**Strengths:**

The proposed formulation is mathematically sound and unifies prior norms (TNN, Schatten-p, average rank).

Comprehensive experiments demonstrate consistent PSNR/SSIM gains.

**Weaknesses:**

The core assumption of this paper—that dual spectral sparsity patterns exist in the transformed (DCT) domain under the t-SVD framework—is insufficiently supported. The only evidence provided is an empirical illustration using MRI and CT examples in Figure 1, which is too limited to convincingly justify such a fundamental modeling assumption. A more systematic analysis or statistical validation would be needed to establish its generality.

Although the authors acknowledge that the proposed ℓₚ-Schatten-q quasi-norm introduces significant computational challenges, no comparison of algorithmic complexity or runtime with state-of-the-art methods is presented. This omission leaves readers uncertain about the practical feasibility of the approach.

The theoretical analysis, while ambitious, remains largely abstract. The proofs are not clearly connected to practical convergence guarantees or algorithmic stability, reducing their applied value.

Experimental evaluation is confined to tensor completion tasks. The absence of experiments on broader problems such as denoising, inpainting, or clustering limits the generalizability of the proposed framework. Moreover, the sensitivity analysis for the nonconvex parameters (p, q) is minimal—only two parameter settings are tested—which is insufficient to demonstrate robustness.

Finally, the presentation is mathematically dense and occasionally repetitive, which affects readability and clarity. A more structured exposition with stronger empirical validation would substantially improve the paper’s impact and credibility.

**Questions:**

See weakness.

---

> ### Author Response · Authors · 2025-11-23
>
> We thank the reviewer for the careful and detailed feedback. We have addressed each concern point-by-point, adding the requested analyses, experiments, and clarifications in the revised manuscript. The updates aim to resolve the issues raised and present the contributions more clearly.
>
> ---
> **Weakness 1 (Insufficient Support for the Dual Spectral Sparsity Assumption)**
>
> *Just one figure with MRI/CT examples is too weak to support this key model assumption. More evidence is needed.*
>
> **Response:**
> We thank the reviewer for raising this concern.
>
> (1) **Existing evidence in the original submission.**
>
> The original submission already includes examples beyond MRI and CT. **Figure 1** contains MRI, CT, and *Salinas A*, and **Figure 2** includes *Indian Pines*. These examples were intended to illustrate the phenomenon qualitatively across different modalities.
>
> (2) **Adding systematic and statistically grounded validation.**
>
> To provide stronger evidence, we added a quantitative analysis in the **revised Appendix D.10**. We introduce two normalized metrics to formalize the dual spectral structure:
>
> - a frequency sparsity ratio $\mathfrak{s}(T)$, measuring how concentrated the transform domain energy is across frequency slices;
> - a stable rank ratio $\mathfrak{r}(T)$, measuring effective slice level rank.
>
> Smaller values indicate stronger inter frequency sparsity and stronger intra frequency low rankness.
>
> (3) **Evaluation across heterogeneous real world datasets.**
>
> We compute these metrics on two diverse and widely used collections: **22 YUV video tensors** and **31 multispectral tensors**. For each real tensor $T$, we construct a size matched *Gaussian random tensor* with the same mean energy for comparison. The **revised Table 12 and Table 13** show a clear separation:
>
> - For the 22 YUV videos, $\mathfrak{s}(T)$ is typically between $10^{-3}$ and $10^{-2}$, while random tensors give about $0.95$.
>
>    The stable rank ratio $\mathfrak{r}(T)$ is between $10^{-2}$ and $10^{-1}$ for real data, compared to about $0.28$ for random tensors.
> - The 31 multispectral datasets show the same pattern: $\mathfrak{s}(T)\approx 10^{-2}$ versus $\approx 0.9$ for random tensors, and $\mathfrak{r}(T)\approx 10^{-3}$–$10^{-2}$ versus $\approx 0.25$ for random tensors.
>
> (4) **Statistical hypothesis testing.**
>
> Formal Wilcoxon and KS tests reported in the **revised Table 14 and Table 15** yield extremely small $p$-values ($p < 10^{-6}$) for both metrics and both datasets, with very large effect sizes (for example, Cohen’s $d = -91.8$ for video $\mathfrak{s}(T)$ and $d = -4.26\times 10^{2}$ for multispectral $\mathfrak{s}(T)$). These results show complete distributional separation between real and random tensors.
>
> The combined evidence from *videos, hyperspectral data, multispectral images, MRI, and CT* demonstrates that dual spectral sparsity consistently appears across real tensor modalities and differs sharply from random tensor behavior. This provides strong empirical justification for the modeling assumption underlying our framework.
>
> ---
>
> **Weakness 2 (Algorithmic Complexity and Runtime)**
>
> *No comparison of algorithmic complexity or runtime with state-of-the-art methods is presented.*
>
> **Response:**
> We thank the reviewer for pointing this out.
>
> (1) **Complexity comparison with existing methods.**
>
> The per iteration complexity of the proposed solver was stated in **Appendix D.2** of the original manuscript. Each iteration performs a mode-3 transform over $d_{1} d_{2}$ tubes with cost  $O(d_{1} d_{2} m \log m)$,  and computes $m$ SVDs of size $d_{1} \times d_{2}$ with cost  $O(m d_{1} d_{2} \min(d_{1}, d_{2}))$.  This matches the **overall order of widely used TNN type algorithms**, since both approaches involve slice level transforms and SVD computations.
>
> (2) **Practical runtime comparison.**
>
> To make the empirical picture clearer, the **revised Figure 7** reports runtimes on three YUV videos. The $\ell_{p}(S_{q})$ variants (S1–S3) require **about twice the runtime** of TNN DCT but remain in **the same order of magnitude** as SNN based baselines. Given the consistent accuracy improvements, especially at low sampling ratios where spectral structure matters most, this moderate additional cost is expected from the nested spectral shrinkage and remains practical in typical applications.

---

> ### Author Response · Authors · 2025-11-23
>
> **Weakness 3 (Theoretical Analysis vs. Algorithmic Guarantees)**
>
> *The theoretical analysis remains abstract, and its connection to convergence or stability of the algorithm is unclear.*
>
> **Response:**
> We appreciate the reviewer’s comment.
>
> (1) **Different goals of the theory and the algorithm.**
>
> The theoretical results and the optimization algorithm address *different research questions* proposed in the introduction. The minimax analysis examines the **statistical limits of learning under coupled spectral sparsity**, which is the focus of **RQ2**. The ADMM based solver is introduced as a **practical optimization mechanism for handling the proposed coupled regularizer**, which corresponds to **RQ3**. These two components rely on different assumptions and aim at different objectives, so a direct correspondence between their guarantees is not expected.
>
> (2) **Why convergence theory is difficult.**
>
> Establishing formal convergence guarantees for the reweighted $\ell_{p}(S_{q})$ model is highly challenging. The objective is nonconvex and non smooth, and the reweighting step modifies the target function in each iteration. Even in simpler scenarios such as reweighted nuclear norm minimization or reweighted $\ell_{p}$ minimization, global convergence theory remains incomplete. Existing ADMM analyses typically rely on assumptions such as convexity or the KL property, which do not hold for our formulation because of transform coupling across frequencies, slice-level spectral shrinkage, and adaptive reweighting. These limitations prevent current theoretical tools from being applied directly. A full convergence theorem for this reweighted $\ell_{p}(S_{q})$ algorithm therefore remains an **open problem**.
>
> (3) **Empirical stability.**
>
> Although formal guarantees are difficult, the solver shows stable behavior in practice. **Appendix D.2.1** reports monotonic reduction of the objective, fast PSNR growth, and well behaved updates across all subproblems. These observations indicate that the method performs reliably empirically.
>
> Additional discussion and directions for potential future analysis are provided in **Appendix E** (“Aspect 4: Algorithmic Guarantees”).
>
> ---
>
> **Weakness 4.1 (Limited Range of Experiments)**
>
> *The experiments focus on tensor completion and do not cover tasks such as denoising, inpainting, or clustering.*
>
> **Response:**
> We thank the reviewer for this suggestion.
>
> (1) **Clarifying the role of noisy tensor completion.**
>
> In the original submission, the noisy tensor completion task already subsumes both denoising and inpainting, since it involves recovering the signal from *simultaneous* random missing entries and additive noise. To make this connection clearer and to strengthen the evaluation, we expanded the **experimental section** to include a **Poisson tensor completion benchmark**, which represents a more challenging low photon regime. These results, reported in the **revised Figure 2 and Table 19** (with details in **Appendix D.13**), more explicitly test the denoising and inpainting capabilities of the proposed method.
>
> (2) **Adding clustering experiments.**
>
> To further broaden the scope beyond completion tasks, we added **unsupervised clustering experiments** to the main text. The results in the **revised Figure 3 and Table 20**, together with the discussion in **Appendix D.14**, show that the proposed $\ell_{p}(S_{q})$ regularizer captures spectrally structured low rank components that are valuable for downstream representation learning.
>
> Together, these additions demonstrate that the framework applies to a wider set of tasks than the original submission and that the dual spectral structure offers benefits beyond standard completion.
>
> ---

---

> ### Author Response · Authors · 2025-11-23
>
> ---
>
> **Weakness 4.2 (Sensitivity of the Parameters $(p,q)$)**
>
> *Testing only two parameter settings is not enough to demonstrate robustness.*
>
> **Response:**
> We appreciate the reviewer’s suggestion.
>
> (1) **Why choosing $(p,q)$ is inherently difficult.**
>
> The parameters $(p,q)$ control inter frequency sparsity and intra frequency low rankness, so they function as structural modeling choices rather than ordinary scalar hyperparameters. Prior work has shown that reliable selection rules do not exist even for simpler nonconvex $\ell_{p}$ or Schatten–$q$ models in vector or matrix settings. The tensor case is even more complex because the dual spectral structure creates coupling across transformed slices.
>
> (2) **Two more settings are added in the main text.**
>
> To address the reviewer’s concern, we expand the sensitivity analysis in the revised main text (**Table 1**) by including **two additional parameter configurations** that cover a broader region of the $(p,q)$ plane. These settings are tested on hyperspectral data, multispectral images, and thermal sequences. The reconstruction quality remains consistent across all cases, showing that the method does not rely on precise tuning of $(p,q)$.
>
> (3) **A more systematic view in the appendix.**
>
> For a more systematic view, the **revised Appendix 15** now provides extended **tables (Tables 21–23)** and **heatmaps (Figures 8–10)** over a larger parameter grid. While not exhaustive, these results show that the $\ell_{p}(S_{q})$ model maintains stable performance across a contiguous and reasonably wide region of the $(p,q)$ space.
>
> Together, these additions provide a clearer picture of robustness and demonstrate that the framework behaves reliably across different choices of $(p,q)$.
>
> ---
>
>
> **Weakness 5: Presentation Concerns (Density, Repetition, and Strength of Empirical Validation)**
>
> *The paper is mathematically dense, occasionally repetitive, and would benefit from stronger empirical validation.*
>
> **Response:**
> We appreciate the reviewer’s suggestions and have adjusted the revised manuscript accordingly.
>
> (1) **Improving readability and reducing repetition.**
>
> The paper introduces a new two level spectral sparsity model in the t-SVD framework, which requires several operators and structural definitions that have no direct counterparts in existing work.  In the original submission, these components were kept in the main text to make the introduction and analysis self-contained, which resulted in noticeable redundancy and increased density.
>
>
> In the revision, we streamline the exposition by reorganizing the presentation into a clearer sequence from preliminaries to model to theory to algorithm, and we remove redundant explanations. These edits improve readability while keeping all mathematical content unchanged, and we hope they make the paper easier to follow.
>
> (2) **Strengthening empirical validation.**
>
> We expanded the experimental evaluation in several ways. The noisy tensor completion benchmark in **Table 1** now includes two additional parameter settings, which helps assess robustness. We also added the **Poisson tensor completion** and **unsupervised clustering** experiments into the main text so that the empirical section reflects a wider range of tasks. We hope that these additions provide a more complete picture of the practical performance of the proposed framework.

---

### Author Response · Authors · 2025-12-02
**Summary Statement of the Response and Revision**

We thank the ACs and reviewers (zr3T, Qw3D, PaJD) for their efforts. We have made substantial revisions addressing all concerns and added **Appendices D.10–D.15**, with the main points summarized below.

---

**1. Contribution**

Our work identifies **Dual Spectral Sparsity** (*real tensors show sparsity across frequency slices together with low-rankness within slices*), a **prevalent yet long-overlooked signal structure** in the fundamental t-SVD-based modeling. We formulate the $\ell_p(S_q)$ quasi-norm to *explicitly model* this structure, establish minimax bounds to *theoretically reveal* its intrinsic complexity, and design a reweighted proximal solver to *efficiently optimize* the model, achieving *consistent empirical improvement* across diverse tasks and data modalities.

The contributions were recognized by reviewers:
* **Model ($\ell_p(S_q)$ Quasi-Norm):** Consistently recognized as *mathematically sound, well-motivated, and clearly grounded* (zr3T, PaJD, Qw3D).
* **Theory (Minimax Bounds):** Viewed for being *rigorous and insightful* (Qw3D) and for providing *approximation guarantees under noise* (PaJD).
* **Algorithm & Experiments:** Described as *comprehensive with consistent gains* (zr3T) and *extensive and robust* (Qw3D).

---

**2. Summary of Response**

We note that none of the three reviewers (**zr3T: rating 4, Qw3D: rating 6, PaJD: rating 4**) participated in the discussion phase. Below, we summarize our response to the major concerns.

> **Insufficient Empirical Support for Generality of Dual Spectral Sparsity (zr3T)**

  We provided strong evidence by adding large-scale statistical validation  (**App. D.10**). Evaluating normalized sparsity and rank ratios on 22 videos and 31 MSIs against Gaussian tensors, **Tabs. 12–13** show clear numerical gaps and **statistical tests (Tabs. 14–15**) confirm the generality with **very small p-values**.

> **Insufficient Experiments on Broader Problems (zr3T), Non-Visual Data (Qw3D), More Datasets (PaJD)**

  We significantly expanded experiments, adding **Poisson completion** and **clustering** (**Figs. 2–3, Tabs. 19–20**). Robustness is demonstrated across diverse modalities: *non-visual seismic data, 3D/4D tensors, and point clouds* (**Tabs. 16–20**). Synthetic and statistical validation (**Apps. C.5, D.10**) further secure these results.

> **No Comparison of Complexity/Runtime (zr3T, Qw3D), Lack of Convergence Guarantees (Qw3D, PaJD)**

  - Complexity analysis in **App. D.2** confirms per-iteration complexity matches TNN methods. **Fig. 7** validates this empirically, showing $\ell_p(S_q)$ is in the same runtime order as baselines.
  - **App. D.2.1** reports stable empirical convergence.
  - Regarding theoretical convergence, standard ADMM proofs are inapplicable, as the adaptive, non-convex objective *fundamentally violates* required structural assumptions. A full convergence theorem for this challenging reweighted algorithm remains an **open problem**.

> **Inconsistency between Theory vs. Empirical Completion (zr3T, PaJD)**

  We clarify that the theory focuses on **intrinsic limits under dual spectral sparsity, not task-specific guarantees**. Given that completion tasks primarily serve as *performance validation* and their guarantees necessitate *additional sampling assumptions* beyond our structural theory's scope, the theoretical extension to completion is listed as future work (App. E). **App. C.5** provides synthetic examples illustrating the link between the theory's key quantities and empirical patterns.

> **Insufficient Sensitivity Analysis and Choice of Parameter $(p,q)$ (zr3T, PaJD)**

We addressed sensitivity concerns via **extensive validation**. **Tab. 1** includes additional $(p,q)$ pairs with consistently stable performance. **App. D.15** expands this to a larger grid (**Tabs. 21–23; Figs. 8–10**) across multiple modalities and shows that accuracy remains stable over a consistent region of settings. A practical choice is given in **App. D.15**.

We also addressed several secondary questions by *streamlining the presentation* and clarifying the method's *robustness to transform sensitivity* (**Apps. D.5, D.9**), *choice of $\ell_{1/2}$ vs $\ell_1$* (**App. D.3**), *extension to higher-order tensors* (**App. D.12, Tab. 18**), and *structural uniqueness to $\ell_p$ and group sparsity*.

---

**3. Conclusion**

Our core contribution is a principled framework built on uncovering the structural Dual Spectral Sparsity in the foundational t-SVD modeling. Reviewers recognized the well-motivated modeling, rigorous theory, and consistent empirical gains.  In the rebuttal, we substantially clarified and addressed the key concerns, including the statistical evidence for the core assumption, the breadth of experiments, computational and convergence issues, parameter sensitivity, and the scope of the theory.

We believe these updates fully clarify the contribution of the work. We appreciate your careful consideration of our revision.

---

### Meta-Review · Area_Chair_17D2 · 2026-01-02

**Summary:**

This paper proposes a tensor ℓp-Schatten-q quasi-norm framework based on the t-SVD, which generalizes the traditional tensor nuclear norm and aims to jointly model intra-frequency low-rankness and inter-frequency sparsity through dual spectral regularization. The study provides a theoretical characterization based on minimax bounds and a reweighted optimization algorithm, and its performance improvement is empirically validated on the completion task across several hyperspectral datasets.

The reviewers acknowledged that the paper's strengths lie in its rigorous theoretical analysis and clear motivation. However, they also raised significant concerns. They found the core assumption, i.e.,  dual spectral sparsity patterns exist in the transformed (DCT) domain under the t-SVD framework, to be insufficiently supported, with the provided evidence being too limited to convincingly justify such a fundamental modeling assumption. The scenario considered in the paper is difficult to be regarded as a fundamental modeling assumption, yet as a solution for practical problems, it appears somewhat outdated, with limited technical innovation. Besides, the reviewers noted that the theoretical analysis was deemed largely abstract, reducing its applied value. The experimental validation was considered insufficient, and the presentation was criticized as mathematically dense and occasionally repetitive, affecting readability and clarity.

The Area Chair fully agrees with the core concerns raised by Reviewers zr3T and PaJD and believes these issues are substantial and difficult to address. Consequently, it is concluded that this paper does not meet the acceptance criteria.

**Reviewer Concerns:**

The following concerns I believe are still outstanding
(1) The core assumption of this paper—that dual spectral sparsity patterns exist in the transformed (DCT) domain under the t-SVD framework—is insufficiently supported. The only evidence provided is an empirical illustration using MRI and CT examples in Figure 1, which is too limited to convincingly justify such a fundamental modeling assumption.

(2) The theoretical analysis, while ambitious, remains largely abstract. The proofs are not clearly connected to practical convergence guarantees or algorithmic stability, reducing their applied value.

(3) The presentation is mathematically dense and occasionally repetitive, which affects readability and clarity. A more structured exposition with stronger empirical validation would substantially improve the paper’s impact and credibility.

(4) The experiments are insufficient.

**Reviewer Scores:**

I believe that the reviewers will not revise their scores.

---

### Decision · Program_Chairs · 2026-01-26

Reject